# An amphipathic helix in Brl1 is required for nuclear pore complex biogenesis in *S. cerevisiae*

**Annemarie Kralt[1†], Matthias Wojtynek[1,2†], Jonas S Fischer[1†], Arantxa Agote-Aran[1], Roberta Mancini[1], Elisa Dultz[1], Elad Noor[3], Federico Uliana[1], Marianna Tatarek-Nossol[4], Wolfram Antonin[4], Evgeny Onischenko[5], Ohad Medalia[2], Karsten Weis[1]\***

[1]Institute of Biochemistry, Department of Biology, ETH Zurich, Zurich, Switzerland; [2]Department of Biochemistry, University of Zurich, Zürich, Switzerland; [3]Department of Plant and Environmental Sciences, Weizmann Institute of Science, Rehovot, Israel; [4]Institute of Biochemistry and Molecular Cell Biology, Medical School, RWTH Aachen University, Aachen, Germany; [5]Department of Biological Sciences, University of Bergen, Bergen, Norway

**Abstract** The nuclear pore complex (NPC) is the central portal for macromolecular exchange between the nucleus and cytoplasm. In all eukaryotes, NPCs assemble into an intact nuclear envelope (NE) during interphase, but the process of NPC biogenesis remains poorly characterized. Furthermore, little is known about how NPC assembly leads to the fusion of the outer and inner NE, and no factors have been identified that could trigger this event. Here, we characterize the transmembrane protein Brl1 as an NPC assembly factor required for NE fusion in budding yeast. Brl1 preferentially associates with NPC assembly intermediates and its depletion halts NPC biogenesis, leading to NE herniations that contain inner and outer ring nucleoporins but lack the cytoplasmic export platform. Furthermore, we identify an essential amphipathic helix in the luminal domain of Brl1 that mediates interactions with lipid bilayers. Mutations in this amphipathic helix lead to NPC assembly defects, and cryo-electron tomography analyses reveal multilayered herniations of the inner nuclear membrane with NPC-like structures at the neck, indicating a failure in NE fusion. Taken together, our results identify a role for Brl1 in NPC assembly and suggest a function of its amphipathic helix in mediating the fusion of the inner and outer nuclear membranes.

*For correspondence:
karsten.weis@bc.biol.ethz.ch

†These authors contributed equally to this work

Competing interest: The authors declare that no competing interests exist.

## Editor's evaluation

The article makes an important advance in our understanding of the nuclear pore complex (NPC) biogenesis mechanism, which has remained a central challenge in the field. Specifically, a compelling combination of in vitro and in vivo data implicates Brl1 as an assembly factor that associates with nascent NPCs. An essential amphipathic helix in Brl1 binds to highly curved membranes in a manner that is required for inner and outer nuclear membrane fusion.

## Introduction

Virtually all biological processes are carried out by multiprotein complexes, and their faithful assembly is therefore crucial for cellular function (*Hartwell et al., 1999*). The nuclear pore complex (NPC) is one of the largest cellular protein complexes, with a total mass of 60–120 MDa. In all eukaryotes, NPCs perforate the double lipid bilayer of the nuclear envelope (NE) and mediate macromolecular exchange

between nucleus and cytoplasm (*Wente and Rout, 2010*). NPCs are assembled from multiple copies of ~30 different proteins known as nucleoporins (NUPs), which amount to hundreds of proteins in the mature complex due to the NPC's eightfold rotational symmetry (*Fernandez-Martinez and Rout, 2021*; *Lin and Hoelz, 2019*). NUPs are organized in well-defined subcomplexes (*Figure 1A*), where the membrane ring (MR), central channel (CC), and inner ring (IR) in the plane of the NE are sandwiched by two outer rings composed of Y-complexes. Asymmetrically attached to this scaffold are the cytoplasmic export platform (CP) and the nuclear basket (NB) (*Figure 1A*; *Fernandez-Martinez and Rout, 2021*; *Lin and Hoelz, 2019*).

The architecture of the NPC has recently been elucidated in great detail (*Akey et al., 2022*; *Bley et al., 2022*; *Huang et al., 2022a*; *Huang et al., 2022b*; *Li et al., 2022*; *Mosalaganti et al., 2022*; *Petrovic et al., 2022*; *Schuller et al., 2021*; *Tai et al., 2022*; *Zhu et al., 2022*; *Zimmerli et al., 2021*). Yet far less is known about how this gigantic complex assembles and gets embedded into the NE. In metazoan cells, which undergo an open mitosis, two types of NPC assembly mechanisms have been described: mitotic reassembly of NPCs at the end of cell division and de novo formation of NPCs during interphase (*Doucet et al., 2010*; *Otsuka and Ellenberg, 2018*; *Schooley et al., 2012*). Organisms that undergo closed mitosis, such as the budding yeast *Saccharomyces cerevisiae*, exclusively rely on interphase NPC assembly to create new NPCs (*Winey et al., 1997*). Here, NUP complexes punch a hole into the intact NE in order to create the protein-lined membrane tunnel that spans the NE. This requires a poorly understood fusion event between the inner nuclear membrane (INM) and outer nuclear membrane (ONM) during which the integrity of the NE diffusion barrier is not compromised (*Doucet and Hetzer, 2010*; *Rothballer and Kutay, 2013*).

NPC assembly events are rare (e.g., in yeast ~1 NPC forms every 2 min) (*Winey et al., 1997*) and capturing them in situ has been challenging. Therefore, NPC biogenesis has mainly been studied using genetic perturbations that inhibit its maturation. A shared phenotype of many NPC assembly mutants is the appearance of NE herniations, which likely correspond to halted NPC assembly intermediates (*Thaller and Patrick Lusk, 2018*). The orientation of these herniations – always bulging out towards the cytoplasm – suggests an inside-out mechanism of NPC assembly, which is also supported by observations of interphase assembly states in human cells (*Otsuka et al., 2016*). To characterize the precise maturation order and assembly kinetics of native NPC biogenesis in budding yeast, we recently developed a mass spectrometry-based approach that we termed KARMA (*Kinetic Analysis of Incorporation Rates in Macromolecular Assemblies*) (*Onischenko et al., 2020*). This revealed that NPCs form by sequential assembly of NUPs starting with the central scaffold, followed by the outer cytoplasmic and nucleoplasmic parts and concluded by the late binding of Mlp1, consistent with an inside-out assembly mechanism (*Onischenko et al., 2020*).

To date, very few non-NPC proteins have been shown to participate in NPC assembly. This is in contrast to, for example, ribosome biogenesis, where ~180 *trans*-acting assembly factors are known to interact during the maturation process. These are critical for ribosome assembly but are not part of the final structure (*Kressler et al., 2010*; *Strunk and Karbstein, 2009*). The few proteins suggested to promote interphase NPC assembly include the membrane-bending reticulons (*Dawson et al., 2009*), Torsin ATPases (*Laudermilch et al., 2016*; *Rampello et al., 2020*), the Ran GTPase and its regulators (*Ryan et al., 2003*), and, in budding yeast, a group of three small NE/ER-located transmembrane proteins: Brl1, its paralogue Brr6, and Apq12 (*de Bruyn Kops and Guthrie, 2001*; *Hodge et al., 2010*; *Lone et al., 2015*; *Saitoh et al., 2005*; *Scarcelli et al., 2007*; *Zhang et al., 2018*; *Zhang et al., 2021*). Temperature-sensitive alleles of *BRL1* and *BRR6* or deletion of *APQ12* show NE herniations, an altered cellular membrane composition, synthetic interactions with lipid biosynthesis pathways, and sensitivity to drugs influencing membrane fluidity (*Hodge et al., 2010*; *Lone et al., 2015*; *Scarcelli et al., 2007*; *Zhang et al., 2021*). Brl1, Brr6, and Apq12 can be co-immunoprecipitated, which suggests they form a complex (*Lone et al., 2015*), and they have been found to physically interact with NUPs (*Zhang et al., 2018*). Interestingly, overexpression of Brl1 but not Brr6 can bypass the function of Nup116 and Gle2 in NPC assembly (*Zhang et al., 2018*; *Liu et al., 2015*), suggesting that Brl1 and Brr6 act differently during NPC maturation.

Here, we take advantage of our KARMA method (*Onischenko et al., 2020*) to identify NPC biogenesis factors. We show that Brl1 transiently binds to immature NPCs and that depletion of Brl1 impairs NPC assembly, resulting in NE herniations that contain the central scaffold NUPs but lack the cytoplasmic export platform (Nup82, Nup159). We further identify an essential luminal amphipathic helix

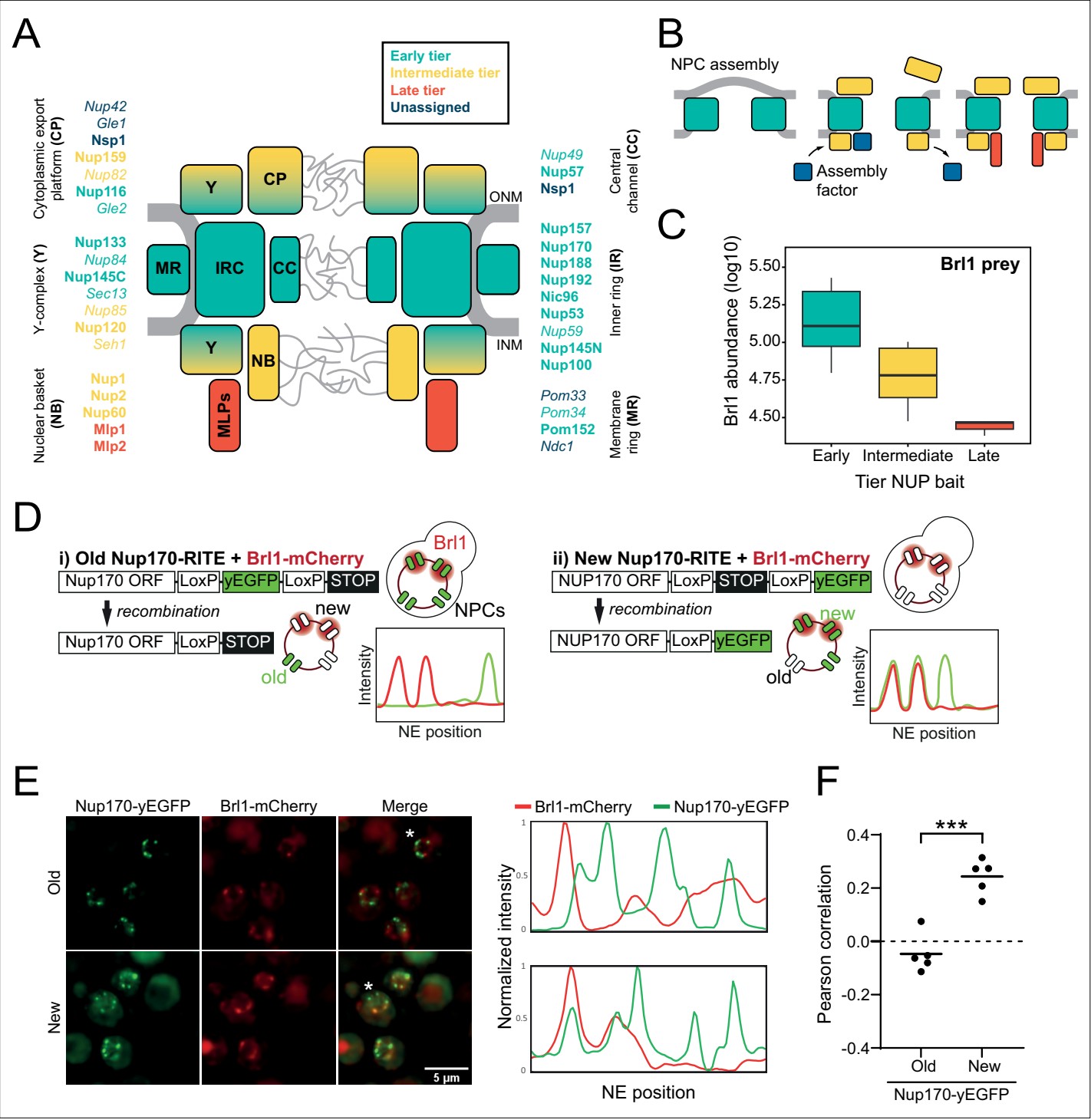

**Figure 1.** Brl1 preferentially binds young nuclear pore complexes (NPCs). (**A**) Scheme of the NPC architecture. The colors indicate the assembly order as found in *Onischenko et al., 2020*. Nucleoporins (NUPs) that were reproducibly identified in Brl1 affinity purifications are shown in bold. (**B**) Schematic illustrating the transient binding of an NPC assembly factor during NPC assembly. (**C**) Enrichment of Brl1 in affinity pulldowns from *Onischenko et al., 2020* using baits from the different assembly tiers. Early and intermediate tiers contain four different baits each; the late tier is represented by Mlp1 with three biological replicates for each bait. (**D**) Schematic representation of the recombination-induced tag exchange (RITE) strategy to visualize Brl1-mCherry co-localization with old or new NPCs marked by Nup170-yEGFP and the expected NE fluorescence intensity profiles. (**E**) Representative co-localization images of Brl1-mCherry with old or new Nup170-yEGFP marked NPCs using the RITE strategy described in (**D**). Cells were imaged ~30 min or ~5 hr after recombination induction, respectively. Fluorescence intensity profiles along the NE are displayed for cells denoted with an asterisk. (**F**)

*Figure 1 continued on next page*

*Figure 1 continued*

Pearson's correlation between Nup170-yEGFP and Brl1-mCherry fluorescence intensity profiles along the NE in (**E**). Individual points reflect the average of a biological replicate with a minimum of 28 analyzed NE contours per condition. Two-tailed Student's *t*-test (n = 5, p-value=0.00015).

The online version of this article includes the following source data and figure supplement(s) for figure 1:

**Source data 1.** Label-free Brl1 intensities with 10 nucleoporin (NUP) baits from *Onischenko et al., 2020*; related to *Figure 1C*.

**Source data 2.** Pearson's correlation coefficients for Brl1-mCherry and new or old Nup170-yeGFP.

**Figure supplement 1.** Proteomic characterization of Brl1 nuclear pore complex (NPC) interactions.

**Figure supplement 1—source data 1.** Label-free nucleoporin/nuclear transport receptor (NUP/NTR) intensities with Brl1 bait (related to *Figure 1—figure supplement 1B*).

(AH) in Brl1 that interacts with membranes and, when mutated, leads to the formation of large, multi-layered NE herniations containing immature NPCs that we structurally characterize by cryo-electron tomography (cryo-ET). Our results identify Brl1 as an essential NPC assembly factor and suggest that Brl1 mediates the fusion step between the INM and ONM during interphase NPC biogenesis via its AH.

## Results

### Brl1 binds to assembling nuclear pore complexes

Relying on a large KARMA dataset that contains kinetic interaction profiles for 10 different NUP baits, we recently demonstrated that yeast NPCs assemble sequentially, starting with the symmetrical core NUPs (early tier), followed by the majority of asymmetric NUPs (intermediate tier), and concluded by the assembly of two NB NUPs Mlp1 and Mlp2 (late tier) (*Figure 1A and B*; *Onischenko et al., 2020*). This analysis also identified a large number of non-NUP proteins that interact with the baits. We sought to exploit our dataset to uncover potential NPC assembly factors. Since such factors are expected to selectively bind to the NPC during its biogenesis but are not part of the mature structure, they should be enriched in early tier NUP pulldowns versus late tier ones (*Figure 1B*). Interestingly, out of ~1500 co-purified non-NUP proteins, Brl1 – a factor previously implicated in NPC biogenesis (*Lone et al., 2015*; *Zhang et al., 2018*) – displayed the second highest enrichment score (*Figure 1— figure supplement 1A*), decreasing in abundance approximately fivefold from early to late tier baits (*Figure 1C*). Only Her1, a protein with unknown biological function, had a higher early-to-late enrichment ratio. To confirm Brl1's binding preference for early assembling NUPs, we performed the reciprocal affinity pulldowns (APs) with endogenously tagged Brl1. In full agreement, early tier NUPs were enriched over the ones from intermediate and late assembly tiers (*Figure 1—figure supplement 1B*).

Brl1's preference for 'young' NPCs was validated by live-cell imaging using the recombination-induced tag exchange (RITE) approach (*Verzijlbergen et al., 2010*). We genetically tagged Nup170, which binds early during NPC biogenesis, with a RITE construct. This allowed us to specifically mark either old or newly synthesized Nup170 by removing or introducing a yEGFP-tag through inducible genetic recombination (*Figure 1D*). Since Nup170 binds early during NPC biogenesis, it can be assumed that some of the foci formed by newly synthesized Nup170-yEGFP represent NPC assembly intermediates. As a measure of Brl1 association with young and old NPCs, we monitored co-localization between Brl1-mCherry and either new or old Nup170-yEGFP using cross-correlation of the NE fluorescence signals. As evidenced by a higher cross-correlation score and in agreement with our proteomic data, Brl1 co-localized better with young than with old NPCs (*Figure 1E and F*). Together, these results indicate that Brl1 preferentially binds to young or immature NPCs, which is consistent with a function of Brl1 during NPC biogenesis.

Taking advantage of our KARMA workflow, we next set out to determine the stage during which Brl1 acts in NPC biogenesis more precisely. In KARMA, newly synthesized proteins are pulse labeled by heavy-isotope amino acids followed by the pulldown of the NPC via an endogenously tagged affinity bait at several post-labeling time points (*Figure 2A*; *Onischenko et al., 2020*). The extent of metabolic labeling of any co-isolated protein is indicative of its average age in the AP fraction (*Figure 2A*). Therefore, the 'young' structural intermediates that are bound by a bona fide assembly factor during biogenesis should display a higher metabolic labeling rate in APs compared to the labeling of bulk cellular proteins. By contrast, structural components that join after the assembly factor

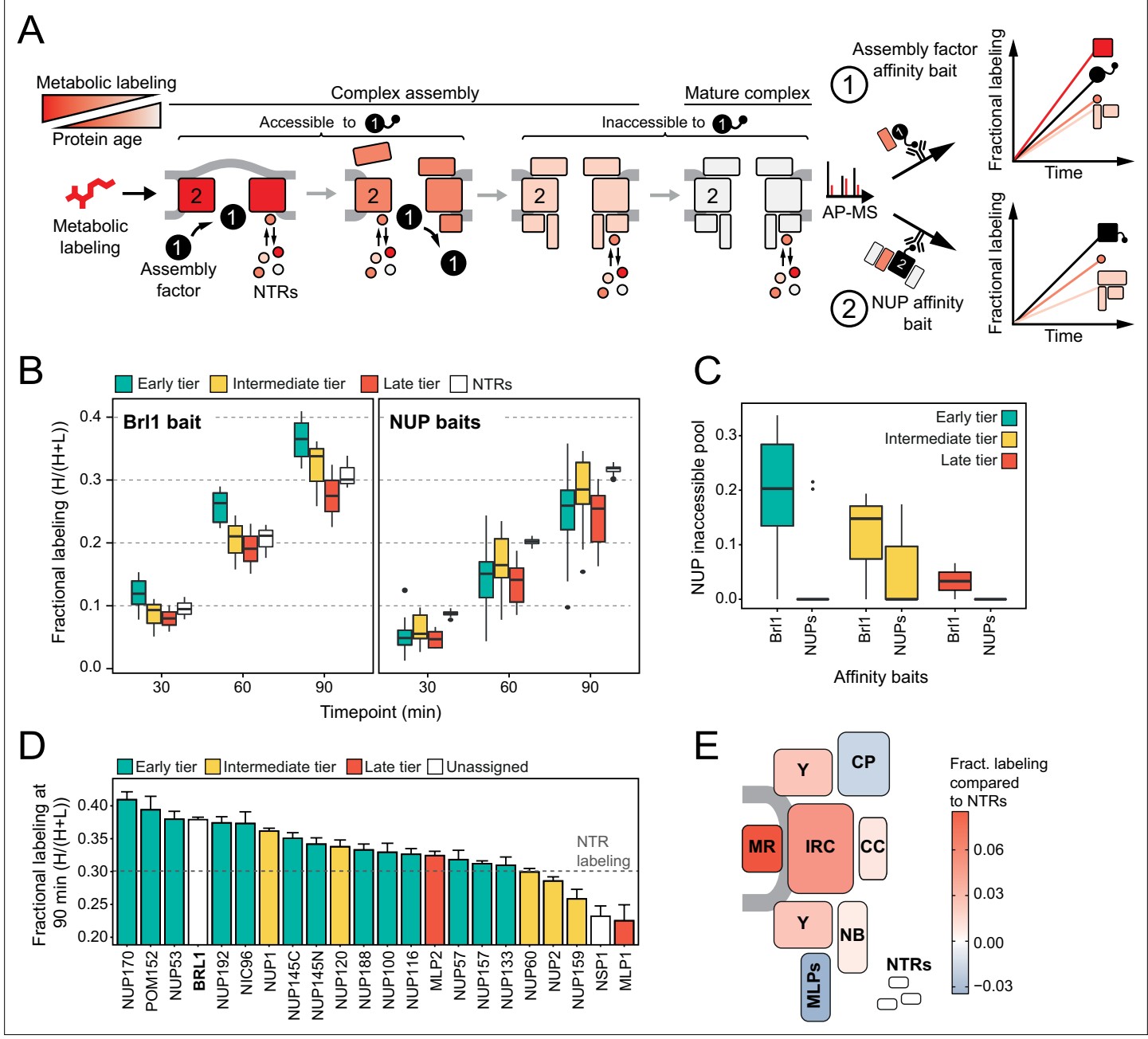

**Figure 2.** Mapping Brl1 association with nuclear pore complex (NPC) assembly intermediates using KARMA (*K*inetic *A*nalysis of *I*ncorporation *R*ates in *M*acromolecular *A*ssemblies). (**A**) Principles of KARMA: newly synthesized proteins are pulse-labeled followed by the affinity purification of the nucleoporin (NUP) complexes through a tagged NPC-binding protein. The extent of metabolic labeling is then quantified by mass spectrometry and corresponds to the average protein age in the affinity-purified fraction. An assembly factor selectively binds young NPCs, thus leading to high metabolic labeling rates for NUPs present in the intermediates (1). This is not the case for proteins that join after the assembly factor completely or partially dissociates or when the process is probed with an NUP bait (2). (**B**) Comparison of the labeling rates for NUPs and nuclear transport receptors (NTRs) in KARMA assays with Brl1 bait (left, this study) and with 10 different NUP baits (right, *Onischenko et al., 2020*). Median of three biological replicates. (**C**) Inaccessible pool of NUPs in KARMA assays with Brl1 compared to NUP baits (*Onischenko et al., 2020*), evaluated using a three-state kinetic state model (KSM) (*Onischenko et al., 2020*). (**D**) Barplot depicting the extent of metabolic labeling for different NUPs in KARMA assays with Brl1 bait after 90 min. The dotted line indicates the median NTR labeling. Median ± SD of three biological replicates. (**E**) Fractional labeling values from 2D averaged for NPC subcomplexes and offset by NTR labeling projected onto the NPC scheme.

The online version of this article includes the following source data for figure 2:

**Source data 1.** Fractional labeling of nucleoporins/nuclear transport receptors (NUPs/NTRs) in Brl1 KARMA (*K*inetic *A*nalysis of *I*ncorporation *R*ates in *M*acromolecular *A*ssemblies) assays (related to *Figure 2* and *Figure 1—figure supplement 1*).

*Figure 2 continued on next page*

*Figure 2 continued*

**Source data 2.** Inaccessible pools as estimated by the three-step kinetic state model (KSM) (*Onischenko et al., 2020*) for Brl1 and NUP baits (related to *Figure 2C*).

has left the NPC assembly site are not expected to show this effect, even if the assembly factor does not dissociate completely (*Figure 2A*). Nuclear transport receptors (NTRs) that bind the NPC highly transiently serve as a reference for bulk cellular protein labeling to discriminate between young and old proteins.

In KARMA assays with endogenously tagged Brl1, we were able to detect most NUPs (*Figure 1A*) with highly reproducible labeling readouts between biological replicates (*Figure 1—figure supplement 1C and D*). Strikingly, the NUP labeling rates observed with Brl1 as bait were overall significantly higher compared to the ones in KARMA assays with NUP baits (*Onischenko et al., 2020*; *Figure 2B*). On top, we observed that in Brl1 pulldowns, early tier NUPs were labeled outstandingly fast, exceeding NUPs from the intermediate or late tiers and even the NTRs – our reference of the bulk cellular proteins (*Figure 2B and D*, *Figure 1—figure supplement 1D*). In line with this, our quantitative analysis of NUP metabolic labeling rates using a previously developed kinetic state model (KSM) (*Onischenko et al., 2020*), revealed that early tier NUPs become partly inaccessible to the Brl1 bait in mature NPCs (*Figure 2C*; see section 'Kinetic state modeling' in Appendix 1), suggesting that Brl1 dissociates at later stages of NPC assembly (*Figure 1B*). Although most NUPs from the late and intermediate tier were still detected in the KARMA assays, they did not display elevated labeling rates and even showed significant labeling delays as in the case of Mlp1, Nup159, and Nsp1 (*Figure 2D*). Altogether, these results show that Brl1 preferentially binds NPC assembly intermediates that are composed of the central scaffold (early tier) but lack the peripheral nucleoplasmic and cytoplasmic structures (intermediate and late tier) (*Figure 2E*). Of note, the labeling differences we observed cannot be explained by variations in NUP turnover as evidenced by the analysis of NUP labeling rates in the source cell lysates (see section 'Analysis of protein labeling in source cell lysates' in Appendix 1). Despite the different labeling rates in KARMA assays, the analysis of Brl1 APs from mixtures of labeled and unlabeled yeast lysates showed almost complete intermixing of Brl1-bound NUP complexes during the AP procedure, pointing to a highly dynamic binding of Brl1 to NPC assembly intermediates (for details, see section 'Lysate intermixing assays' in Appendix 1).

## Depletion of Brl1 interferes with NPC maturation

Having established that Brl1 interacts with immature NPCs, we wanted to elucidate how the absence of Brl1 affects NPC assembly. Since Brl1 is encoded by an essential gene, we used the auxin-inducible degron (AID) system, which allows for the acute depletion of proteins (*Figure 3—figure supplement 1A*; *Nishimura et al., 2009*). Upon addition of auxin, ~65% of Brl1 was rapidly degraded within 15 min (*Figure 3—figure supplement 1B*), leading to a reduction in growth rate (*Figure 3—figure supplement 1C*). To characterize whether Brl1 degradation affected the NPC ultrastructure, we treated cells for 4–4.5 hr with auxin and then subjected them to cryo-focused ion beam (FIB) milling and cryo-electron tomography (cryo-ET). As expected, we found mature NPCs (*Figure 3Aii*, white arrow, and *Figure 3—video 1*) in the NE of auxin-treated cells, but also detected small electron-dense INM evaginations (*Figure 3Aiii* and *Figure 3—video 2*) along the NE. Additionally, we observed that Brl1-depleted cells have electron-dense NE herniations (*Figure 3A*, black arrows, and *Figure 3—videos 1 and 2*) similar to the ones commonly observed in NPC assembly mutants (*Thaller and Patrick Lusk, 2018*) and previously also seen for Brl1/Brr6 double-depleted cells (*Zhang et al., 2018*). In our control strain lacking the auxin receptor *OsTir1*, no herniations could be detected after auxin treatment (*Figure 3B*, *Figure 3—videos 3 and 4*). However, we infrequently observed INM evaginations (*Figure 3B*, *Figure 3—video 3*), indicating that these could represent regular NPC intermediates.

Interestingly, the herniations that we observed upon Brl1 degradation were often clustered and enclosed by a continuous ONM (*Figure 3Aiii and iv*, *Figure 3—videos 1 and 2*). Closer inspection revealed densities likely corresponding to the IR (*Figure 1A*) at the apex of the INM (*Figure 3—figure supplement 1Dii*). Subtomogram averaging and single subtomograms of the NE herniations also indicate the presence of a nucleoplasmic density, presumably corresponding to the nucleoplasmic Y-complex ring as previously reported by Allegretti and coworkers (*Figure 3—figure supplement 1Dii*; *Allegretti et al., 2020*). While the subtomogram averaging of INM evaginations did not

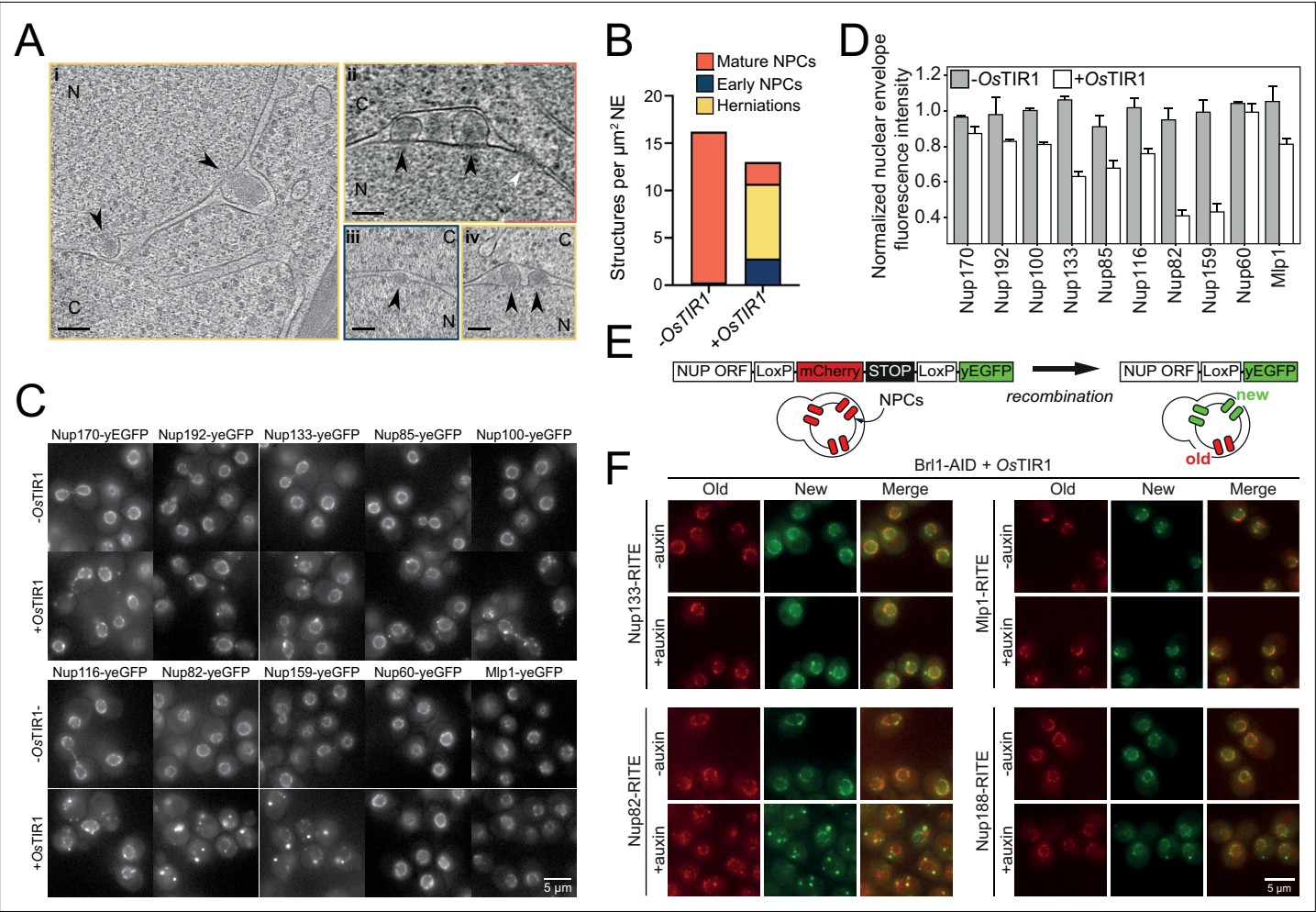

**Figure 3.** Brl1 degradation interferes with nuclear pore complex (NPC) assembly. (**A**) Tomographic slices of focused ion beam (FIB)-milled, 4–4.5 hr auxin-treated Brl1-AID cells showing the structures quantified in (**B**). Image frames colored according to the color code used in (**B**). Scale bar 100 nm; black arrows: herniations; white arrow: NPC; N: nucleus; C: cytoplasm; slice thickness **i** and **iii**: 1.4 nm; **ii** and **iv**: 2.8 nm. Panels **i** and **ii** were cropped from tomographic slices from the tomograms in *Figure 3—videos 1 and 2*. (**B**) Quantification of 27 tomograms (8.5 µm² NE) and 51 (16.7 µm² NE) for -*Os*TIR1 and +*Os*TIR1, respectively. (**C**) Example fluorescent micrographs of yEGFP-tagged nucleoporins (NUPs) in 4–4.5 hr auxin-treated Brl1-AID ± *Os*TIR1 cells. (**D**) Normalized fluorescence intensity signal in the nuclear envelope in ±*Os*TIR1 Brl1-AID cells treated with 500 µM auxin for 4–4.5 hr. Mean ± SEM of a minimum of two biological replicates. (**E**) Recombination-induced tag exchange (RITE) method is combined with a CRE-EBD recombinase to conditionally switch fluorescence tags upon β-estradiol addition. (**F**) NUP RITE fusion protein localization in the Brl1-AID background 3 hr after treating cells with auxin (+auxin) or ethanol (-auxin). Recombination was induced 30 min prior to auxin addition.

The online version of this article includes the following video, source data, and figure supplement(s) for figure 3:

**Figure supplement 1.** Characterization and subtomogram analysis of Brl1 depletion.

**Figure supplement 1—source data 1.** Uncropped Western blots (related to *Figure 3—figure supplement 1B*, *Figure 5E and F*).

**Figure 3—video 1.** Sequential sections of a cryo-tomogram from Brl1-depleted cells.
https://elifesciences.org/articles/78385/figures#fig3video1

**Figure 3—video 2.** Sequential sections of a cryo-tomogram from Brl1- depleted cells.
https://elifesciences.org/articles/78385/figures#fig3video2

**Figure 3—video 3.** Sequential sections of a cryo-tomogram from non-depleted Brl1 control cells.
https://elifesciences.org/articles/78385/figures#fig3video3

**Figure 3—video 4.** Sequential sections of a cryo-tomogram from non-depleted Brl1 control cells.
https://elifesciences.org/articles/78385/figures#fig3video4

reveal distinct densities likely because of their high heterogeneity and the limited number of analyzed subtomograms, the average of mature NPCs extracted from the same dataset displayed a similar architecture as previously reported in higher resolution subtomogram averages (*Akey et al., 2022*; *Allegretti et al., 2020*; *Figure 3—figure supplement 1Dii and iii*, *Figure 3—figure supplement 1E*). Occasionally, we also observed luminal densities at the herniations, probably corresponding to the Pom152 luminal ring (*Akey et al., 2022*; *Zimmerli et al., 2021*; *Upla et al., 2017*; *Figure 3—figure supplement 1F*). This is in line with our KARMA data, suggesting that Pom152 is already present in assembling NPCs prior to Brl1 recruitment (*Figure 2D and E*, *Figure 1—figure supplement 1D*).

To further characterize the composition of the NPC intermediates in Brl1-depleted cells, we investigated the localization of yEGFP-tagged Nups after auxin addition (*Figure 3C and D*). Consistent with our EM data, the IR complex NUPs (Nup170 and Nup192), the Y-complex members (Nup133 and Nup85), and linker NUPs (Nup100 and Nup116) retained a prominent NE localization, while the

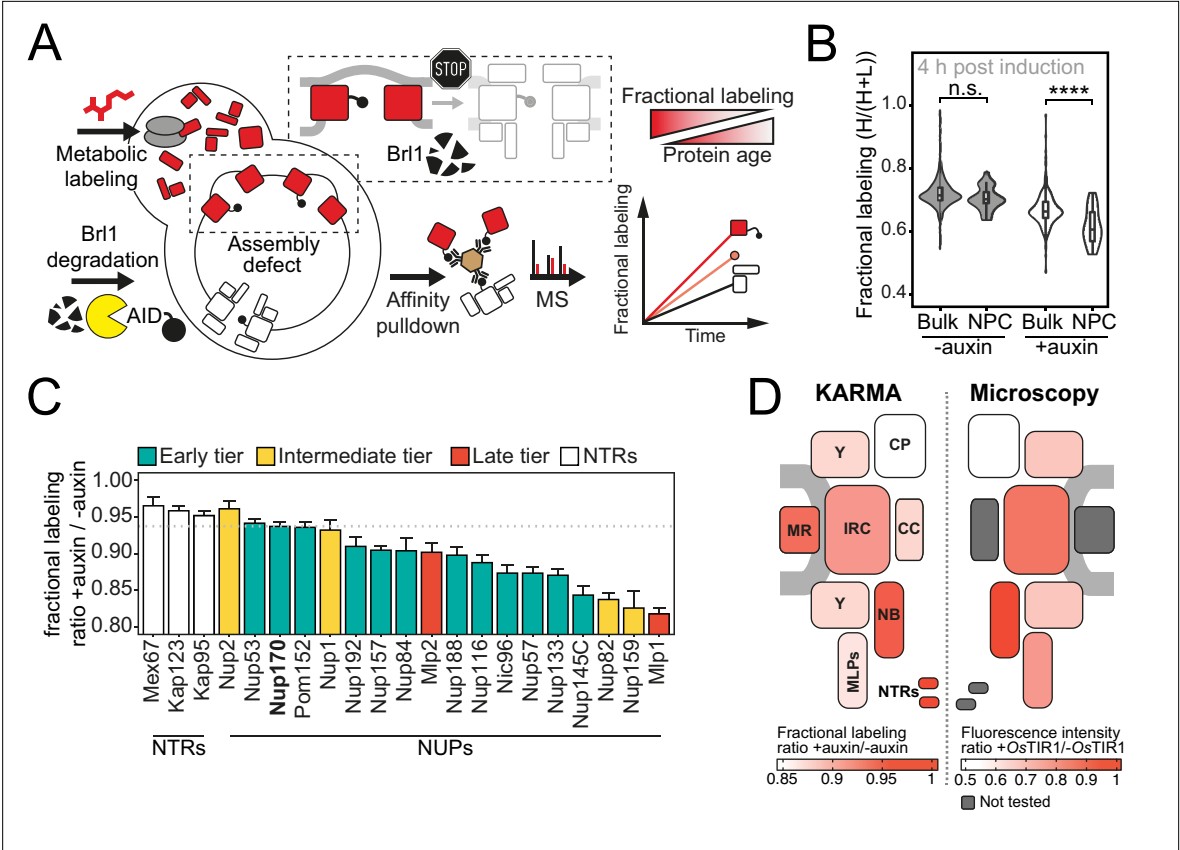

**Figure 4.** Proteomic characterization of nuclear pore complex (NPC) assembly intermediates induced by Brl1 depletion. (**A**) Depiction of the metabolic labeling assays to examine NPC assembly effects that occur upon Brl1 degradation. Newly synthesized proteins are pulse-labeled simultaneously with the auxin-induced depletion of Brl1. Mature NPCs and assembly intermediates are purified via affinity-tagged Nup170. Newly made nucleoporins (NUPs) that depend on Brl1 for their incorporation cannot be purified with Nup170, thus diminishing the extent of their metabolic labeling in Nup170 affinity pulldown (AP) after Brl1 depletion. (**B**) Fractional labeling of bulk proteins compared to NUPs in KARMA (*Kinetic Analysis of Incorporation Rates in Macromolecular Assemblies*) assays with affinity-tagged Nup170 in Brl1-AID cells treated with auxin (+auxin) or ethanol (-auxin) for 4 hr. Data points correspond to the median values in three biological replicates. Two-tailed Student's *t*-test (p-value: n.s. >0.05 and ****<0.0001). (**C**) Fractional labeling ratio of NUPs (bars) and bulk proteins (dotted line) in Nup170 APs from Brl1-AID cells treated with auxin (+auxin) or ethanol (-auxin). Mean ± SEM of three biological replicates and three time points (4, 4.5, and 5 hr post treatment, n = 9). Mlp1 and Mlp2 are missing in one replicate of the 4.5 time point (n = 8). (**D**) Left: fractional labeling ratios from (**C**) averaged per subcomplex and projected onto the NPC schematic. Right: nuclear envelope fluorescence intensity signal ratio from *Figure 3D* averaged for NPC subcomplexes and projected onto the NPC schematic.

The online version of this article includes the following source data and figure supplement(s) for figure 4:

**Source data 1.** Fractional labeling in Brl1-AID Nup170 affinity pulldowns (APs) with and without auxin treatment (related to *Figure 4*).

**Figure supplement 1.** Exchange rates of nuclear pore complex (NPC) assembly intermediates in Brl1-depleted cells.

**Figure supplement 1—source data 1.** Fractional labeling in the lysis intermixing tests for Nup170 bait (related to *Figure 4—figure supplement 1*).

cytoplasmic export platform NUPs (Nup82 and Nup159) were mislocalized in bright foci. Interestingly, the NB NUPs (Nup60 and Mlp1) also readily localized at the NE. We thus conclude that NPC structures that accumulate upon Brl1 depletion contain the central scaffold and the NB structure but lack the cytoplasmic face of the NPC (*Figure 4D*, right).

To exclude that mature NPCs are affected by the depletion of Brl1, we monitored NUPs synthesized before and after Brl1 depletion separately using RITE (*Figure 3E*; *Verzijlbergen et al., 2010*). New Mlp1, Nup133, and Nup188 mainly still localized to the NE homogeneously, with occasional single foci observed for new Nup133 and Nup188. At the same time, newly synthesized Nup82 formed multiple and bright foci either in the cytoplasm or NE (*Figure 3F*). By contrast, the localization of old proteins was not affected for any tested NUP. Together, our results reveal that removal of Brl1 triggers the formation of NE herniations as a consequence of halted NPC assembly, whereas previously assembled NPCs are not affected by the lack of Brl1.

To systematically explore the composition of the NPC assembly intermediates that accumulate in the absence of Brl1, we once more employed metabolic labeling coupled to affinity purification mass spectrometry. We used Nup170 as an affinity bait since it binds early during NPC maturation (*Onischenko et al., 2020*), enabling us to purify both mature NPCs and intermediate structures upon Brl1 depletion (*Figure 4A*). To this end, we pulse-labeled newly synthesized proteins in parallel with the induction of Brl1 degradation, and subsequently quantified the metabolic labeling for all co-purified proteins. For NUPs that are able to assemble into intermediates in the absence of Brl1, we expect to find a mixture of unlabeled (old) and labeled (new) proteins in Nup170 APs. However, for NUPs dependent on Brl1 for their assembly, only pre-assembled, old proteins will be captured. Thus, proteins dependent on Brl1 for their incorporation are expected to have slower labeling rates (*Figure 4A*).

In Brl1-depleted cells, the metabolic labeling of NUPs was generally slower than for the bulk of co-purified proteins. Such a delay was not observed in control cells, implying that the NPC maturation process is affected when Brl1 is depleted (*Figure 4B*). Importantly, the labeling delay was not identical for all NUPs (*Figure 4C*). While most MR, NB, and IR complex NUPs were labeled comparable to the dynamic NTRs, the cytoplasmic export platform NUPs and Mlp1 incorporated labeling substantially slower (*Figure 4D*, left). This is in agreement with the densities observed by cryo-ET and corroborates that the observed herniations are indeed incomplete NPC assembly intermediates that have not yet acquired the cytoplasmic structure and that Mlp1 is recruited very late to the NPC. Of note, the differences in NUP labeling observed upon Brl1 depletion with Nup170 correlate well with the labeling rates in KARMA assays with Brl1 bait (*Figure 4—figure supplement 1A*). This indicates that most NUPs that assemble after the Brl1-dependent assembly step (slow labeling in KARMA assays with the Brl1 bait) can no longer incorporate into the NPC once Brl1 is degraded (slow labeling in KARMA assays when Brl1 is depleted).

Of note, the metabolic labeling of the bulk of co-purified proteins was also overall delayed upon Brl1 depletion (*Figure 4B*). This is consistent with the decreased growth rate that can be observed in these conditions (*Figure 3—figure supplement 1C*). Interestingly, the lysate intermixing assays showed a significantly higher degree of NUP exchange during the AP procedure in the Nup170 APs when Brl1 was depleted (*Figure 4—figure supplement 1B–D*, section 'Lysate intermixing assays' in Appendix 1). This might suggest that the immature NPCs that accumulate in the absence of Brl1 are less stable than fully assembled NPCs.

## Brl1 contains an essential luminal AH

So far, our analyses showed that Brl1 is an NPC assembly factor: it predominantly interacts with immature NPCs preceding incorporation of the cytoplasmic export platform and its depletion leads to the formation of NE herniations with a continuous ONM, suggesting that Brl1 may act prior to INM-ONM fusion during NPC maturation. We therefore wanted to mechanistically understand how Brl1 promotes NPC biogenesis. Brl1 is composed of a long unstructured N-terminus and two transmembrane domains linked by a luminal domain, which contains four cysteines that form two disulfide bridges (*Figure 5A and C*, *Figure 5—figure supplement 1D–G*; *Zhang et al., 2018*). Such a structural organization was also predicted by AlphaFold (*Figure 5A*, *Figure 5—figure supplement 1*; *Jumper et al., 2021*). The structured part of Brl1 containing the transmembrane and luminal region was predicted with high-confidence scores and agree well with previous experimental findings (*Saitoh*

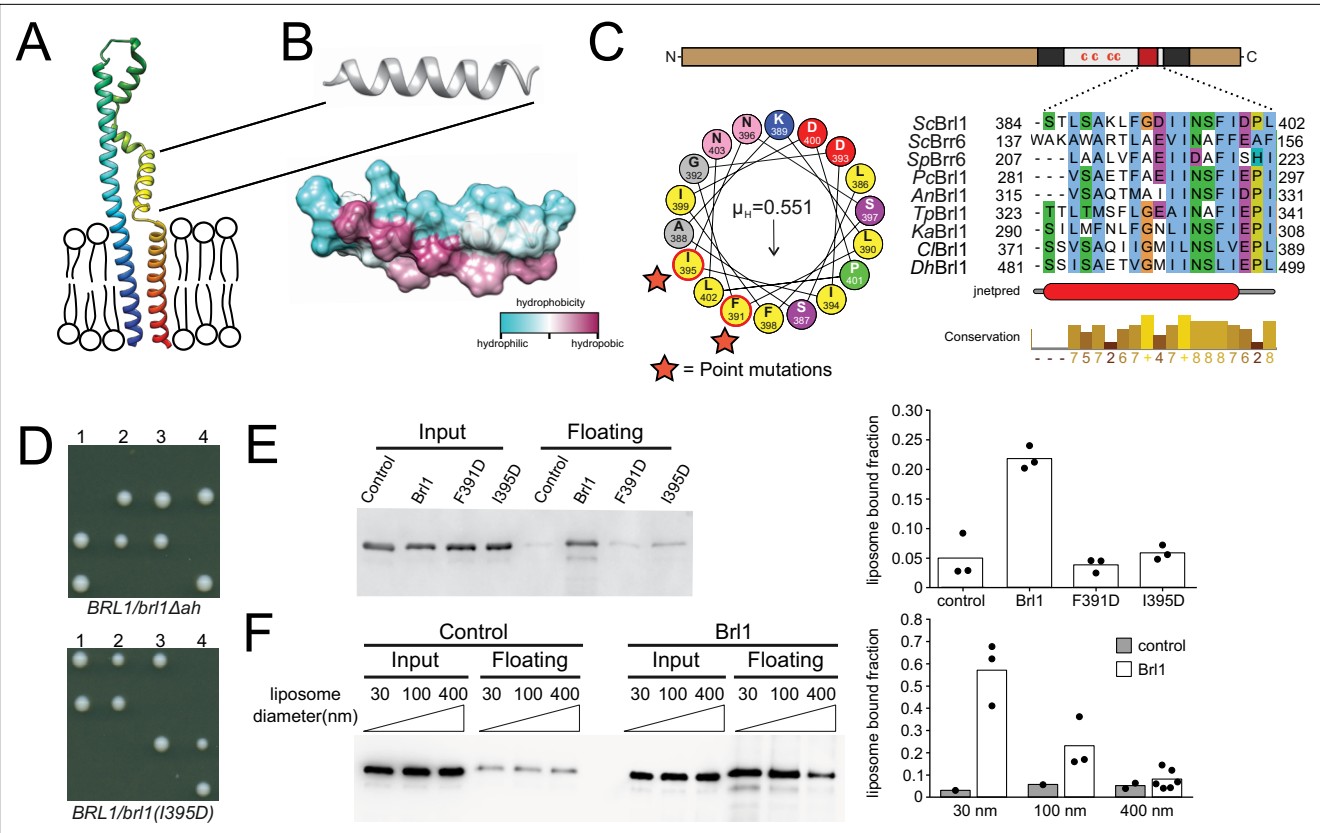

**Figure 5.** A conserved luminal amphipathic helix binds to membranes and is essential for Brl1 function. (**A**) AlphaFold prediction for Brl1 (*Jumper et al., 2021*). Unstructured termini are not shown; blue: N-terminus; red: C-terminus. Transmembrane domain highlighted by the lipid bilayer. (**B**) Predicted amphipathic helix in ribbon and surface representation, colored based on hydrophobicity. (**C**) Upper panel: domain architecture of Brl1: extraluminal N- and C-terminus in brown, transmembrane domains in dark gray, amphipathic helix in red; left panel: helical wheel representation of the amphipathic helix of Brl1 and the hydrophobic moment determined with HeliQuest (*Gautier et al., 2008*). Point mutants are indicated by stars. Right panel: conservation and secondary structure prediction of the amphipathic helix in different fungi (full alignment in *Figure 5—figure supplement 2A and B*). Hydrophobic: blue; negative: magenta; polar: green; glycine: orange; proline: yellow; unconserved: white. Jnetpred4 secondary structure prediction (*Drozdetskiy et al., 2015*): helices are marked as red tubes. *Sc: Saccharomyces cerevisiae; Sp: Schizosaccharomyces pombe; Pc: Pneumocystis carinii; An: Aspergillus nidulans; Tp: Tetrapisispora phaffii; Ka: Kazachstania africana; Cl: Clavispora lusitaniae; Dh: Debaryomyces hansenii.* (**D**) Vertically oriented tetrad offspring of heterozygous Brl1 mutants carrying one allele lacking the amphipathic helix (*brl1Δah*) or a single-point mutation in the hydrophobic side of the helix (*brl1(I395D)*). (**E**) Membrane flotation assay with purified MBP-ahBrl1(377-406)-yEGFP fusion proteins and liposomes made of *E. coli* polar lipids extract. Control: MBP-yeGFP. Mean of three biological replicates, individual data points are indicated. (**F**) Membrane flotation assay with purified MBP-ahBrl1(377-406)-yEGFP fusion using liposomes of different sizes. Control: MBP-yeGFP. Mean of three or six biological replicates for MBP-ahBrl1(377-406)-yEGFP, individual data points are indicated.

The online version of this article includes the following figure supplement(s) for figure 5:

**Figure supplement 1.** AlphaFold structure prediction for Brl1.

**Figure supplement 2.** Conservation and structure prediction of Brl1 homologues in different fungi.

---

et al., 2005; *Zhang et al., 2018*). The N- and C-terminus, on the other hand, had poor prediction scores, as expected for natively disordered regions (*Figure 5—figure supplement 1A–C*). Closer inspection of the predicted Brl1 structure revealed an AH just upstream of the second transmembrane domain (*Figure 5A–C*), which was also suggested by the AH prediction algorithm HeliQuest (*Gautier et al., 2008*; *Figure 5C*).

Amphipathic helices are short motifs capable of binding lipid bilayers, and they have been implicated in bending membranes by inserting into one leaflet of a bilayer, generating a convex curvature (*Ford et al., 2002*; *Wang et al., 2016*). Interestingly, AHs are structural features of many membrane-binding NUPs (*Hamed and Antonin, 2021*) and likely target NUPs to the NPC by curvature sensing (*Floch et al., 2015*). The transmembrane domains, luminal region, and AH in Brl1 (ahBrl1) are highly conserved between organisms with closed mitosis (*Figure 5—figure supplement 2A–C*). The

conservation of ahBrl1 might suggests that it plays a critical role in NPC biogenesis, for example, by mediating the INM-ONM fusion. Indeed, in tetrad dissections of heterozygous yeast strains carrying a mutant allele of *BRL1* either lacking the AH (*brl1Δah*) or disrupting the AH (*brl1(I395D)*), only the two spores that carried the wildtype allele were viable (*Figure 5D*). This shows that ahBrl1 is essential for the function of Brl1 and cell viability.

We hypothesized that ahBrl1 might contribute to the INM-ONM fusion step in NPC biogenesis through interaction with membranes. We therefore tested the membrane-binding capacity of ahBrl1 in vitro using a liposome flotation assay, where we incubated liposomes generated from *E. coli* polar lipid extract with a recombinant MBP-ahBrl1-yEGFP fusion protein (*Figure 5E*). We observed that MBP-ahBrl1-yEGFP was enriched in the floating fraction, whereas fusion proteins that carry single-point mutations disrupting the hydrophobic face of ahBrl1 (F391D and I395D) displayed strongly reduced liposome binding compared to the negative control MBP-yEGFP (*Figure 5E*). Fusion of lipid membranes is typically accompanied by the formation of highly curved fusion intermediates. We therefore tested the affinity of ahBrl1 to membranes with different curvature in a liposome flotation assay with liposomes of various sizes (*Figure 5F*). Indeed, we observed that MBP-ahBrl1-yEGFP preferentially binds to highly curved liposomes with a diameter of 30 nm. Enrichment with 100 nm liposomes was less pronounced and binding to 400 nm liposomes did not exceed the level of the non-lipid-binding control construct (MBP-yEGFP).

Together, these results demonstrate that ahBrl1 binds to highly curved lipid membranes in vitro and is essential for cell viability.

## Overexpression of Brl1(I395D) blocks NPC maturation and leads to herniating INM sheets at NPC assembly site

Since ahBrl1 is required for Brl1's function, we wanted to elucidate its role during NPC assembly. Previously, it was reported that overexpression of Brl1 bypasses the requirements for Nup116 and Gle2 in NPC biogenesis (*Zhang et al., 2018*; *Liu et al., 2015*). We screened the effect of six single-point mutations in ahBrl1 for the ability to rescue growth of the *nup116ΔGLFG P_MET3-NUP188* strain (*Figure 6—figure supplement 1A*). We observed that overexpression of Brl1 mutants, replacing the hydrophobic residues F391, I395, F398, or L402 by the charged aspartic acid, not only failed to rescue the assembly defect but had a dominant negative effect on cell growth (*Figure 6A*, *Figure 6—figure supplement 1A*). When residues at the polar side of the helix (D393 and D400) were substituted to alanine, functionality was not perturbed (*Figure 6—figure supplement 1A*). The dominant negative growth inhibition persisted in the wildtype background (*Figure 6A*), demonstrating that overexpression of Brl1 with an impaired AH alone is toxic.

To understand the causes of the dominant negative effect of ahBrl1 mutant overexpression, we examined the localization of yEGFP-fused Brl1, Brl1Δah, and Brl1(I395D) expressed under a galactose-inducible promoter (*Figure 6B*). Brl1Δah and Brl1(I395D) initially localized to the NE-ER network, occasionally forming bright foci at the NE. However, after 6 hr of expression most of the protein was localized in large NE accumulations (*Figure 6B*). In contrast, overexpression of Brl1 with an unperturbed AH uniformly localized to the NE and ER (*Figure 6B*), as also shown previously (*Saitoh et al., 2005*; *Zhang et al., 2018*). The accumulation of overexpressed Brl1(I395D) was not dependent on endogenous Brl1, as there was no difference in the fraction of cells with NE foci of Brl1(I395D), irrespectively of whether endogenous Brl1 was inducibly degraded or not (*Figure 6—figure supplement 1C*).

Since wildtype Brl1 is unable to fulfill its function upon overexpression of the ahBrl1 mutants, we also wanted to analyze the localization of the endogenous copy of Brl1 in these conditions. Interestingly, we found that yEGFP-tagged Brl1 co-localized with the large BrlI395D-mCherry puncta at the NE (*Figure 6—figure supplement 1B*). This suggests that a sequestration of endogenous Brl1 to these accumulations could potentially lead to the dominant negative effect of the ahBrl1 mutants and that a critical concentration of Brl1 with a functional AH is needed for successful membrane fusion at NPC assembly sites. The dominant negative growth defect of overexpressed ahBrl1 mutants could thus be caused by the formation of toxic assemblies, which also trap the endogenous Brl1 protein.

To test whether Brl1(I395D) can dynamically exchange between NE accumulations or is trapped there, we probed the dynamics of Brl1(I395D)-mCherry at the herniations with fluorescence recovery after photobleaching (FRAP) (*Figure 6C*). We co-expressed either Brl1-mCherry or Brl1(I395D)-mCherry

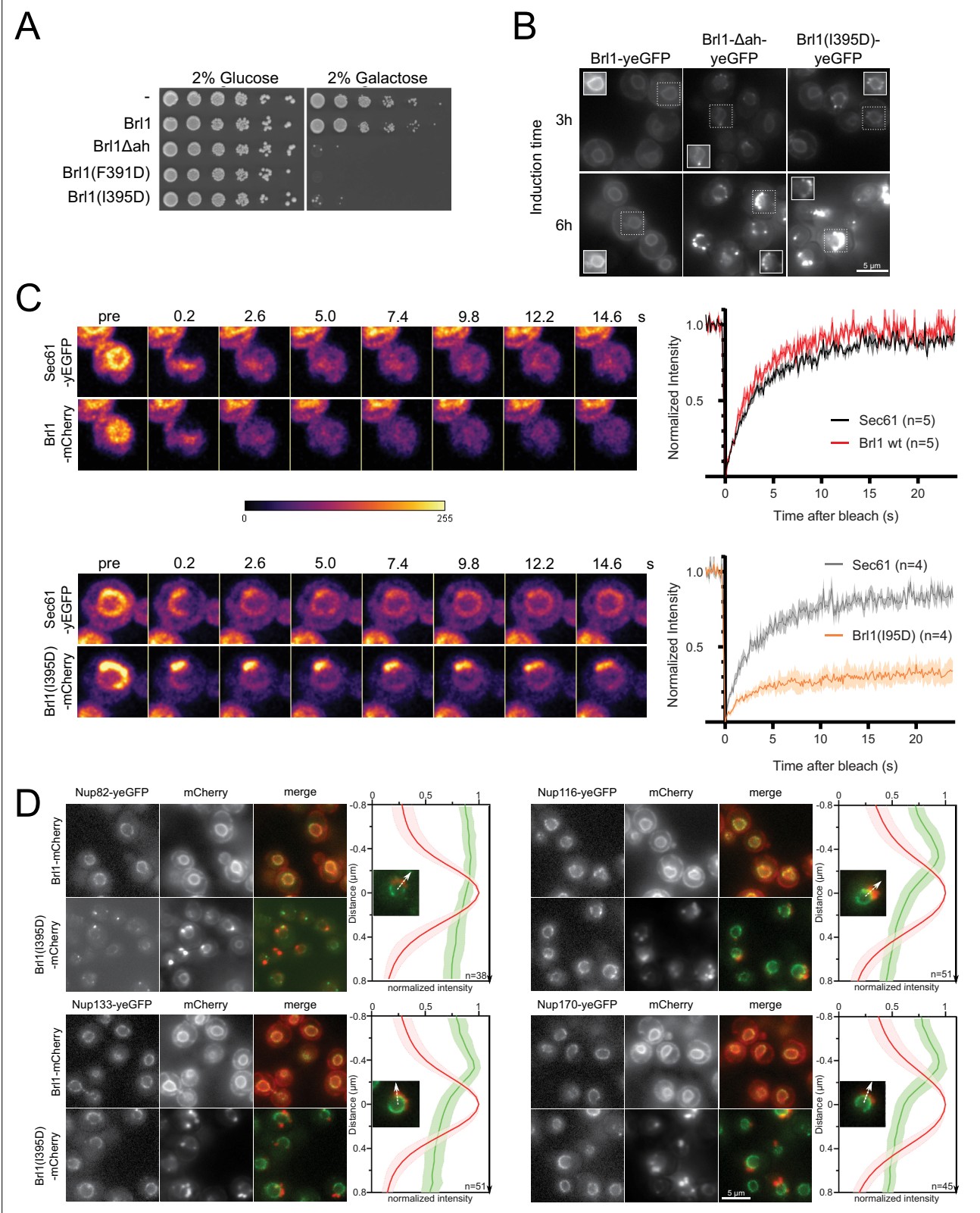

**Figure 6.** Overexpression of Brl1(I395D) with an impaired amphipathic helix interferes with nuclear pore complex (NPC) assembly. (**A**) Spotting assay of wildtype cells expressing Brl1, Brl1Δah, or Brl1(I395D) from the GAL1 promoter in glucose or galactose-containing medium. (**B**) Localization of yEGFP-tagged Brl1, Brl1Δah, or Brl1(I395D) from the GAL1 promoter in SD 2% galactose. Brightness contrast settings of nuclei in insets are adjusted differently. (**C**) Fluorescence recovery after photobleaching of Sec61-yEGFP, Brl1-mCherry and Brl1(I395D)-mCherry. Left panels: representative images of recovery;

*Figure 6 continued on next page*

*Figure 6 continued*

right: corresponding averaged recovery curves (n > 4). One representative experiment of three biological replicates is shown. Images are shown in pseudocolor. (**D**) Co-localization of mCherry-tagged Brl1 or Brl1(I395D) and yEGFP-tagged NUPs: mCherry channel is scaled differently between images. Maximum intensity plots of nucleoporins (NUPs) (green lines) relative to maximum Brl1(I395D)-mCherry signal in nuclear envelope (NE) foci (red line) from cytoplasm (bottom) to nucleoplasm (top). The arrows in the inset and the x-axis indicate the direction of measurement. Average and standard deviation of more than 38 line plots with n > 31 values averaged for each point. A representative image used for the analysis is shown for each condition in the inset.

The online version of this article includes the following source data, source code, and figure supplement(s) for figure 6:

**Source code 1.** Fluorescence recovery after photobleaching (FRAP) analysis.

**Source data 1.** Line plots.

**Figure supplement 1.** Luminal amphipathic helix (AH) of Brl1 is involved in nuclear pore complex (NPC) biogenesis.

with Sec61-yEGFP, a transmembrane protein, that can freely diffuse between the ER/ONM and the INM (*Deng and Hochstrasser, 2006*; *Popken et al., 2015*). We compared the fluorescence recovery of Brl1-mCherry with Sec61-yEGFP in an arbitrary NE region and saw that both proteins fully recover with a comparable half-life ($\tau_{1/2}$) of ~2 s, indicating that they freely diffuse in the membrane of the NE (*Figure 6C*). This is in line with the results of our lysate intermixing experiments pointing to a highly dynamic binding of Brl1 to NPC assembly intermediates (see section 'Lysis intermixing assays' in Appendix 1" and *Appendix 1—figure 1B–D*). Next, we photobleached the fluorescent signal of Brl1(I395D)-mCherry and Sec61-yEGFP in the NE-attached foci and observed that Brl1(I395D)-mCherry has a high immobile fraction that is not replaced over the time scale of 25 s, while Sec61-yEGFP almost fully recovered (*Figure 6C*). The $\tau_{1/2}$ of recovery of the mobile fraction of Brl1(I395D)-mCherry is comparable to Brl1-mCherry. These data suggest that Brl(I395D)-mCherry accumulates in foci at the NE in which it is irreversibly trapped.

This observation motivated us to further characterize the foci at the NE, and we wanted to test whether these NE accumulations of Brl1(I395D)-mCherry also trap NPC components. To this end, we analyzed the co-localization of Brl1(I395D)-mCherry with several yEGFP-tagged Nups: Nup116, Nup133, and Nup170 display regular NE localization and importantly could be detected in the NE regions adjacent to the Brl1(I395D)-mCherry foci (*Figure 6D*). In contrast, Nup82 intensity at NE areas with Brl1(I395D)-mCherry puncta was strongly reduced. This labeling pattern is consistent with the one observed in the NPC herniations that form upon Brl1 depletion (*Figures 3 and 4*), suggesting that overexpressed Brl1(I395D) concentrates adjacent to NPC assembly intermediates composed of the IR and Y-complex but not the cytoplasmic NUPs.

To gain ultrastructural insights into the organization of the Brl1(I395D) accumulations, we investigated cells using cryo-ET on FIB-milled lamella (*Figure 7A–C*, *Figure 7—videos 1 and 2*). We observed mature NPCs, INM evaginations, and NE herniations as already seen in Brl1-depleted cells (*Figure 7Ai–iii*, *Figure 7B*, and *Figure 7—video 1*). No herniations could be observed in control cells (*Figure 7B*, *Figure 7—videos 3; 4*). To our surprise upon Brl1(I395D) overexpression, we also found large multilayered herniations with diameters up to ~600 nm, so far not reported in any other NPC assembly mutant (*Figure 7Aiv–vi*, *Figure 7—videos 1; 2*). These onion-like structures are composed of elongated INM herniations curling over each other with up to four stacked double bilayers. Of note, intermembrane distances were remarkably constant with two discrete widths of the innermost sheets, suggesting two different maturation modes for the onion-like herniations (see section 'Model for the development of "onion-like" herniations' in Appendix 1). Unlike the herniations in Brl1-depleted cells (*Figure 2A*), these structures were not filled with electron-dense material and only occasionally enclosed small patches of aggregate-like densities (*Figure 7Av–vi*, *Figure 7—videos 1; 2*). Single subtomograms and the subtomogram average of 47 herniations confirm the presence of an NPC intermediate with a diameter of 97 nm at the bases of these herniations (*Figure 7C*). Densities that likely correspond to the IR and the nucleoplasmic Y-complex ring but not the cytoplasmic side of the NPC can be distinguished. Although our average did not allow for unambiguous assignment or structure fitting, these densities look similar to the structures we observed in herniations of Brl1-depleted cells (*Figure 3—figure supplement 1Dii*) and the previously reported herniation structure in *nup116Δ* cells at 37°C (*Allegretti et al., 2020*), and are in a good agreement with the NUP localization patterns observed by fluorescence microscopy (*Figure 6D*).

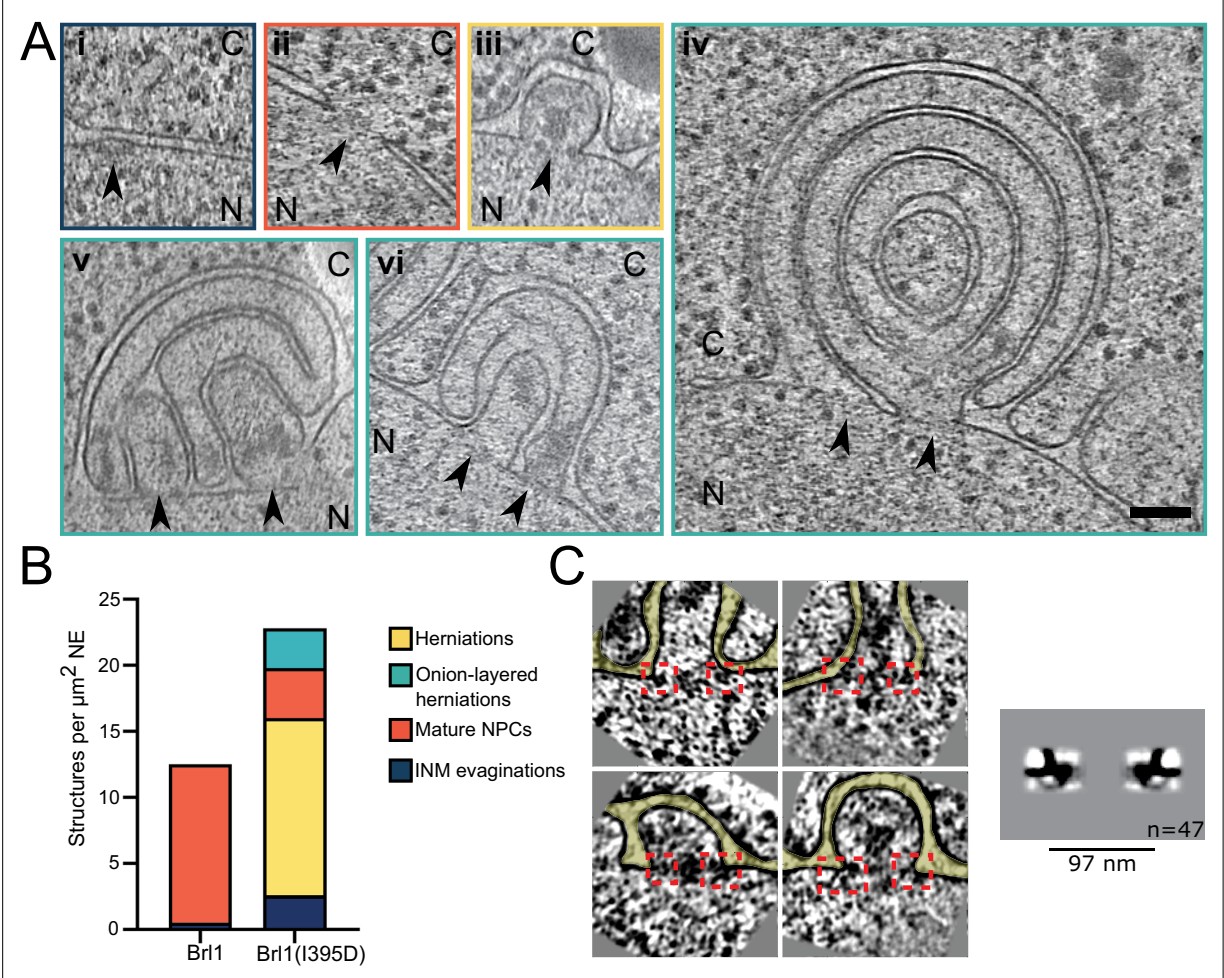

**Figure 7.** Brl1(I395D) overexpression leads to the formation of multilayered nuclear envelope (NE) herniations. (**A**) Tomographic slices of the nuclear pore complex (NPC)-like structures quantified in (**B**), observed in focused ion beam (FIB)-milled cells overexpressing Brl1(I395D), scale bar: 100 nm. N: nucleus, C: cytoplasm, slice thickness: 2.1 nm, arrows indicate NPC-like structures. Image frames colored according to the color code used in (**B**). Panels **iv** and **vi** are tomographic slices from the tomogram in *Figure 7—video 1*. (**B**) Quantification of observed structures in Brl1(I395D) cells and control condition; 17 (5.1 µm² NE) and 50 (9.8 µm² NE) tomograms were quantified for cells overexpressing Brl1 or Brl1(I395D), respectively. (**C**) Single subtomograms and the subtomogram average of 47 herniations in Brl1(I395D) overexpressing cells; box size of subtomograms is 270 nm; cytoplasm is at the top in each image. Red boxes indicate NPC-like densities at the neck of herniations.

The online version of this article includes the following video for figure 7:

**Figure 7—video 1.** Sequential sections of a cryo-tomogram from Brl1(I395D)-overexpressing cells.
https://elifesciences.org/articles/78385/figures#fig7video1

**Figure 7—video 2.** Sequential sections of a cryo-tomogram from Brl1(I395D)-overexpressing cells.
https://elifesciences.org/articles/78385/figures#fig7video2

**Figure 7—video 3.** Sequential sections of a cryo-tomogram from Brl1-overexpressing cells.
https://elifesciences.org/articles/78385/figures#fig7video3

**Figure 7—video 4.** Sequential sections of a cryo-tomogram from Brl1-overexpressing cells.
https://elifesciences.org/articles/78385/figures#fig7video4

Altogether, these results demonstrate the critical role of Brl1's AH during NPC maturation. The observation that the essential luminal ahBrl1 has a propensity to bind highly curved membranes and that Brl1 acts prior to INM-ONM fusion raises the possibility that Brl1 acts as a fusogen with membrane deforming properties. By deforming the INM, Brl1 could assist in the last NPC maturation step: the formation of a nucleo-cytoplasmic transport channel.

## Discussion

The NPC is one of the largest cellular protein complexes, yet only a few non-NPC proteins have been implicated in its biogenesis. One such factor is the integral membrane protein Brl1. However, at which stage Brl1 acts in the NPC assembly process or the mechanistic details have remained elusive. In this study, we show that Brl1 specifically associates with early NPC assembly intermediates, and we provide evidence for its role in membrane fusion.

Based on its binding capacity to structural NUPs, it was previously proposed that Brl1 associates with NPC maturation intermediates (*Zhang et al., 2018*). Using our recently developed KARMA method (*Onischenko et al., 2020*), we now demonstrate that Brl1 indeed preferentially interacts with newly synthesized NUPs and primarily co-localizes with newly produced NUP assemblies in live cells (*Figure 1D–F*). Furthermore, functional inactivation of Brl1 stalls NPC assembly without affecting previously assembled NPCs (*Figure 3E and F*). This leads to the accumulation of NE herniations that have a continuous ONM and contain incompletely assembled NPCs lacking the cytoplasmic export platform (*Figure 3*, *Figure 4*, and *Figure 3—figure supplement 1*). Thus, our results clearly identify Brl1 as an NPC assembly factor.

Depletion of Brl1 leads to the formation of incomplete NPC structures that contain the IR, MR, Y-complex, and NB NUPs. The cytoplasmic Nup159 and Nup82 are absent from the intermediates but are instead mislocalized in cytoplasmic foci, as previously seen in other NPC assembly mutants (*Hodge et al., 2010*; *Scarcelli et al., 2007*; *Makio et al., 2009*; *Onischenko et al., 2009*; *Onischenko et al., 2017*; *Figure 3C and D*). In light of the observed NE herniations in Brl1-depleted cells (*Figure 3A and B*, *Figure 3—figure supplement 1E*), the fusion of the INM and ONM appears to be a prerequisite for the recruitment of the cytoplasmic Nup159-Nup82-Nsp1 complex. Thus, our data support an inside-out mode of interphase NPC assembly, similar to previous observations in yeast and mammalian cells (*Wente and Blobel, 1993*; *Murphy et al., 1996*; *Zabel et al., 1996*; *Otsuka et al., 2016*; *Onischenko et al., 2020*; *Makio et al., 2009*; *Marelli et al., 2001*). Interestingly, in Brl1-depleted cells the Y-complex NUPs display a reduced NE fluorescence signal and slow fractional labeling in our proteomic assays (*Figures 3C and D and 4C and D*). This suggests that only the nucleoplasmic Y-complex ring is present in the intermediates. This is also in line with our cryo-ET data (*Figure 3—figure supplement 1E*) and with previous results in *nup116Δ* cells (*Allegretti et al., 2020*), suggesting that INM-ONM fusion is needed before the cytoplasmic Y-ring can be recruited to the assembling NPC.

Curiously, intermixing rates of the NUP complexes when affinity isolated with early recruited NUP Nup170 significantly increased upon Brl1 depletion and were overall much higher than previously reported for APs with Mlp1 that preferentially associates with otherwise matured NPC structures (*Figure 4—figure supplement 1*; *Onischenko et al., 2020*). This may indicate that NPC assembly intermediates are inherently unstable prior to pore membrane fusion.

Interestingly, Mlp1 and other NB components such as NUP1, which join very late during the normal course of NPC assembly (*Onischenko et al., 2020*), were still readily recruited to the NE upon Brl1 depletion (*Figure 3C–F*; *Onischenko et al., 2020*). This indicates that the NB constituents including Mlp1 might assemble in a kinetically slow process that does not strictly depend on membrane fusion. Consistent with this, complexes between NB NUPs Nup60, Nup2, and Mlp1 could be reconstituted in the absence of other NUPs in vitro (*Cibulka et al., 2022*).

The fusion of INM and ONM is a crucial step during de novo NPC assembly in interphase. Membrane fusion does not occur spontaneously, and based on previously characterized membrane fusion events, it is likely that two NE lipid bilayers must be brought into proximity to initiate the fusion of the membranes (*Peeters et al., 2022*). While the fusion event itself is expected to be fast and thus difficult to investigate, potential assembly intermediate states in which INM and ONM approach each other but are not yet fused can be observed in cells with NPC assembly defects (*Thaller and Patrick Lusk, 2018*; *Makio et al., 2009*) and rarely also in normal cells (*Otsuka et al., 2016*) and our cryo-ET data (*Figure 3—video 3*, *Figures 3B and 7B*). It has been suggested that NUPs and other proteins containing amphipathic helices are important players in the formation and stabilization of these early NPC-intermediates since they can bind to and deform membranes (*Schooley et al., 2012*; *Dawson et al., 2009*; *Cibulka et al., 2022*; *Voeltz et al., 2006*; *Wang et al., 2021*). In this study, we identified a membrane-binding AH within the luminal domain of Brl1 that is essential for its function in NPC assembly as genetic perturbations that abolish membrane binding lead to severely impaired NPC biogenesis. Interestingly, this AH preferentially binds to highly curved lipid membranes, is highly

conserved in organisms with closed mitosis, and is a shared feature of proteins associated with NPC assembly such as Brr6, Apq12, and ER-bending reticulons (*Dawson et al., 2009*; *Zhang et al., 2021*). Taken together, these results emphasize the emerging role of AH motifs in NPC assembly.

Brr6 is a paralogue of Brl1 with the same topology and orientation in the NE. Interestingly, Brr6 also contains a predicted luminal AH, indicating that both proteins might function similarly. Furthermore, it has been shown that Brr6 co-localizes at Brl1 foci at the NE and physically interacts with Brl1 (*Lone et al., 2015*; *Saitoh et al., 2005*; *Zhang et al., 2018*). Although thermosensitive *brr6* and *brl1* mutants can be rescued by *BRL1* and *BRR6* expression, respectively (*Saitoh et al., 2005*), deletion of BRL1 or BRR6 cannot be rescued by overexpression of the respective paralogue. Furthermore, several NPC assembly mutants such as *gle2Δ*, *nup116Δ*, and *nup116ΔGLFG PMET3-NUP188* can only be rescued by Brl1 overexpression. This demonstrates that despite similar sequence (44% sequence similarity of the structured parts) and structure, Brl1 and Brr6 do not act redundantly in NPC assembly. This is also in agreement with the differential localization of these two proteins: Brl1 mainly localizes to the INM, whereas Brr6 can be found in both NE leaflets (*Zhang et al., 2018*). Thus, it seems likely that Brl1 and

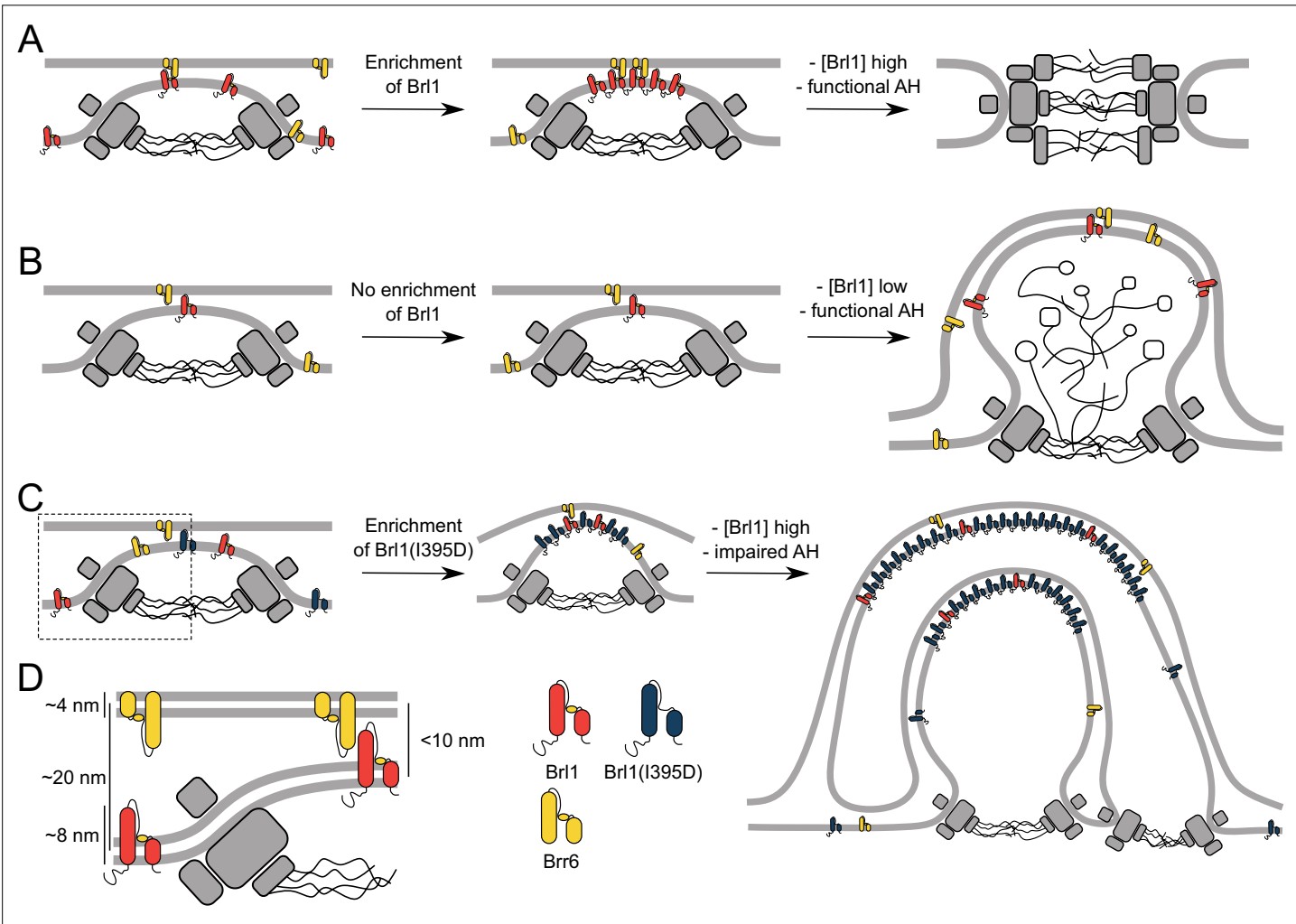

**Figure 8.** The role of Brl1 during nuclear pore complex (NPC) assembly. (**A**) Brl1 (red) enriches on the inside of NPC maturation intermediates and promotes inner nuclear membrane–outer nuclear membrane (INM-ONM) fusion through the membrane-binding amphipathic helix (AH) motif and likely in cooperation with Brr6 (yellow). (**B**) If Brl1 cannot reach the critical concentration required to promote membrane fusion, unresolved nuclear envelope (NE) herniations, filled with electron-dense material, appear. (**C**) Overexpressed Brl1(I395D) with a perturbed AH (blue) concentrates at the NPC assembly site. It remodels the NE membranes and leads to expanded multilayered herniations but ultimately fails to induce membrane fusion. (**D**) Brl1 at the INM can only physically interact with Brr6 or Brl1 at the ONM when the NE leaflets approach as it is the case at NPC assembly sites. Dimensions based on our cryo-electron tomography (cryo-ET) data (*Figure 3—figure supplement 1D*), structure prediction (*Figure 5—figure supplement 1*), and measurements of the NE (*Appendix 1—figure 2B*).

Brr6 act in concert during NPC assembly and membrane fusion; however, the detailed function of Brr6 and the role of additional NUPs and potential assembly factors like Apq12 remains unclear and awaits further characterization.

How does Brl1 promote interphase NPC assembly? Our observations that NPC assembly intermediates that form in the absence of Brl1 already contain membrane-binding NUPs (*Figure 4D*) suggest that they play a key role in deforming the INM, leading to INM evaginations (*Figure 8A–C*, left). We propose that Brl1 is recruited to and concentrated at these NPC assembly sites (*Figure 8A*). This view is supported by the punctate localization pattern of endogenously tagged Brl1-yEGFP (*Lone et al., 2015*), co-localization of Brl1-puncta with newly synthesized NUPs (*Figure 1E*), and accumulation of dysfunctional Brl1 mutants at stalled NPC assembly sites (*Figure 6B and D*, *Figure 7*). The mechanisms by which Brl1 is recruited and concentrated at assembly sites are not clear but the unstructured N-terminus of Brl1 might contribute. This is supported by the non-punctate localization of Brr6 that contains only a short N-terminus (*Lone et al., 2015*). A localization preference of Brl1 to highly curved membranes of INM evaginations could be an alternative explanation, which is supported by the binding preference of ahBrl1 to highly curved lipid membranes (*Figure 5F*). Irrespectively, it seems likely that a high local concentration of Brl1 is critical for membrane fusion as overexpression of Brl1 can rescue assembly defects in multiple NUP mutants (*Zhang et al., 2018*; *Liu et al., 2015*).

Our results show that ahBrl1 is required for the INM-ONM fusion event since cells that express Brl1 with an impaired AH are not viable and overexpression of Brl1(I395D) inhibits NPC biogenesis, leading to the formation of incomplete NPC assemblies with multilayered INM herniations (*Figure 7A*). Brl1(I395D) accumulates irreversibly in these structures as shown by the high concentration and slow mobility in herniations (*Figure 6B and C*). Interestingly, the surface of the highly curved liposomes to which ahBrl1 preferentially binds topologically resembles the luminal side of INM evaginations. Both structures have a similar high positive curvature (*Figure 3—figure supplement 1D*, *Figure 5F*, and model in *Figure 8A*). This raises the possibility that Brl1 can bind, and potentially induce and/or stabilize highly curved, energetically unfavorable pre-fusion intermediates.

Since overexpressed Brl1(I395D) strongly accumulates at NPC assembly sites and induces the formation of highly curved onion-like membrane sheets, we speculate that in the absence of a functional AH Brl1 can still mediate membrane remodeling but not INM and ONM fusion. This points to an important role of ahBrl1 in the membrane fusion event (*Figure 8C*). But how could Brl1 and its AH mediate the fusion of the INM and ONM? Interestingly, the AlphaFold structure of Brl1 not only predicted an AH but also revealed the presence of a luminal, ~8-nm-long continuous alpha-helix (*Figure 5A*) that is stabilized by disulfide bridges (*Figure 5—figure supplement 1D*; *Zhang et al., 2018*). Whereas this helix is too short to span the entire ~20 nm of the NE lumen (*Appendix 1—figure 2B*), it is conceivable that at INM herniations, where the two leaflets approach each other, this helix could interact with proteins in the ONM (*Figure 8D*). Intriguingly, a helix of similar length is also predicted in Brr6 (*Figure 5—figure supplement 2*). It is tempting to speculate that Brl1 at the INM interacts with Brr6 at the ONM at early NPC assembly sites and that this interaction leads to INM-ONM fusion mediated by the conserved AHs present in both proteins. This possibility is supported by the physical interaction of these proteins (*Lone et al., 2015*; *Zhang et al., 2018*), functional requirement of both paralogues for NPC biogenesis (*Zhang et al., 2018*), and differential localization patterns of Brl1 and Brr6 in immunogold labeling assays wherein Brl1 predominantly localizes at the INM while Brr6 is equally distributed between INM and ONM (*Zhang et al., 2018*).

Aside from the direct role in membrane fusion, Brl1 might also affect the lipid composition of the NE. Indeed, it has been proposed that Brl1 forms a sensory complex with Brr6 and Apq12 that controls membrane fluidity (*Lone et al., 2015*). During NE fusion and other NPC assembly steps, the membrane curvature of the NE is extensively modulated and changes in lipid composition, either globally or locally at NPC assembly sites, could facilitate this process. In fact, in Apq12-overexpressing cells, phosphatidic acid (PA) accumulates at sites of ONM overproliferation (*Zhang et al., 2021*; *Romanauska and Köhler, 2018*). A similar PA accumulation was reported at *nup116Δ* herniations, indicating that PA might be a relevant effector during NPC assembly (*Thaller et al., 2021*). However, the effects of Brl1, Brr6, and Apq12 on lipid composition are somewhat controversial (*Lone et al., 2015*; *Scarcelli et al., 2007*; *Zhang et al., 2018*), requiring better tools to understand the role of lipid environment in NPC biogenesis. Of note, membrane proliferation or remodeling can also be induced by an overexpression of membrane proteins without necessarily altering the overall lipid composition.

For example, overexpression of transmembrane proteins induces the formation of karmellae (*Wright et al., 1988*), expansions of the NE/ER membranes. Similarly, overexpression of AH-containing NUPs was shown to induce NE overproliferation, resulting in multiple, stacked membrane cisternae (*Marelli et al., 2001*; *Mészáros et al., 2015*) that were also observed upon overexpression of Brl1 or Brr6 (*Zhang et al., 2018*). Therefore, it is likely that NE overproliferation also plays a role in the generation of the onion-like herniations that we observe in cells overexpressing the dominant-negative Brl1 variant, Brl(I395D). In the future, it will be important to manipulate NE lipids and characterize the effects of membrane composition in NPC assembly.

# Materials and methods

## Key resources table

| Reagent type (species) or resource | Designation | Source or reference | Identifiers | Additional information |
|---|---|---|---|---|
| Chemical compound, drug | IP6 | Sigma-Aldrich | P5681 | AID experiments |
| Chemical compound, drug | Auxin | Sigma-Aldrich | I2886 | AID experiments |
| Chemical compound, drug | β-Estradiol | Sigma-Aldrich | E8875 | RITE experiments |
| Chemical compound, drug | Hygromycin B | Roche | 10843555001 | RITE experiments |
| Chemical compound, drug | Concanavalin A (ConA) | Sigma-Aldrich | C2010 | Fluorescence microscopy |
| Chemical compound, drug | IPTG | AppliChem | A10080025 | Protein expression |
| Chemical compound, drug | Protease inhibitor cocktail | Sigma-Aldrich | P8215 | MS assays |
| Chemical compound, drug | Purified IgG protein from rabbit serum | Sigma-Aldrich | I5006 | MS assays |
| Chemical compound, drug | Coomassie Brilliant Blue R-250 | Bio-Rad | 161-0400 | MS assays |
| Chemical compound, drug | Sequencing grade porcine trypsin | Promega | V5113 | MS assays |
| Chemical compound, drug | L-Lysine:2HCL ($^{13}C_6$, 99%; $1^5N_2$, 99%) | Cambridge Isotope Laboratories | CNLM-291-H-0.25 | MS assays |
| Chemical compound, drug | Iodoacetamide | Sigma-Aldrich | I1149 | MS assays |
| Chemical compound, drug | Ammonium bicarbonate | Sigma-Aldrich | 09832 | MS assays |
| Chemical compound, drug | Formic acid 99–100% | VWR Chemicals | 20318.297 | MS assays |
| Chemical compound, drug | iRT Kit | Biognosis | Ki-3002-1 | MS assays |
| Chemical compound, drug | BioPureSPN MINI Columns Silica C18 | The Nest Group, Inc | HUM S18V | MS assays |
| Chemical compound, drug | BioPureSPN MACRO Columns Silica C18 | The Nest Group, Inc | HMM S18V | MS assays |
| Peptide, recombinant protein | DNase I | Roche | 10104159001 | Protein purification |
| Chemical compound, drug | cOmplete Protease Inhibitor Cocktail | Roche | 05053489001 | Protein purification |
| Other | Ni-NTA Agarose | QIAGEN | 30210 | Protein purification |
| Antibody | α-V5 (mouse monoclonal) | Invitrogen | R960-25 | Western blotting, 1:2000 |
| Antibody | α-Hexokinase (rabbit monoclonal) | US Biologicals | H2035-01 | Western blotting, 1:3000 |
| Antibody | α-Mouse IgG Alexa Fluor 680 (goat polyclonal) | Thermo Fisher Scientific | A-21057 | Western blotting, 1:10,000 |
| Antibody | α-Rabbit IgG IRDye800CW (goat polyclonal) | Li-COR Biosciences | 926-32211 | Western blotting, 1:10,000 |
| Antibody | α-EGFP (mouse monoclonal) | Roche | 11814460001 | Liposome experiments, 1:2000 |
| Antibody | α-Mouse IgG peroxidase conjugate (goat polyclonal) | Calbiochem | 401215 | Liposome experiments, 1:5000 |
| Other | Zymolyase 100T | ICN | 320932 | Tetrad dissection |
| Other | *E. coli* polar lipids | Avanti Polar Lipids | 100600C | Liposome experiments |

*Continued on next page*

*Continued*

| Reagent type (species) or resource | Designation | Source or reference | Identifiers | Additional information |
|---|---|---|---|---|
| Other | 18:1 Liss Rhodamine PE | Avanti Polar Lipids | 810150C | Liposome experiments |
| Software, algorithm | NuRim | *Rajoo et al., 2018*; *Vallotton et al., 2019* | | Data analysis |
| Software, algorithm | ImageJ/Fiji | *Schindelin et al., 2012* | | Data acquisition |
| Software, algorithm | NIS Elements | Nikon | | Data acquisition |
| Software, algorithm | MATLAB | MathWorks | | Data analysis |
| Software, algorithm | IMOD | https://bio3d.colorado.edu/imod/ | | Data analysis |
| Software, algorithm | SerialEM | *Mastronarde, 2003* | | Data acquisition |
| Software, algorithm | UCSF Chimera | *Pettersen et al., 2004* | | Data visualization |
| Software, algorithm | SBGrid | *Morin et al., 2013* | | Data processing |
| Software, algorithm | TOM toolbox | *Nickell et al., 2005*; *Förster et al., 2005*, https://www.biochem.mpg.de/6348566/tom_e | | Data processing |
| Software, algorithm | AlphaFold2.1.1 | *Jumper et al., 2021* | | Data analysis |
| Software, algorithm | AlphaFold visualization | https://colab.research.google.com/drive/1CizC7zmYvFkav5qfBbWxhgUHrOxwym2w | | Data analysis |
| Software, algorithm | COBALT | https://www.ncbi.nlm.nih.gov/tools/cobalt/re_cobalt.cgi | | Sequence alignment |
| Software, algorithm | Jalview | *Waterhouse et al., 2009* | | Sequence alignment |
| Software, algorithm | Excel | Microsoft | | Data analysis |
| Software, algorithm | R v. 4.1.2 | R Project | | Data analysis |
| Software, algorithm | Prism 7 Prism 9 | GraphPad | | Visualization |
| Software, algorithm | Inkscape 1.1 | https://inkscape.org/ | | Visualization |
| Software, algorithm | Illustrator v. 26.0.3 | Adobe | | Visualization |

## Plasmids and yeast strains construction

Plasmids were generated according to standard molecular cloning techniques. The plasmids used in this study are listed in *Table 1*. Standard yeast genetic protocols were used for plasmid transformation and integration of linear DNA fragments into the yeast genome by homologous recombination. Strains used in this study are listed in *Table 2*. The heterozygous yeast strains *BRL1/brl1Δah* (lacking amino acids 376–402) and *BRL1/brl1(I395D)* were generated with CRISPR-Cas9 genome editing. Cloning details are available on request.

## Yeast culturing conditions

Unless otherwise stated, yeast cultures were grown to mid-log phase for at least 12 hr at 30°C. For Western blot analysis and fitness assays, cells were cultured in YPD medium (1% yeast extract, 2% peptone, 2% dextrose) and for microscopy and proteomic analyses in synthetic complete medium (SCD, 6.7 g/L yeast nitrogen base without amino acids, 2% dextrose) supplemented with the necessary amino acids and nucleobases. Auxin-inducible degradation of Brl1 in log-phase yeast cultures with $OD_{600}$ = 0.1–0.2 was induced by addition of IP6 (4 µM phytic acid dipotassium salt, Sigma-Aldrich, P5681) and either auxin (+auxin, 500 µM indole-3-acetic acid in ethanol, Sigma-Aldrich, I2886) or the equivalent amount of ethanol (-auxin) for the solvent control. Strains with galactose-inducible Brl1 constructs were pre-cultured in SC medium containing 2% raffinose. Expression was induced by supplementing 2% galactose to log-phase cultures $OD_{600}$ = 0.1–0.2. For the metabolic labeling experiments, cells were initially grown in SCD containing light lysine (light SCD, 25 mg/L) and then pulse labeled by medium exchange to SCD containing $^{13}C_6$, $^{15}N_2$ l-lysine (heavy SCD, Cambridge Isotope Laboratories, 25 mg/L).

## Western blotting

Auxin-inducible degradation was performed as described above (yeast culturing conditions). At each post-degradation time point, an amount of cells corresponding to 2 $OD_{600}$ was collected by

**Table 1.** Plasmids used in this study.

| ID | Description | Details | Source |
|---|---|---|---|
| pKW4689 | pRS306-pGAL1-yEGFP | Integration in ura3 locus, expression of yEGFP | This study |
| pKW4558 | pRS306-pGAL1-ahBrl1-yEGFP | Integration in ura3 locus, expression of ahBrl1-yEGFP | This study |
| pKW4578 | pRS306-pGAL1-MRATSK | Integration in ura3 locus, overexpression of small peptide MRATSK | This study |
| pKW4568 | pRS306-pGAL1-BRL1 | Integration in ura3 locus, overexpression of Brl1 | This study |
| pKW4649 | pRS306-pGAL1-BRL1-yEGFP | Integration in ura3 locus, overexpression of Brl1-yEGFP | This study |
| pKW4651 | pRS306-pGAL1-BRL1Δah-yEGFP | Integration in ura3 locus, overexpression of BRL1Δah-yEGFP | This study |
| pKW4712 | pRS306-pGAL1-BRL1(I395D)-yEGFP | Integration in ura3 locus, overexpression of Brl1(I395D)-yEGFP | This study |
| pKW589 | pRS305 | Integration in leu2 locus | *Sikorski and Hieter, 1989* |
| pKW4915 | pRS305-pGAL1-BRL1-mCherry | Integration in leu2 locus, overexpression of Brl1-mCherry | This study |
| pKW4919 | pRS305-pGAL1-BRL1(I395D)-mCherry | Integration in leu2 locus, overexpression of Brl1(I395D)-mCherry | This study |
| pKW1358 | YIplac204-pTPI1-dsRED-HDEL | Integration in trp1 locus, ER marker | *Bevis and Glick, 2002* |
| pKW1964 | pFA6A-link-yEGFP-CaURA3 | PCR template for endogenous tagging with yEGFP | *Sheff and Thorn, 2004* |
| pKW3359 | pFA6a-NH605-3V5-IAA7-KanMX6 | PCR template for endogenous tagging with V5-IAA7 | *Derrer et al., 2019* |
| pKW2874 | pNH603-pGPD1-osTIR1-HIS3MX | Integration in his3 locus, OsTIR1 expression | *Derrer et al., 2019* |
| pKW2830 | pNH603-pGPD1-osTIR1-LEU2 | Integration in leu2 locus, OsTIR1 expression | *Chan et al., 2018* |
| pKW354 | pRS426 | 2μ–URA3 plasmid | *Christianson et al., 1992* |
| pKW4468 | pRS426-BRL1 | 2μ–URA3 plasmid containing BRL1 locus | This study |
| pKW4659 | pSV272-6xHis-MBP-TEV-yEGFP | Bacterial expression of 6xHis-MBP-TEV-yEGFP | This study |
| pKW4660 | pSV272-6xHis-MBP-TEV-ahBrl1-yEGFP | Bacterial expression of 6xHis-MBP-TEV-ahBrl1-yEGFP | This study |
| pKW4683 | pSV272-6xHis-MBP-TEV-ahBrl1(F391D)-yEGFP | Bacterial expression of 6xHis-MBP-TEV-ahBrl1(F391D)-yEGFP | This study |
| pKW4684 | pSV272-6xHis-MBP-TEV-ahBrl1(I395D)-yEGFP | Bacterial expression of 6xHis-MBP-TEV-ahBrl1(I395D)-yEGFP | This study |
| pKW4612 | pRS306-pGAL1-BRL1(F391D) | Integration in ura3 locus, overexpression of Brl1(F391D) | This study |
| pKW4613 | pRS306-pGAL1-BRL1(I395D) | Integration in ura3 locus, overexpression of Brl1(I395D) | This study |
| pKW4614 | pRS306-pGAL1-BRL1(F398D) | Integration in ura3 locus, overexpression of Brl1(F398D) | This study |
| pKW4615 | pRS306-pGAL1-BRL1(L402D) | Integration in ura3 locus, overexpression of Brl1(L402D) | This study |
| pKW4616 | pRS306-pGAL1-BRL1(D393A) | Integration in ura3 locus, overexpression of Brl1(D393A) | This study |
| pKW4617 | pRS306-pGAL1-BRL1(D400A) | Integration in ura3 locus, overexpression of Brl1(D400A) | This study |
| pKW3061 | pRS303-GPD-CRE-EBD78 | Integration in his3 locus CRE-EBD78 expression | *Terweij et al., 2013* |
| pKW804 | pFA6-S-TEV-ZZ-KanMX | PCR template for endogenous tagging with S-TEV-ZZ | *Onischenko et al., 2020* |
| pKW953 | pFA6-S-TEV-ZZ-URA3 | PCR template for endogenous tagging with S-TEV-ZZ | This study |
| pKW4746 | pFa6a-loxP-HA-mCherry-HygMX-loxP-GFP | PCR template for endogenous tagging with loxP-HA-mCherry-HygMX-loxP-GFP | *Colombi et al., 2013* |
| pKW2407 | pFA6a-mCherry-NAT | PCR template for endogenous tagging with mCherry | *Onischenko et al., 2020* |
| pKW4945 | pFA6a-loxP-yEGFP_URA3-loxP | PCR template for endogenous tagging with loxP-yEGFP_URA3-loxP | This study |
| pKW4946 | pFA6a-loxP-URA3-loxP_yEGFP | PCR template for endogenous tagging with loxP-URA3-loxP_yEGFP | This study |

centrifugation and lysed by a 15 min incubation in 0.1 M sodium hydroxide. Subsequently, cells were pelleted, resuspended in 50 µL Laemmli sample buffer (10% glycerol, 2% SDS, 5% 2-mercaptoethanol, 100 mM DTT, 0.04% bromophenol blue, 62.5 mM Tris–HCl pH 6.8) and heat denatured for 5 min at 95°C. Proteins were electrophoretically separated on an 8% polyacrylamide gel and then wet-transferred to a nitrocellulose membrane (Amersham Protran 0.2 NC, GE Healthcare). Prior to antibody incubation, membranes were blocked for at least 2 hr in 5% PBST-milk (1× PBS pH 7.4, 0.1% Tween-20, 5% dry milk). Then, membranes were incubated with primary antibody for 1 hr at room

**Table 2.** Yeast strains used in this study.

| ID | Genotype | Source |
|---|---|---|
| KWY165 | haploid, MATa leu2-3,112 trp1-1 can1-100 ura3-1 ade2-1 his3-11,15 [phi+] | W303 haploid |
| KWY166 | diploid, leu2-3,112 trp1-1 can1-100 ura3-1 ade2-1 his3-11,15 [phi+] | W303 diploid |
| KWY1602 | MAT(alpha) his3-1 leu2-0 lys2-0 ura3-0 | BY4742 |
| KWY9200 | KWY165, BRL1::BRL1-V5-IAA7-KanMX trp1::pADH1-dsRED-HDEL-TRP1 | This study |
| KWY9204 | KWY165, BRL1::BRL1-V5-IAA7-KanMX his3::pGPD1-OsTIR1 trp1::pADH1-dsRED-HDEL-TRP1 | This study |
| KWY9268 | KWY165, BRL1::BRL1-V5-IAA7-KanMX trp1::pADH1-dsRED-HDEL-TRP1 NUP82::NUP82-yEGFP-URA3 | This study |
| KWY9269 | BRL1::BRL1-V5-IAA7-KanMX his3::pGPD1-OsTIR1 trp1::pADH1-dsRED-HDEL-TRP1 NUP82::NUP82-yEGFP-URA3 | This study |
| KWY9270 | KWY165, BRL1::BRL1-V5-IAA7-KanMX trp1::pADH1-dsRED-HDEL-TRP1 NUP116::NUP116-yEGFP-URA3 | This study |
| KWY9271 | KWY165, BRL1::BRL1-V5-IAA7-KanMX his3::pGPD1-OsTIR1 trp1::pADH1-dsRED-HDEL-TRP1 NUP116::NUP116-yEGFP-URA3 | This study |
| KWY9272 | KWY165, BRL1::BRL1-V5-IAA7-KanMX trp1::pADH1-dsRED-HDEL-TRP1 NUP100::NUP100-yEGFP-URA3 | This study |
| KWY9273 | KWY165, BRL1::BRL1-V5-IAA7-KanMX his3::pGPD1-OsTIR1 trp1::pADH1-dsRED-HDEL-TRP1 NUP100::NUP100-yEGFP-URA3 | This study |
| KWY9274 | KWY165, BRL1::BRL1-V5-IAA7-KanMX trp1::pADH1-dsRED-HDEL-TRP1 NUP133::NUP133-yEGFP-URA3 | This study |
| KWY9275 | KWY165, BRL1::BRL1-V5-IAA7-KanMX his3::pGPD1-OsTIR1 trp1::pADH1-dsRED-HDEL-TRP1 NUP133::NUP133-yEGFP-URA3 | This study |
| KWY9276 | KWY165, BRL1::BRL1-V5-IAA7-KanMX trp1::pADH1-dsRED-HDEL-TRP1 NUP170::NUP170-yEGFP-URA3 | This study |
| KWY9277 | KWY165, BRL1::BRL1-V5-IAA7-KanMX his3::pGPD1-OsTIR1 trp1::pADH1-dsRED-HDEL-TRP1 NUP170::NUP170-yEGFP-URA3 | This study |
| KWY9908 | KWY165, BRL1::BRL1-V5-IAA7 KanMX trp1::ADH1-dsRED-HDEL-TRP1 NUP60::NUP60-yEGFP-URA3 | This study |
| KWY10108 | KWY165, BRL1::BRL1-V5-IAA7 KanMX his3::pGPD1-OsTIR1 trp1::pADH1-dsRED-HDEL-TRP1 NUP60::NUP60-yEGFP-URA3 | This study |
| KWY9909 | KWY165, BRL1::BRL1-V5-IAA7 KanMX trp1::pADH1-dsRED-HDEL-TRP1 NUP159::NUP159-yEGFP-URA3 | This study |
| KWY10109 | KWY165, BRL1::BRL1-V5-IAA7 KanMX his3::pGPD1-OsTIR1 trp1::pADH1-dsRED-HDEL-TRP1 NUP159::NUP159-yEGFP-URA3 | This study |
| KWY9911 | KWY165, BRL1::BRL1-V5-IAA7 KanMX trp1::pADH1-dsRED-HDEL-TRP1 MLP1::MLP1-yEGFP-URA3 | This study |
| KWY10110 | KWY165, BRL1::BRL1-V5-IAA7 KanMX his3::pGPD1-OsTIR1 trp1::pADH1-dsRED-HDEL-TRP1 MLP1::MLP1-yEGFP-URA3 | This study |
| KWY9939 | KWY165, BRL1::BRL1-V5-IAA7 KanMX trp1::pADH1-dsRED-HDEL-TRP1 NUP85::NUP85-yEGFP-URA3 | This study |
| KWY10107 | KWY165, BRL1::BRL1-V5-IAA7 KanMX his3::pGPD1-OsTIR1 trp1::pADH1-dsRED-HDEL-TRP1 NUP85::NUP85-yEGFP-URA3 | This study |
| KWY10029 | KWY165, BRL1::BRL1-V5-IAA7 KanMX trp1::pADH1-dsRED-HDEL-TRP1 NUP192::NUP192-yEGFP-URA3 | This study |
| KWY10111 | KWY165, BRL1::BRL1-V5-IAA7 KanMX his3::pGPD1-OsTIR1 trp1::pADH1-dsRED-HDEL-TRP1 NUP192::NUP192-yEGFP-URA3 | This study |
| KWY5540 | KWY165, nup116ΔGLFG NUP188::HIS3MX-pMet3-3xHA-Nup188 | *Onischenko et al., 2017* |
| KWY8891 | KWY165, ura3::pGAL1-BRL1-yeGFP-CaURA3 | This study |
| KWY8893 | KWY165, ura3::pGAL1-BRL1Δah-yeGFP-CaURA3 | This study |
| KWY9070 | KWY165, ura3::pGAL1-BRL1(I395D)-yeGFP-CaURA3 | This study |
| KWY8894 | KWY165, ura3::pGAL1-MRATSK-CaURA3 | This study |
| KWY8895 | KWY165, ura3::pGAL1-BRL1-CaURA3 | This study |
| KWY8896 | KWY165, ura3::pGAL1-BRL1Δah-CaURA3 | This study |
| KWY8898 | KWY165, ura3::pGAL1-BRL1(I395D)-CaURA3 | This study |
| KWY10154 | KWY165, NUP82::NUP82-yeGFP-CaURA3 leu2::pRS305-LEU2 | This study |
| KWY10155 | KWY165, NUP82::NUP82-yeGFP-CaURA3 leu2::pGAL1-BRL1-mCherry-LEU2 | This study |
| KWY10159 | KWY165, NUP82::NUP82-yeGFP-CaURA3 leu2::pGAL1-BRL1(I395D)-mCherry-LEU2 | This study |
| KWY10161 | KWY165, NUP116::NUP116-yeGFP-CaURA3 leu2::pRS305-LEU2 | This study |
| KWY10162 | KWY165, NUP116::NUP116-yeGFP-CaURA3 leu2::pGAL1-BRL1-mCherry-LEU2 | This study |
| KWY10166 | KWY165, NUP116::NUP116-yeGFP-CaURA3 leu2::pGAL1-BRL1(I395D)-mCherry-LEU2 | This study |
| KWY10168 | KWY165, NUP133::NUP133-yeGFP-CaURA3 leu2::pRS305-LEU2 | This study |
| KWY10169 | KWY165, NUP133::NUP133-yeGFP-CaURA3 leu2::pGAL1-BRL1-mCherry-LEU2 | This study |

*Table 2 continued on next page*

*Table 2 continued*

| ID | Genotype | Source |
|---|---|---|
| KWY10173 | KWY165, NUP133::NUP133-yeGFP-CaURA3 leu2::pGAL1-BRL1(I395D)-mCherry-LEU2 | This study |
| KWY10175 | KWY165, NUP170::NUP170-yeGFP-CaURA3 leu2::pRS305-LEU2 | This study |
| KWY10176 | KWY165, NUP170::NUP170-yeGFP-CaURA3 leu2::pGAL1-BRL1-mCherry-LEU2 | This study |
| KWY10180 | KWY165, NUP170::NUP170-yeGFP-CaURA3 leu2::pGAL1-BRL1(I395D)-mCherry-LEU2 | This study |
| KWY10260 | KWY165, SEC61::SEC61-yEGFP-CaURA3 leu2-3::pRS305-LEU2 | This study |
| KWY10261 | KWY165, SEC61::SEC61-yEGFP-CaURA3 leu2-3::BRL1-mCherry-LEU2 | This study |
| KWY10262 | KWY165, SEC61::SEC61-yEGFP-CaURA3 leu2-3::BRL1(I395D)-mCherry-LEU2 | This study |
| KWY8876 | KWY165, nup116ΔGLFG NUP188::HIS3MX-pMet3-3xHA-Nup188 ura3::pGAL1-BRL1Δah-URA3 | This study |
| KWY8874 | KWY165, nup116ΔGLFG NUP188::HIS3MX-pMet3-3xHA-Nup188 ura3::pGAL1-BRL1-URA3 | This study |
| KWY8877 | KWY165, nup116ΔGLFG NUP188::HIS3MX-pMet3-3xHA-Nup188 ura3::pGAL1-MRATSK-URA3 | This study |
| KWY8882 | KWY165, nup116ΔGLFG NUP188::HIS3MX-pMet3-3xHA-Nup188 ura3::pGAL1-BRL1(F391D)-URA3 | This study |
| KWY8883 | KWY165, nup116ΔGLFG NUP188::HIS3MX-pMet3-3xHA-Nup188 ura3::pGAL1-BRL1(I395D)-URA3 | This study |
| KWY8884 | KWY165, nup116ΔGLFG NUP188::HIS3MX-pMet3-3xHA-Nup188 ura3::pGAL1-BRL1(F398D)-URA3 | This study |
| KWY8885 | KWY165, nup116ΔGLFG NUP188::HIS3MX-pMet3-3xHA-Nup188 ura3::pGAL1-BRL1(L402D)-URA3 | This study |
| KWY8886 | KWY165, nup116ΔGLFG NUP188::HIS3MX-pMet3-3xHA-Nup188 ura3::pGAL1-BRL1(D393A)-URA3 | This study |
| KWY8887 | KWY165, nup116ΔGLFG NUP188::HIS3MX-pMet3-3xHA-Nup188 ura3::pGAL1-BRL1(D400A)-URA3 | This study |
| KWY9079 | KWY165, trp1::dsRed-HDEL-TRP1 ura3::pGAL1-ahBRL1-yEGFP-CaURA3 | This study |
| KWY9075 | KWY165, trp1::dsRed-HDEL-TRP1 ura3::pGAL1-yEGFP-CaURA3 | This study |
| KWY10418 | KWY165, Brl1-mCherry::Nat his3::CRE-EBD78 NUP170::NUP170-loxP-GFP_URA3-loxP_STOP | This study |
| KWY10419 | KWY165, Brl1-mCherry::Nat his3::CRE-EBD78 NUP170::NUP170-loxP-STOP_URA3-loxP_GFP | This study |
| KWY9964 | KWY1602, BRL1::BRL1-S-TEV-ZZ-KanMX | This study |
| KWY10453 | KWY1602, BRL1::BRL1-V5-IAA7-KanMX his3::pGPD1-OsTIR1 NUP170::NUP170-S-TEV-ZZ-URA3 | This study |
| KWY10241 | KWY1602, BRL1::BRL1-V5-IAA7-KanMX his3::pGPD1-OsTIR1 | This study |
| KWY10697 | KWY1602, BRL1::BRL1-V5-IAA7-KanMX his3::pGPD1-OsTIR1 ura3::Nup82-yEGFP leu2-3::BRL1(I395D)-mCherry-LEU2 | This study |
| KWY10445 | KWY1602, BRL1::BRL1-V5-IAA7-KanMX leu2::pGPD1-OsTIR1 MLP1::MLP1-V5-loxP-mCherry_HygMX-loxP_GFP his3::CRE-EBD78 | This study |
| KWY10450 | KWY1602, BRL1::BRL1-V5-IAA7-KanMX leu2::pGPD1-OsTIR1 NUP82::NUP82-V5-loxP-mCherry_HygMX-loxP_GFP his3::CRE-EBD78 | This study |
| KWY10451 | KWY1602, BRL1::BRL1-V5-IAA7-KanMX leu2::pGPD1-OsTIR1 NUP188::NUP188-V5-loxP-mCherry_HygMX-loxP_GFP his3::CRE-EBD78 | This study |
| KWY10452 | KWY1602, BRL1::BRL1-V5-IAA7-KanMX leu2::pGPD1-OsTIR1 NUP133::NUP133-V5-loxP-mCherry_HygMX-loxP_GFP his3::CRE-EBD78 | This study |
| KWY9485 | KWY166, BRL1/brl1Δah | This study |
| KWY9489 | KWY166, BRL1/brl1(I395D) | This study |

temperature (RT), washed three times 10 min in PBST (1× PBS pH 7.4; 0.1% Tween-20) followed by 30 min incubation with secondary antibody at RT. Membranes were washed again three times for 10 min in PBST before fluorescence signal was imaged with the CLx ODYSSEY (Li-COR). Primary antibodies used were mouse monoclonal α-V5 (Invitrogen, R960-25; 1:2000) and rabbit monoclonal α-hexokinase (US Biologicals, H2035-01; 1:3000). Secondary antibodies used were goat α-mouse IgG Alexa Fluor 680 (Thermo Fisher Scientific, A-21057; 1:10,000) and goat α-rabbit IgG IRDye800CW (Li-COR Biosciences, 926-32211; 1:10,000).

## Spot plating assay

For spot assays of strains overexpressing galactose-inducible Brl1 derivatives, strains were grown to saturation in SC medium supplemented with 2% raffinose and 0.1% glucose. Cells were plated

on synthetic medium agar plates supplemented with 2% galactose in a fivefold serial dilution series starting with an $OD_{600}$ of 1.0 using a 48-pin frogger. Strains derived from the *nup116ΔGLFG PMET3-NUP188* background were pre-cultured in SCD lacking methionine and spotted on synthetic medium agar plates supplemented with or without methionine (400 μg/ml).

## Tetrad dissection

Diploid yeast cells were grown on YPD for 1 day at 30°C and then transferred to sporulation plates (SPO; 1% potassium acetate, amino acids to 25% of normal concentration, 0.05% glucose, 2% agar) and incubated for 5 days at RT. To digest the ascus wall, a pinhead-sized cell mass was incubated in 5 μL of Zymolyase 100T 1 mg/mL (ICN) for 3 min at 30°C. Then, 300 μL water was added to stop the digestion, cells were shortly vortexed and spread on a YPD plate. Tetrads were dissected using a Nikon Eclipse Ci-S dissecting scope and incubated for 2 days at 30°C. Spore clones were tested for genotype segregations by sequencing.

## Fluorescence microscopy

Cells were immobilized in a 384-well glass-bottom plate (MatriPlate) coated with concanavalin A (Sigma-Aldrich). Imaging was performed with a ×100 Plan-Apo VC objective (NA 1.4, Nikon) on a Nikon inverted epifluorescence Ti microscope equipped with a Spectra X LED light source (Lumencore) using the NIS Elements software (Nikon) at 30°C unless indicated differently. Images were acquired with a Flash 4.0 sCMOS camera (Hamamatsu) and processed using ImageJ software.

Imaging of strains expressing the Nup170-RITE constructs was performed with a ×100 Plan Apo lambda objective (NA 1.45 oil DIC WD 0.13 mm, Nikon) on a Nikon inverted Widefield Ti2-E microscope equipped with a Spectra III light engine and an Orca Fusion BT camera using the NIS Elements software (Nikon) at RT. Images were processed using the Denoise.ai and Clarify.ai algorithms from NIS Elements software and Fiji (*Schindelin et al., 2012*).

## Fluorescence recovery after photobleaching

FRAP experiments were performed at RT on a Leica TCS SP8-AOBS microscope using a 63 × 1.4 NA Oil HC PL APO CS2 objective. Unidirectional scanner at speed of 1400 Hz, NF488/561/633, an AU of 1.5, and a FRAP booster for bleaching were applied for every FRAP experiment using the PMT3 (500–551 nm) and PMT5 (575–694 nm) detectors. Image sizes of 512 × 75 at 80 nm/px were used together with line accumulation of two, yielding a time interval of 120 ms per frame. Then, 20 pre-bleach and 200 post-bleach frames were acquired. A 488 nm argon laser line was used at 20% base power in addition to a 561 nm DPSS laser line. Imaging was conducted with 1.5% laser intensity with a gain of 800 to illuminate the GFP and 0.3% of the 561 laser power to illuminate mCherry. Bleaching was performed in a manually defined elliptical region comprising approximately one-third of the cell nucleus at 100% laser power of both laser lines for 120 ms. For the case of mutant Brl1, the region was chosen to encompass part of a bright region (herniation). The mobility of GFP-labeled proteins in the bleached NE region was evaluated by quantifying the signal recovery in the bleached region. Extracellular background ($I_{bg}$) was subtracted from the intensity of the bleached region ($I_{bl}$) and the values were bleach-corrected by normalizing for total cell intensity ($I_{total}$) resulting in $(I_{bl}\text{-}I_{bg})/(I_{total}\text{-}I_{bg})$ (*Bancaud et al., 2010*) using custom-written scripts (*Figure 6—source code 1*) in MATLAB (MathWorks) and plotted with Prism 7 (GraphPad).

## Fluorescence microscopy of RITE constructs

All strains expressing NUP-RITE constructs were grown to mid-log phase in SCD supplemented with 300 μg/mL hygromycin B (Roche) to select for non-recombined cells. Prior to imaging, cells were centrifugally collected and recovered for 1 hr in SCD without hygromycin B. Recombination was induced by addition of β-estradiol (1 μM f.c., Sigma-Aldrich) and cells were imaged 3 hr post induction.

Strains expressing NUP170-RITE constructs were grown to mid-log phase in SD-URA to select for non-recombined cells. Prior to imaging, recombination was induced by addition of β-estradiol (1 μM f.c., Sigma-Aldrich) and uracil and cells were imaged ~30 min (new Nup170-RITE) or ~5 hr (old Nup170-RITE) post induction.

## Quantitative image analysis

We used the automated imaging analysis pipeline NuRim to quantify the fluorescence intensity signal in the NE for various NUP GFP fusion proteins (*Rajoo et al., 2018*; *Vallotton et al., 2019*). In brief,

nuclear contours were called in an unbiased manner based on the fiducial marker dsRED-HDEL. Fluorescence intensities of NUP-yEGFP along these contours were then extracted in ImageJ. NE intensity profiles with large foci in the NE were excluded by using an intensity value standard variation cutoff of 200; in Brl1-depleted cells, this accounted for maximum 35% of the generated masks. Brightness and contrast of the presented images were adjusted the same for all images in one panel unless otherwise indicated using Fiji. Graphical representation of the data was carried out in R.

For the co-localization plots (*Figure 6D*, *Figure 6—figure supplement 1B*), at least 36 line plots (exact number indicated in respective figures) were manually generated in Fiji. Values for each line plot were centered according to the peak intensity of the Brl1(I395D)-mCherry signal and plotted as mean with SD. Graphs were created with Prism 9.

In strains expressing NUP170-RITE fusion proteins, the NE contours were manually delineated based on the Brl1-mCherry signal and the intensity profiles obtained using Fiji. Pearson's correlation coefficient between intensity values in green and red channels was calculated. Only cells with foci in both red and green channels were selected for quantification. The following cells were excluded: NE contours with no signals in any of the two channels, cells with a strong red background signal, and cells that did not undergo recombination.

The fraction of cells with an NE foci of overexpressed Brl1(I395D) upon Brl1 depletion or treatment with a solvent control (*Figure 6—figure supplement 1C*),were manually counted in a blinded manner.

## Recombinant protein expression and purification

The fusion proteins 6xHis-MBP-TEV-yEGFP, 6xHis-MBP-TEV-ahBrl1-yEGFP, 6xHis-MBP-TEV-ahBrl1-(F391D)-yEGFP, and 6xHis-MBP-TEV-ahBrl1(I395D)-yEGFP were expressed in *E. coli* BL21 RIL cells. Bacteria were cultured in 1 L YT (0.8% Bacto Tryptone, 0.5% Bacto Yeast Extract, 86 mM sodium chloride) to $OD_{600}$ = 0.8–1.0 at 37°C, and protein expression was induced by adding 0.2 mM IPTG (AppliChem A10080025) and cells were grown overnight. The next day cells were harvested in a AF6.100 rotor (Herolab) for 15 min at 5000 rpm at 4°C. Pellets were resuspended in 20 mL Tris–HCl (20 mM, pH 7.5) supplemented with 10 µg/mL DNase I (Roche, 10104159001) and ½ tablet cOmplete Protease Inhibitor Cocktail (Sigma-Aldrich, 05053489001). Cells were lysed using the Avestin Emulsiflex c5 (ATA Scientific) and centrifuged at 4°C for 15 min at 12,000 rpm in the SS-34 rotor (Thermo Scientific). Supernatant was filtered through a 0.45 µm filter, applied to ~1 mL Ni-NTA Agarose (QIAGEN 30210), and incubated for 1 hr at 4°C. The agarose was washed thoroughly with 20 mM Tris–HCl pH 7.5, 500 mM NaCl, 30 mM imidazole prior to elution with 20 mM Tris–HCl pH 7.5, 500 mM NaCl, 400 mM imidazole. Purified proteins were dialyzed overnight in 20 mM Tris pH 7.5, 150 mM NaCl at 4°C, concentrated in 1 mL in a Vivaspin Turbo 4 (30,000 MWCO, Sartorius VS04T22) and further purified on a Superdex 75 10/300 gel filtration column (GE Healthcare).

## Liposome-binding assay

Liposome generation and flotation was performed as described in *Vollmer et al., 2015*. In short, *E. coli* polar lipids (Avanti Polar Lipids) dissolved in chloroform and supplemented with 0.2 mol% 18:1 Liss Rhodamine PE (Avanti Polar Lipids) were vacuum-dried on a rotary evaporator, dissolved as liposomes in PBS by freeze/thawing cycles and extruded by passages through Nuclepore Track-Etched Membranes (Whatman) with defined pore sizes using an Avanti Mini-Extruder to generate small unilamellar liposomes of defined sizes. For liposome flotations, proteins (6 µM) were mixed 1:1 with liposomes (6 mg/mL) and floated for 2 hr at 55,000 rpm in a TLS-55 rotor (Beckman) at 25°C through a sucrose gradient. Binding efficiency was determined by Western blot analysis using an EGFP antibody (Roche, 11814460001, 1:2000). As secondary antibody, an anti-mouse, horseradish peroxidase conjugated antibody (Calbiochem, 401215, 1:5000) was used. The ImageQuant LAS-4000 system (Fuji) and the AIDA software were used to compare band intensities of start materials with floated liposome fraction.

## Sequence alignment

Sequence alignment was performed using the COBALT web server (https://www.ncbi.nlm.nih.gov/tools/cobalt/re_cobalt.cgi) and visualized using Jalview (*Waterhouse et al., 2009*).

## Cryo-FIB milling of yeast cells

Brl1 of exponentially growing yeast cells was inducibly depleted as described above. As control for the Brl1 degradation, cells lacking *Os*TIR1 were treated for 4–4.5 hr with auxin. Brl1(I395D)-overexpressing cells were grown as described above, and as a control, cells overexpressing Brl1 were cultured for 6 hr in SC 2% galactose. Cells were pipetted onto Quantifoil Cu R2/1 grids (Quantifoil), blotted for ~4 s, and plunge frozen using a manual plunger. Blotting was performed manually from the backside of the grid. Cryo FIB-milling was performed essentially as previously described (*Wagner et al., 2020*). In brief, the grids were transferred to a Leica BAF060 system equipped with a Leica cryo transfer system at –160°C and grids were coated with ~5 nm Pt/C. Afterward, the grids were transferred to a Zeiss Auriga 40 Crossbeam FIB-SEM equipped with cryostage and cryo-transfer shuttle. An organometallic platinum layer was deposited using the integrated gas injection system. Cells were milled in three steps at 30 kV using rectangle patterns (240 pA to ~200 µm, 120 pA to ~100 µm, 50/30 pA to <0.3 µm) to a target thickness of <250 nm, and samples were stored in liquid nitrogen until data acquisition.

## Cryo-electron tomography

Tilt series of FIB-milled lamella were acquired using a Titan Krios equipped with a Gatan Quantum Energy Filter and a K2 Summit electron detector or a Titan Krios G3i equipped with a Gatan BioQuantum Energy Filter and K3 direct electron detector at 300 kV. Tilt series were acquired using SerialEM (*Mastronarde, 2003*) at a pixel size of 3.4 Å at the specimen level. The target defocus was set to –4 to –7 µm, and tilt series were acquired using a dose symmetric tilt scheme (*Hagen et al., 2017*) from –65° to 55° with an increment of 3° and a total dose of ~140 electrons per angstrom squared.

## Tomogram reconstruction

Movie frames were aligned using IMODs alignframes function (*Mastronarde and Held, 2017*). Tilt series were processed and aligned using the IMOD suite. Alignment was performed using the 4× binned projections and the patch tracking function in IMOD. Outliers in patch tracking (e.g., patch aligning on ice contamination) were manually corrected. Occasionally, contaminations on top of the lamella were used as fiducial markers. Overview tomograms for particle picking were reconstructed using the SIRT-like filter with 12× iterations and 4× binning. NPCs and NPC herniations coordinates and rough orientation along the NE were picked and determined manually.

## Quantification of herniations and NPCs

For the quantification of herniations and NPCs in Brl1-depleted cells, we used 51 tomograms. For this analysis, we also included tomograms with lower quality, which we did not include in the subtomogram analysis described below. For the control condition, we used 27 tomograms of cells subjected to the same treatment but without *Os*TIR1 plasmid. For the quantification of herniations and NPCs in Brl1(I395D)-overexpressing cells, 50 tomograms were analyzed. For our control condition in cells overexpressing Brl1 without the point mutation, we used 17 tomograms. To compensate for the different surface area of NE in tomograms, we normalized the number of NPCs and herniations by the area of NE in each tomogram. For this, we manually segmented the NE in three tomographic slices using the drawing tool in IMOD. Segmentations for all other slices were interpolated. We then calculated the distance between segmentation points to determine the total visible surface area in MATLAB and used Prism 9 (GraphPad) for visualization.

## Subtomogram averaging

Subtomograms containing NPCs or herniations were reconstructed in IMOD from unbinned, dose-filtered and CTF-corrected tilt series. CTF was corrected as described previously by estimating the mean defocus by strip-based periodogram averaging. With the information for the mean defocus, the tilt angle, and axis orientation, the defocus gradient for each projection was calculated, and according to the defocus gradient, each projection was CTF-corrected by phase flipping (*Eibauer et al., 2015*). CTF-corrected stacks were dose-filtered using the IMOD mtffilter function and subtomograms reconstructed using IMOD.

We reconstructed 85 herniation-containing subtomograms from 31 tomograms of Brl1-depleted cells. Based on the curvature of the ONM, herniations were classified manually into INM evaginations (n = 25) and herniations (n = 60). When the ONM was not or only slightly deformed, we classified the herniation as an INM evagination (examples in *Figure 3—figure supplement 1D*). As a control, we reconstructed 29 mature NPC from 19 tomograms of the same dataset. For Brl1(I395D)-overexpressing cells, we reconstructed 47 herniations from 21 tomograms.

Prealigned full NPCs/herniations were aligned using iterative missing wedge-weighted subtomogram alignment and averaging using the TOM toolbox (*Nickell et al., 2005*; *Förster et al., 2005*) by merging the half set averages after each iteration as a template for the next iteration. 8× binned subtomograms were aligned using eightfold rotational symmetry. For averaging mature NPCs and Brl1(I395D) herniations, we further extracted eight protomers (4× binned) according to the eightfold symmetry of the NPC. Protomers outside the lamella were excluded by manual inspection. For mature NPCs, we used 179 protomers (53 excluded from 232 protomers) for the final average. For Brl1(I395D) herniations, we used 237 protomers (139 excluded from 376 protomers) for the final average.

For the different forms of herniations in Brl1-depleted conditions, protomer alignment did not improve the maps. We think that resolution of these averages is limited because of the high heterogeneity of herniations in overall shape and membrane curvature. We also believe that the electron-dense center of herniations in Brl1-depleted cells limited the resolution of our average. Several trials with different masks, bandpass filters, and classification based on membrane curvature did not improve resolution. Furthermore, our subtomogram average of herniations in Brl1(I395D)-overexpressing cells, which do not have an electron-dense center, shows distinct IR-like densities and is better resolved although less subtomograms were used.

Resolution was determined using masked half maps and the web server https://www.ebi.ac.uk/emdb/validation/fsc. Final maps were filtered according to the achieved resolution at Fourier shell correlation (FSC) 0.5 (INM evaginations: 12 nm, herniations (Brl1-AID): 11 nm, Brl1(I395D) herniation: 8 nm, mature NPC: 8 nm). The full-pore map for the mature NPC and the Brl1(I395D) herniations was stitched from single protomers by fitting the protomer average into the full-NPC map in UCSF Chimera (*Pettersen et al., 2004*).

## AlphaFold prediction

To predict the structure of Brl1 and Brl1 homologues, we used the Python script for AlphaFold2.1.1 (*Jumper et al., 2021*) implemented in SBGrid with standard settings and the *mode_preset=monomer_ptm* setting. Since we predicted the structure of Brl1 locally, it is not identical to the structure in the AlphaFold database. However, the structured part is almost identical (rmsd: 1.35 Å) and only the unstructured N- and C-termini deviate significantly between the structures. Visualization of prediction metrics was generated using the following Jupyter Notebook in Anaconda: available here.

## Dimension measurements on onion-like herniations in Brl1(I395D)-overexpressing cells

4× binned tomograms of Brl1(I395D)-overexpressing cells were processed in Fiji using a Gaussian blur with a sigma of 1 and contrast was inverted. Per onion-like herniation, 3–4 line plots were generated and exported to MATLAB. Peaks (= membranes) of the line plots were determined by Gaussian fit of the peaks. Eleven onion-like herniation from eight tomograms were analyzed. The same procedure was performed on the NE of tomograms of Brl1-overexpressing cells. Six to nine line plots per NE were generated and five NE from five tomograms were analyzed. Only tomograms where the herniation or the NE were roughly perpendicular in the section were used. Visualization and statistical tests were performed in Prism 9.

## Visualization of tomograms and subtomograms

Snapshots of single NPCs or herniations were extracted from 4× binned tomograms reconstructed in IMOD using the SIRT-like filter with 12 iterations and visualized using tom_volxyz (*Figure 3—figure supplement 1D*, *Figure 7C*). All tomographic slices shown were reconstructed using IMOD's SIRT-like filter with 12 iterations, and slice thickness is indicated in the figure legends.

All procedures were implemented in MATLAB and using the TOM toolbox. Chimera, IMOD, and AlphaFold were used as part of SBGrid (*Morin et al., 2013*).

## Preparation of IgG-coupled Dynabeads

IgG-coupled Dynabeads were prepared as described in *Alber et al., 2007*. 150 mg of magnetic Dynabeads were resuspended in 9 mL fresh 0.1 M sodium phosphate buffer (22.5 mM monosodium phosphate, 81 mM disodium phosphate, pH 7.4). Bead suspension was vortexed for 30 s followed by a 10 min incubation at RT under constant agitation. Then, beads were placed onto a magnetic holder, clear buffer was aspirated off, and beads were washed once with 4 mL 0.1 M sodium phosphate buffer. Antibody mix was prepared by resuspending 50 mG rabbit IgG powder in 2.1 mL distilled water and spinning down the mixture for 10 min at 15,000 × $g$ in a tabletop centrifuge precooled to 4°C. Clear supernatant was transferred to a fresh Falcon tube and 4.275 mL 0.1 M sodium phosphate buffer was added. To this, 3 M ammonium sulfate buffer (3 M ammonium sulfate dissolved in 0.1 M sodium phosphate buffer) was added slowly, constantly shaking the mixture. The antibody mix was then filtered through a 22 µm Millex GP filter and was ready for use. The magnetic Dynabeads were incubated with the antibody mix for ~20 hr on a rotating wheel at 30° C. Thereafter, beads were briefly washed once with 100 mM glycine HCl pH 2.5, 10 mM Tris–HCl pH 8.8 and 100 mM freshly prepared triethylamine. This was followed by four 5 min washes with PBS pH 7.4 and two 10 min washes with PBS pH 7.4 containing 0.5% Triton X-100. Beads were finally resuspended in a total of 1 mL PBS supplemented with 0.02% sodium azide, resulting in a concentration of 100 mg beads/mL and stored at 4°C.

## Metabolic labeling assays

Yeast strains harboring endogenously tagged Brl1-ZZ or Nup170-ZZ fusion proteins were cultured for a minimum of 16 hr at 30°C in light SCD. Cell culture samples equivalent to 250 mL $OD_{600}$ = 1.0 were collected by filtration on an 0.8 µL nitrocellulose membrane. During harvesting, the cells were briefly washed twice with 25 mL distilled water directly on the filter membrane and then snap-frozen in liquid nitrogen. Samples corresponding to the 0 hr time point were collected immediately before labeling onset. Thereafter, cell cultures were pulse labeled as follows: the amount of log-phase cell cultures corresponding to 650 mL of $OD_{600}$ = 1.0 was washed on the filter with 50 mL heavy SCD containing $^{13}C_6$, $^{15}N_2$ l-lysine (25 mg/L, Cambridge Isotope Laboratories) and reinoculated in heavy SCD. For the experiments with the Brl1-AID constructs, cultures were split in half and switched to heavy SCD containing IP6 (4 µM f.c.) and either auxin (500 µM f.c.) or the equivalent volume of ethanol for the solvent control. Post-labeling time points were collected at regular intervals as described above. During the time course, all cultures were maintained in logarithmic growth by periodic dilution with the respective prewarmed medium.

## Affinity pulldowns

All the following procedures were performed under ice-cold conditions. Frozen yeast pellets were resuspended in 1 mL lysis buffer (20 mM HEPES pH 7.5, 50 mM KOAc, 20 mM NaCl, 2 mM MgCl$_2$, 1 mM DTT, 10% v/v glycerol) and transferred into 2 mL screw-cap micro tubes (Sarstedt Inc) pre-filled with ~1 mL of 0.5 mm glass beads (BioSpec Products). Cell material was spun down in a tabletop centrifuge, and the tubes were filled up completely with lysis buffer. During this step, extra care was taken to avoid any air inclusion. Cells were mechanically lysed with a mini BeadBeater-24 (BioSpec Products) in four 1 min cycles at 3500 oscillations per minute with 1 min cooling intermissions in ice water. Cell lysates were then spun down for 30 s at 15,000 × $g$ in a tabletop centrifuge precooled to 4°C. 150 µL of the supernatant was frozen in liquid nitrogen for the analysis of the source cell lysates. For the APs, 1 mL of the supernatant was supplemented with 110 µL 10× Detergent mix (protease inhibitor cocktail [Sigma-Aldrich], 5% v/v Triton x-100, 1% v/v Tween-20 in lysis buffer) and 2 mg IgG Dynabeads, pre-equilibrated two times with equilibration buffer (0.5% v/v Triton X-100 and 0.1% v/v Tween-20 in lysis buffer). The remaining supernatant was frozen in liquid nitrogen for the analysis of the source cell lysates. Following a 30 min incubation of the AP samples at 4°C under constant agitation, the beads were washed twice with 1 mL wash buffer (0.1% v/v Tween-20 in lysis buffer). Proteins were eluted in 40 µL 1× Laemmli sample buffer for 2 min at 50°C. Finally, elutes were completely denatured at 95°C for 5 min and frozen in liquid nitrogen.

## In-gel tryptic digestion

Eluted proteins were electrophoretically concentrated by SDS-PAGE in a 4% acrylamide stacking gel. Proteins were visualized by incubation with Coomassie SimplyBlue SafeStain (Invitrogen), followed by

destaining for at least 14 hr in distilled water. Protein bands were cut out and processed according to a standard in-gel digestion protocol. In brief, disulfide bonds were reduced with dithiothreitol (6.5 mM DTT in 100 mM ammonium bicarbonate) for 1 hr at 60°C, proteins were alkylated with iodoacetamide (54 mM in 100 mM ammonium bicarbonate) for 30 min at 30°C in the dark, and finally tryptically digested with 1.25 µg of sequencing grade porcine trypsin (Promega) in 100 mM ammonium bicarbonate at 37°C for 16 hr. The resulting peptides were loaded in pre-equilibrated C18 BioPureSPN mini columns (The Nest Group, Inc), washed, and desalted three times with Buffer A (0.1% formic acid in HPLC-grade water), eluted three times with 50 µL Buffer B (50% acetonitrile, 0.1% formic acid in HPLC-grade water), and finally recovered in 12.5 µL Buffer A supplemented with iRT peptides (1:50 v:v, Biognosys).

## Tryptic digestion of source cell lysates

The source lysates of Brl1 APs 90 min post labeling were adjusted to 50 µL with a protein concentration of 4 µg/µL with lysis buffer as determined by the Bradford method (Bio-Rad). Samples were diluted with 200 µL guanidine chloride (7 M in 100 mM ammonium bicarbonate) to reach a final guanidine chloride concentration of 5.6 M. Disulfide bonds were reduced with DTT (6.5 mM f.c.) at 37°C for 45 min and alkylated with iodoacetamide supplemented to 54 mM f.c. at 30°C in the dark for 30 min. The samples were then diluted to a final guanidine chloride concentration of 1 M with 100 mM ammonium bicarbonate and digested with sequencing grade porcine trypsin (Promega, 1:100 trypsin:protein) for 22 hr at 37°C. Digestion was quenched by addition of 3% (v/v) of 100% formic acid (pH ~ 2.0) and peptides were desalted in a BioPureSPN MACRO spin columns (The Nest Group, Inc) as described above (Tryptic in-gel digestion). Tryptic peptides were diluted to 1 µg/µL with Buffer A based on $OD_{280}$ readouts and the samples were spiked with 1:50 (v:v) iRT peptides (*Escher et al., 2012*) for the mass spectrometry acquisition.

## Lysate intermixing tests

For the lysate intermixing tests, 200 $OD_{600}$ of an untagged cell culture grown in heavy medium was mixed with the equivalent amount of cell culture expressing an affinity-tagged protein and grown in light medium. The mixture was subjected to the affinity isolation procedure and processed for mass spectrometric analysis as described above. For Brl1-AID strains, the depletion was induced 5 hr prior to harvesting by addition of IP6 (4 µM f.c.) and either auxin (500 µM f.c.) or ethanol for the solvent control.

## DDA MS assays

Unlabeled Brl1 AP samples were assayed in a data-dependent acquisition mode (DDA) for subsequent spectral library generation (see section 'DIA MS data extraction'). LC-MS/MS analysis was performed on an Orbitrap Fusion Lumos Tribrid mass spectrometer (Thermo Scientific) coupled to an EASY-nLC 1200 system (Thermo Scientific). Peptides were separated on an Acclaim PepMap 100 C18 (25 cm length, 75 µm inner diameter) with a two-step linear gradient from 5 to 30% acetonitrile in 120 min and from 30 to 40% acetonitrile in 10 min at a flow rate of 300 nL/min. The DDA acquisition mode was set to perform one MS1 scan followed by MS2 scans for a cycle time of 3 s. The MS1 scan was performed in the Orbitrap ($R$ = 120,000, 100,000 AGC target, maximum injection time of 100 ms and scan range 350–1400 m/z). Peptides with charge state between 2 and 7 were selected for fragmentation (isolation window: 1.6 m/z and fragmentation with HCD, NCE 28%) and MS2 scans were acquired in a Orbitrap ($R$ = 30,000, 100,000 AGC target, maximum injection time of 54 ms). A dynamic exclusion of 30 s was applied.

## DIA MS assays

Data-independent acquisition (DIA) assays were performed on two different instrument setups (Orbitrap Fusion Lumos Tribrid [DIA:A] for the Brl1 AP samples and Orbitrap QExactive+ [DIA:B] for the Nup170 AP samples and the lysis intermixing assays).

### DIA:A

LC-MS/MS analysis was performed on an Orbitrap Fusion Lumos Tribrid mass spectrometer (Thermo Scientific) coupled to an EASY-nLC 1200 system (Thermo Scientific). Peptides were separated as

described in the section 'DDA MS assays.' DIA was performed with the following parameters: one MS1 scan (350–2000 m/z) with variable windows from 350 to 1150 m/z with 1 m/z overlap for a cycle time of 3 s. Ions were fragmented with HCD (NCE 28%). The MS1 scan was performed at 120,000 R, 200,000 AGC target, and 100 ms injection time, the MS2 scan at 30,000 R, 500,000 AGC target, and 54 ms injection time.

### DIA:B

LC-MS/MS was performed on an Orbitrap QExactive+ mass spectrometer (Thermo Fisher) coupled to an EASY-nLC-1000 liquid chromatography system (Thermo Fisher). Peptides were separated using a reverse-phase column (75 µm ID × 400 mm New Objective, in-house packed with ReproSil Gold 120 C18, 1.9 µm, Dr. Maisch GmbH) across a two-step linear gradient: from 3 to 25% acetonitrile in 160 min and from 24 to 40% in 20 min at a flow rate of 300 nL/min. DIA was performed with the following parameters: one MS1 scan (350–1500 m/z) with 20 variable windows from 350 to 1400 m/z with 1 m/z overlap. Ions were fragmented with HCD (NCE 25%). The MS1 scan was performed at 70,000 R, 3,00,000 AGC target, and 120 ms injection time, the MS2 scan at 35,000 R, 1,000,000 AGC target, and auto injection time.

## PRM MS assays

Parallel reaction monitoring (PRM) assays were performed with the two different instrument setups described in the section 'DIA MS assays' (Orbitrap Fusion Lumos Tribrid [PRM:A] and Orbitrap QExactive+ [PRM:B]).

### PRM:A

Peptides were separated as described in the section 'DIA:A.' MS analysis of the targeted peptides was set up with the combination of one untargeted MS1 scan (120,000 R, 200,000 AGC Target, injection time 100 ms) followed by 106 scheduled targeted scans (AGC = 450,000, resolution and injection time was variable based on peptide response) using an isolation window of 1.8 m/z and HCD fragmentation (NCE = 28%).

### PRM:B

Peptides were separated using a reverse-phase column (75 µm ID × 400 mm New Objective, in-house packed with ReproSil Gold 120 C18, 1.9 µm, Dr. Maisch GmbH) across a linear gradient from 5 to 40% acetonitrile in 90 min. MS acquisition of the targeted peptide was set up with the combination of one untargeted MS1 scan (70,000 R, 3,000,000 AGC Target, injection time 100 ms) followed by 55 scheduled targeted scan (AGC = 1,050,000, resolution 35,000, and 110 ms injection time) using an isolation window of 1.8 m/z and HCD fragmentation (NCE = 27%).

## PRM data analysis

The metabolic labeling of proteins in the source cell lysates was analyzed by PRM MS 90 min after the pulse labeling onset. Probed proteins included NUPs that exhibited outstandingly high or low labeling kinetics in the Brl1 AP, two NTRs (Kap123 and Mex67), and two randomly picked co-purified proteins (Rrp5 and Acc1). Precursors for the targeted analysis were selected based on good labeling consistency with other peptides of the same protein, high intensity and low number of missing values in the Brl1 APs. Peptides with missed cleavage sites or with cysteine and methionine residues were excluded when possible. All proteins were represented by 2–5 peptides. Targeted data analysis was performed as described in the section 'PRM MS assays' and the resulting intensities were analyzed with Skyline daily (64 bit, 20.1.1.213 version). Precursor ions identified by at least 3–4 coeluting light and heavy transitions were quantified by manual peak integration. For precursor ions that were well detected in both heavy and light channels, the respective intensities were calculated as the sum of the top 3 most intense transitions in each channel. Fractional protein labeling was quantified as $H/(H + L)$, where H and L are the summed intensities of the above protein-born precursors in heavy and light channels, respectively.

## DIA MS data extraction

Two hybrid spectral libraries were generated with Spectronaut v.15 (Biognosys AG) using the combination of 20 DDA and 30 DIA datasets originating from APs with 10 NUP baits (*Onischenko et al.,*

*2020*), and 4 DIA and 6 DDA datasets from Brl1 and Nup170 bait APs acquired in this study. The label-free assay library contained b and y transition ions (for a total of 3918 protein groups, 75,780 precursors, and 105,089 transitions). The SILAC assay library comprised y transitions only, with the heavy-channel (K+8.014199) generated in silico using the 'inverted spike in' workflow (for a total of 3825 protein groups, 97,069 precursors). Only tryptic peptides with a maximum of two missed cleavages were considered. Carbamidomethylation was set as fixed modification and methionine oxidation was set as variable modification. Spectra were searched against the SGD protein database (downloaded on 13/10/2015, 6713 entries) concatenated with entries for contaminants and iRT peptides using a 1% FDR control at peptide and protein level.

The label-free and SILAC DIA datasets were extracted with the respective spectral libraries using Spectronaut v.15 (Biognosys AG). Default settings were used for the chromatogram extraction, except the machine learning option was set to 'across experiment' and 'cross-run normalization' was excluded. The ion intensities at the fragment level were exported for further analysis in R. Raw MS data, the spectral libraries, and the DIA data extractions generated with Spectronaut are uploaded in the PRIDE repository.

## Labeling quantification in affinity pulldowns

Analysis of protein labeling in KARMA assays with Brl1 bait was implemented in R ('*Labeling_BRL1AP.R*'). Initially, low-quality fragment ions were excluded from further analysis based on the Spectronaut 'F.ExcludedFromQuantification' flag. Additionally, only proteotypic y-type fragment ions with a single lysine residue that were found in both heavy and light channels were retained. The remaining fragment ion intensities were summed for each precursor in heavy and light channels as the respective precursor intensity. Unreliable precursor ions that were detected in fewer than two out of three biological replicates in any of the three post-labeling time points (30, 60, and 90 min) were also excluded. The fractional labeling of the remaining precursor ions was then calculated as $H/(H + L)$, where H and L are the precursor intensities in heavy and light channels. The median protein labeling within each sample was computed as the median fractional labeling of all precursors. As an additional quality criterion, we also computed the root mean square error (RMSE) of the labeling values for every precursor from the respective protein median across all nine samples. For any protein, the precursors with the 50% highest RMSEs were discarded, and the final protein labeling was computed as the median fractional labeling of the remaining high-quality precursors. As a last filtration step, proteins with visually noisy labeling trajectories across the biological replicates and time points were excluded in a blinded manner. For the comparison of NUP labeling rates with Brl1 bait and 10 NUP baits (*Figure 2B*, *Figure 1—figure supplement 1C*), only NUPs reproducibly found with all 11 baits were considered. For *Figure 1—figure supplement 1C*, the median from three biological replicates was taken and labeling values were normalized to the bait labeling.

Protein labeling in the Nup170 APs of Brl1-AID strains ('*Labeling_NUP170AP_BRL1AID.R*') was analyzed the same way as for Brl1 APs, except that precursor ions found in at least one out of three replicates in all post-labeling time points were also considered for quantification. The fractional labeling ratio between the auxin-treated cells and the ethanol solvent control was calculated for each biological replicate and post-labeling time point (4 hr, 4.5 hr, and 5 hr) and the average ± SEM is plotted (*Figure 4C and D*).

For the lysate intermixing assays ('*LysisIntermixingTest.R*'), the protein fractional labeling was quantified essentially as described above except that low-intensity precursor ions (<100) were filtered out and only proteins characterized by more than three precursor ions were considered (due to the low extent of intermixing, Brl1 bait is only characterized by two precursor ions that were found in both heavy and light). To get the intermixing extent, NUP fractional labeling was normalized to the mean fractional labeling of all co-purified proteins.

## Label-free quantification in affinity pulldowns

The exact specification of the quantitative analysis pipeline of protein abundances is given by the respective code in R ('*Label-Free_BRL1AP.R*'). In brief, NUP abundances in the AP with Brl1 bait, low-quality fragment ions were excluded based on Spectronaut 'F.ExcludedFromQuantification' flag. For each proteotypic precursor ion, all the remaining fragment ions were summed and the resulting intensities were median normalized across samples. Precursor ions that were not found in all three

biological replicates were omitted. Protein intensities were calculated based on the average of the top 3 most intense precursor ions, only considering NUPs and NTRs characterized by a minimum of three ions and also reproducibly found in the KARMA assay with Brl1 bait. The intensity of proteins in APs with NUP baits was essentially quantified the same, except that the only precursors found in three replicates with all 10 handles were considered for quantification. To assess the enrichment differences between the early and late tier baits for all 1523 co-purified proteins (*Figure 1—figure supplement 1A*), for each bait the median protein intensity of three biological replicates was taken. Then, the fold difference between the median of all baits from a respective assembly tier was calculated. To focus on non-NPC proteins, NUPs and NTRs were excluded.

## Statistics and data visualization

No statistical method was used to estimate sample sizes. The statistical analysis and data exclusion criteria are discussed throughout the text. Statistical tests were carried out in R v. 4.1.2 (R Project), Excel (Microsoft), or Prism (GraphPad). The statistical test that was performed, sample size n, and p-values are indicated in the respective figure legends. Figure panels were generated using inkscape 1.1 and Adobe Illustrator v. 26.0.3 (Adobe).

## Acknowledgements

We are grateful to all the Weis lab members for many fruitful discussions and helpful suggestions. We want to thank E Fatti for the purification of Brl1 constructs. AK thanks AC Meinema for valuable discussions on and help with image analyses. We thank P Picotti and A Leitner for the access to MS instruments, and the Center of Microscopy and Image Analysis (ZMB), University of Zürich, and ScopeM, ETH Zürich, for microscopy support. We would like to acknowledge the contributions of S Steiger and J Vailliant to early stages of this work during their student projects. AK was a recipient of a Swiss National Science Foundation Marie Heim-Voegtlin Fellowship (grant number: PMPDP3_171317) and AAA acknowledges the support from an EMBO postdoctoral fellowship (grant number: ALTF 910-2021). This study was supported by project grants from the Swiss National Science Foundation (31003A_179275 to KW and 31003A_179418 to OM), a grant from the Research Council of Norway (NFR 315615 to EO, KW, and EN), and a grant from the German Research Foundation (DFG AN377/7-1 to WA).

## Additional information

### Funding

| Funder | Grant reference number | Author |
| --- | --- | --- |
| Schweizerischer Nationalfonds zur Förderung der Wissenschaftlichen Forschung | 31003A_179275 | Karsten Weis |
| Schweizerischer Nationalfonds zur Förderung der Wissenschaftlichen Forschung | 31003A_179418 | Ohad Medalia |
| Norges Forskningsråd | NFR 315615 | Karsten Weis<br>Elad Noor<br>Evgeny Onischenko |
| Deutsche Forschungsgemeinschaft | DFG AN377/7-1 | Wolfram Antonin |
| Marie Heim-Voegtlin Fellowship | PMPDP3_171317 | Annemarie Kralt |
| EMBO postdoctoral fellowship | ALTF 910-2021 | Arantxa Agote-Aran |

| Funder | Grant reference number | Author |
|--------|------------------------|--------|

The funders had no role in study design, data collection and interpretation, or the decision to submit the work for publication.

## Author contributions

Annemarie Kralt, Jonas S Fischer, Conceptualization, Data curation, Investigation, Writing – original draft, Writing – review and editing; Matthias Wojtynek, Conceptualization, Investigation, Writing – original draft, Writing – review and editing; Arantxa Agote-Aran, Roberta Mancini, Marianna Tatarek-Nossol, Investigation; Elisa Dultz, Data curation, Investigation, Writing – review and editing; Elad Noor, Validation, Investigation, Methodology; Federico Uliana, Formal analysis, Investigation; Wolfram Antonin, Conceptualization, Supervision, Funding acquisition; Evgeny Onischenko, Conceptualization, Data curation, Formal analysis, Funding acquisition; Ohad Medalia, Supervision, Funding acquisition; Karsten Weis, Conceptualization, Supervision, Funding acquisition, Writing – original draft, Project administration, Writing – review and editing

## Author ORCIDs

Annemarie Kralt ⬤ http://orcid.org/0000-0001-6705-2102
Matthias Wojtynek ⬤ http://orcid.org/0000-0002-4027-7130
Jonas S Fischer ⬤ http://orcid.org/0000-0002-8652-1813
Elisa Dultz ⬤ http://orcid.org/0000-0003-1114-5523
Elad Noor ⬤ http://orcid.org/0000-0001-8776-4799
Ohad Medalia ⬤ http://orcid.org/0000-0003-0994-2937
Karsten Weis ⬤ http://orcid.org/0000-0001-7224-925X

## Decision letter and Author response

Decision letter https://doi.org/10.7554/eLife.78385.sa1
Author response https://doi.org/10.7554/eLife.78385.sa2

# Additional files

## Supplementary files

• Appendix 1—figure 1—source data 1. Fractional labeling in the source lysate (parallel reaction monitoring [PRM] assay, related to *Appendix 1—figure 1A*).

• Appendix 1—figure 1—source data 2. Fractional labeling in the lysis intermixing tests with Brl1 bait (related to *Appendix 1—figure 1B–D*).

• Appendix 1—figure 2—source data 1. Distance measurements.

• MDAR checklist

• Transparent reporting form

## Data availability

Representative structural cryo-tomogram data have been deposited in PDB accession codes EMD-14502, EMD-14503, EMD14505 and EMD-14506. All raw mass spec data, the spectral libraries, the DIA data extractions generated with Spectronaut and the R code used for analysis was uploaded in the PRIDE repository under the accession numbers PXD032017, PXD032016, PXD032024 and PXD032034.

*Continued on next page*

The following datasets were generated:

| Author(s) | Year | Dataset title | Dataset URL | Database and Identifier |
|---|---|---|---|---|
| Wojtynek M, Fischer JS, Agote-Aran A, Mancini R, Dultz E, Noor E, Uliana F, Tatrek-Nossol M, Antonin W, Onischenko E, Medalia O, Weis K | 2022 | An Amphipathic Helix in Brl1 is Required for Nuclear Pore Complex Biogenesis in S. cerevisiae - KARMA analysis of Brl1 (DIA) | https://www.ebi.ac.uk/pride/archive/projects/PXD032017 | PRIDE, PXD032017 |
| Kralt A, Wojtynek M, Fischer JS, Agote-Aran A, Mancini R, Dultz E, Noor E, Uliana F, Tatrek-Nossol M, Antonin W, Onischenko E, Medalia O, Weis K | 2022 | Cryo-tomogram of FIB-sectioned Brl1-depleted yeast cell | https://www.emdataresource.org/EMD-14502 | EMDataResource, EMD-14502 |
| Kralt A, Wojtynek M, Fischer JS, Agote-Aran A, Mancini R, Dultz E, Noor E, Uliana F, Tatrek-Nossol M, Antonin W, Onischenko E, Medalia O, Weis K | 2022 | Cryo-tomogram of FIB-sectioned non-depleted Brl1 control cells | https://www.emdataresource.org/EMD-14503 | EMDataResource, EMD-14503 |
| Kralt A, Wojtynek M, Fischer JS, Agote-Aran A, Mancini R, Dultz E, Noor E, Uliana F, Tatrek-Nossol M, Antonin W, Onischenko E, Medalia O, Weis K | 2022 | Cryo-tomogram of FIB-sectioned Brl1(I395D) overexpressing cells | https://www.emdataresource.org/EMD-14505 | EMDataResource, EMD-14505 |
| Kralt A, Wojtynek M, Fischer JS, Agote-Aran A, Mancini R, Dultz E, Noor E, Uliana F, Tatrek-Nossol M, Antonin W, Onischenko E, Medalia O, Weis K | 2022 | Cryo-tomogram of a FIB-sectioned Brl1-overexpressing cell | https://www.emdataresource.org/EMD-14506 | EMDataResource, EMD-14506 |
| Wojtynek M, Fischer JS, Agote-Aran A, Mancini R, Dultz E, Noor E, Uliana F, Tatrek-Nossol M, Antonin W, Onischenko E, Medalia O, Weis K | 2022 | An Amphipathic Helix in Brl1 is Required for Nuclear Pore Complex Biogenesis in S. cerevisiae – targeted analysis of cell lysate (PRM) | https://www.ebi.ac.uk/pride/archive/projects/PXD032016 | PRIDE, PXD032016 |
| Wojtynek M, Fischer JS, Agote-Aran A, Mancini R, Dultz E, Noor E, Uliana F, Tatrek-Nossol M, Antonin W, Onischenko E, Medalia O, Weis K | 2022 | An Amphipathic Helix in Brl1 is Required for Nuclear Pore Complex Biogenesis in S. cerevisiae - KARMA analysis of Nup170 with/without Brl1 degradation (DIA) | https://www.ebi.ac.uk/pride/archive/projects/PXD032024 | PRIDE, PXD032024 |
| Wojtynek M, Fischer JS, Agote-Aran A, Mancini R, Dultz E, Noor E, Uliana F, Tatrek-Nossol M, Antonin W, Onischenko E, Medalia O, Weis K | 2022 | An Amphipathic Helix in Brl1 is Required for Nuclear Pore Complex Biogenesis in S. cerevisiae - Lysate intermixing text (DIA) | http://www.ebi.ac.uk/pride/archive/projects/PXD032034 | PRIDE, PXD032034 |

The following previously published dataset was used:

| Author(s) | Year | Dataset title | Dataset URL | Database and Identifier |
|---|---|---|---|---|
| Onischenko E, Noor E, Fischer JS, Gillet L, Wojtynek M, Vallotton P, Weis K | 2020 | Maturation kinetics of a multiprotein complex revealed by metabolic labeling | https://www.ebi.ac.uk/pride/archive/projects/PXD018034 | PRIDE, PXD018034 |

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

## Appendix 1

### Analysis of protein labeling in source cell lysates

To ensure that the observed differences in labeling kinetics in the Brl1 KARMA assay do not simply stem from the protein turnover, we assessed the labeling of several proteins in the source cell lysates of the APs by PRM MS (*Peterson et al., 2012*). As expected, both NTRs and two randomly picked abundant co-purified proteins showed essentially the same metabolic labeling in the source cell lysates and the corresponding APs (*Appendix 1—figure 1A*). NUPs from different assembly tiers did not show a systematic labeling difference in the source cell lysate, as was the case for the AP (*Appendix 1—figure 1A*). This shows that the labeling differences are specific to the Brl1 AP.

### Lysate intermixing assays

To test for the extent dynamic exchange of NUPs during the Brl1 AP procedure, we took advantage of the lysate intermixing assays (*Tackett et al., 2005*) and quantified the metabolic labeling in AP fractions of equal mixes of wildtype culture grown in heavy lysine medium and a Brl1 affinity-tagged strain grown in light lysine medium (*Appendix 1—figure 1B*). Strikingly, we found that all NUPs co-isolated with Brl1 readily intermixed to more than 80% (*Appendix 1—figure 1C*; *Onischenko et al., 2020*). The high extent of intermixing suggests that Brl1 likely associates with NUPs very dynamic. Interestingly, we also observed a pronounced negative correlation between NUP metabolic labeling in KARMA assays and the intermixing tests (*Figure 4—figure supplement 1D*), which may point to a tighter biding of Brl1 to young NUP assemblies.

The high intermixing rates could also be contributed by a lower stability of early NPC assembly intermediates bound by Brl1 compared to the mature NPCs (*Figure 4—figure supplement 1D*). This view is supported (i) by considerably higher intermixing rates for NUPs co-isolated with the early tier Nup170 bait upon Brl1 depletion (*Figure 4—figure supplement 1D*) and (ii) the much higher NUP dynamic exchange rates observed in the early tier Nup170 APs (~60%) compared to previously reported for the outstandingly late recruited Mlp1 bait (~20%, *Onischenko et al., 2020*), which likely isolates a population of otherwise completely assembled NPCs.

### Kinetic state modeling

The high labeling rates in KARMA assays with the Brl1 bait (*Figure 2B*) and the in vivo fluorescence microscopy (*Figure 1D and E*) both indicate that Brl1 preferentially binds to young NPC assemblies. In the lysis intermixing tests, we found that Brl1 interacts with the NPCs highly dynamically (*Appendix 1—figure 1B–D*) and likely also loosely binds to mature structures. Consistent with this, we still detect intermediate and late NUPs in Brl1 AP fractions (*Figure 1—figure supplement 1B*). To assess the binding preference of Brl1 in a more quantitative manner, we made use of the three-step KSM that we have previously developed (*Onischenko et al., 2020*). Note that the KSM that was originally designed to account for completely inaccessible fraction of mature NPCs (e.g., ones that are sequestered and cannot be isolated), but in the context of Brl1, these fractions have a new meaning reflecting the lower affinity of Brl1 to late complexes. Our KSM analysis revealed that a considerable fraction of primarily early tier NUPs become inaccessible to Brl1 bait (*Figure 2C*). The smaller inaccessible fractions of late and intermediate NUPs indicate Brl1 partly dissociates from early intermediates prior to their assembly. By contrast, NUP baits almost never led to inaccessible pools, consistent with them being constitutively bound and not leaving the mature NPCs (*Figure 2C*).

For the quantitative interpretation of KARMA readouts, it is important to consider the effects of dynamic exchange. In KARMA assays, the intermixing between assembly intermediates bound by the bait and those that are not (*Appendix 1—figure 1E*, red arrows) would diminish deviations of labeling rates from the source cell lysates. It is therefore conceivable that our labeling values (*Figures 2D and 4C*) are underestimated due to post-lysis intermixing. Although our lysate intermixing assays indeed show high exchange rates of Brl1-bound NUPs (*Appendix 1—figure 1C*), dynamic exchange within the bait-bound fraction (*Appendix 1—figure 1E*, green arrows), which does not have an influence on the labeling in KARMA assays, contributes to the labeling in the intermixing assay. In the future, for more accurate evaluation of labeling kinetics it might be advantageous to minimize post-lysis intermixing by a mild fixation step similar to what has been done before (*Hakhverdyan et al., 2021*; *Subbotin and Chait, 2014*).

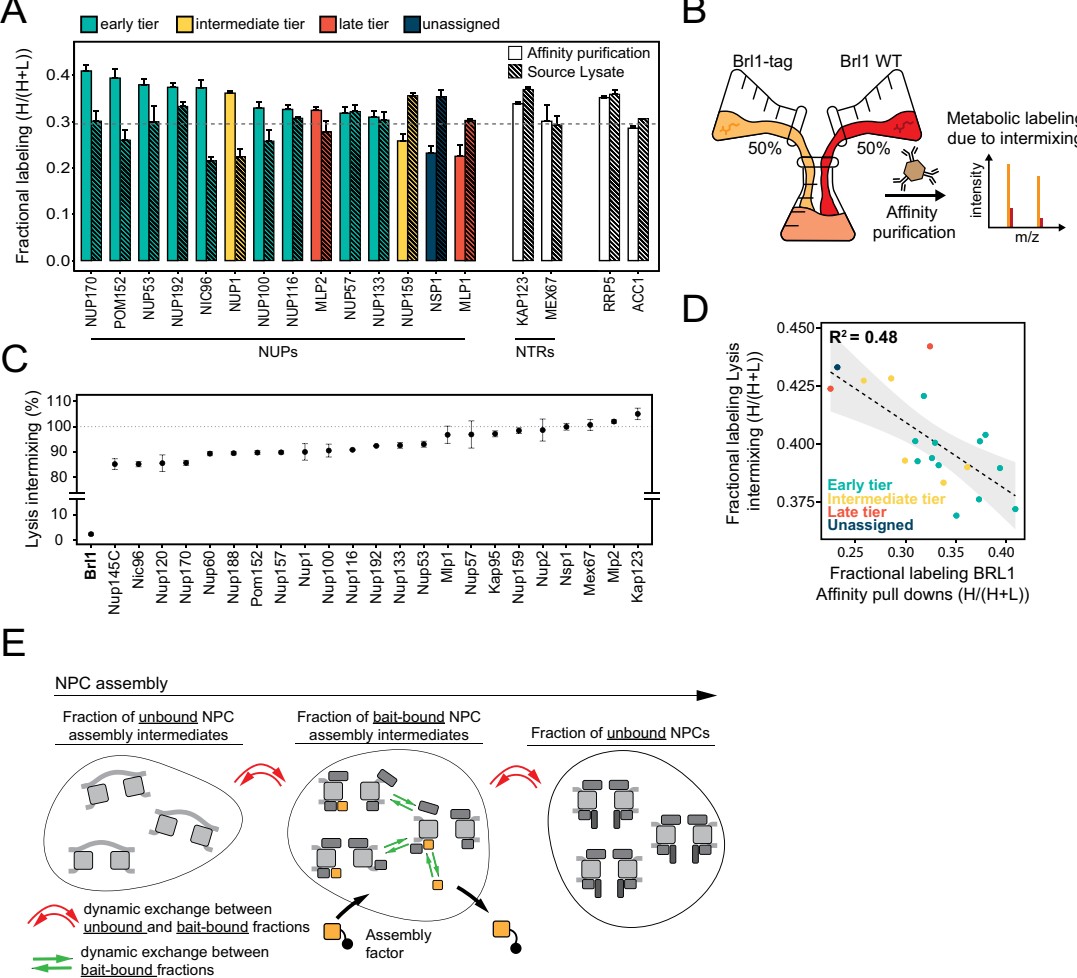

**Appendix 1—figure 1.** Control of protein labeling in KARMA (*Kinetic Analysis of Incorporation Rates in Macromolecular Assemblies*) assays. (**A**) Fractional labeling of nucleoporins (NUPs) and nuclear transport receptors (NTRs) in KARMA assays with Brl1 bait and the respective source lysate, 90 min after labeling onset. Median ± SD of three biological replicates. (**B**) Experiment to test the intermixing dynamics. Equal fractions of an unlabeled Brl1 affinity-tagged strain and a wildtype culture grown in metabolically labeled medium were subjected to the affinity purification procedure. (**C**) Percentage of intermixing for NUPs and NTRs normalized to the mean of all co-purified proteins with Brl1 bait. Median ± SD of three biological replicates. (**D**) Correlation of NUPs between fractional labeling in the intermixing experiment and in KARMA assay with Brl1 bait. Coloring according to the assembly tier. Median of three biological replicates each. (**E**) During complex assembly, proteins can dynamically exchange within the bait-bound fraction (green arrows) or between the bait-bound and unbound fractions (red arrows). The metabolic labeling in KARMA assays is only sensitive to dynamic exchange between bait-bound fractions and unbound fractions, whereas in the lysate intermixing assays both forms of exchange contribute to the observed labeling.

The online version of this article includes the following source data for appendix 1—figure 1:

• **Appendix 1—figure 1—source data 1.** Fractional labeling in the source lysate (parallel reaction monitoring [PRM] assay, related to *Appendix 1—figure 1A*).

• **Appendix 1—figure 1—source data 2.** Fractional labeling in the lysis intermixing tests with Brl1 bait (related to *Appendix 1—figure 1B–D*).

## Model for the development of 'onion-like' herniations

The large multilayered 'onion-like' herniations that form in response to Brl1(I395D) overexpression have not been reported before and the question arises how these structures could assemble at the NE. Interestingly, we noticed a remarkably constant spacing between the two bilayers and the

enclosed nuclear space. Morphometric analysis of the different lipid layers reveals that the middle sheets consisting of two INMs have a very regular spacing of ~13 nm (*Appendix 1—figure 2A and B*). The intermembrane spacing in the outer layer consisting of INM and ONM is significantly wider (~19 nm), which is very similar to the spacing of regular NE in our control condition (~21 nm). Interestingly, the innermost layers show a bimodal distribution with two peaks at heights of the INM-INM middle layers and the INM-ONM outer layers (*Appendix 1—figure 2A and B*). This could be explained by two distinct maturation mechanisms of the onion-like structures. In maturation mode 1, an elongated herniation curls around a part of the cytoplasm and further grows until membrane fusion leads to the enclosure of cytoplasm in the very center of the herniation. Growth and fusion events of subsequent herniations then result in the multilayered herniations (*Appendix 1—figure 2C*). Consistent with this mechanism, we sometimes see ribosome-like densities in the center of the herniations (*Appendix 1—figure 2C*, rightmost panel). In maturation mode 2, a herniation curls over another one, leading to an INM-INM inner bilayer. This is supported by the frequent observation of clustered herniations in which multiple INM sheets are enclosed by a single ONM (*Appendix 1—figure 2D*).

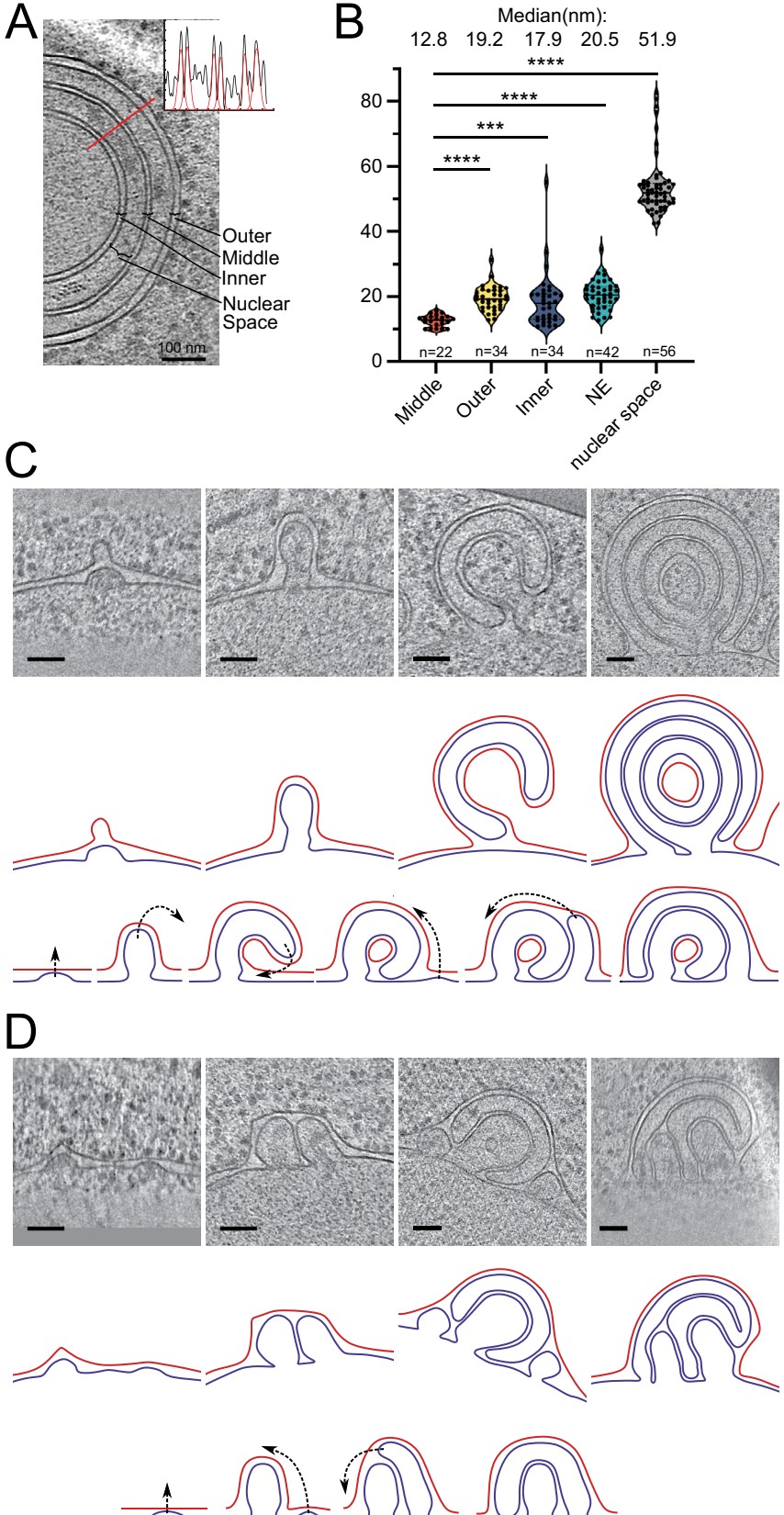

**Appendix 1—figure 2.** Potential maturation processes of onion-like herniations. (**A**) Tomographic slice of an onion-like herniation and an example line plot with fitted Gaussians measured at the indicated red line. Brackets
*Appendix 1—figure 2 continued on next page*

*Appendix 1—figure 2 continued*

indicate how the distances were classified for the plot in **B**. (**B**) Violin plot with individual points of membrane–membrane distances. Mann–Whitney test, ****p-value<0.0001; ***p-value=0.0001. (**C**) Mode 1 for maturation of onion-like herniations. Top panel: tomographic slices of several stages of herniations in Brl1(I395D) overexpressing cells (nucleus always in the bottom); middle panel: membrane segmentation of the herniations of the upper panel. Inner nuclear membrane (INM): blue; outer nuclear membrane (ONM): red; ONM in the center of the very right panel was classified as ONM based on the presence of ribosomes and wider membrane spacing. Lower panel: schematic of how the onion-like herniations mature. (**D**) Same as (**C**) but for mode 2 of the maturation process of onion-like herniations. Scale bar: 100 nm. Slice thickness: 1.4 nm.

The online version of this article includes the following source data for appendix 1—figure 2:

- **Appendix 1—figure 2—source data 1.** Distance measurements.

