## [Editor Report]

The article makes an important advance in our understanding of the nuclear pore complex (NPC) biogenesis mechanism, which has remained a central challenge in the field. Specifically, a compelling combination of in vitro and in vivo data implicates Brl1 as an assembly factor that associates with nascent NPCs. An essential amphipathic helix in Brl1 binds to highly curved membranes in a manner that is required for inner and outer nuclear membrane fusion.

---

## [Decision Letter]

**Decision letter after peer review:**

Thank you for submitting your article "An amphipathic helix in Brl1 is required for membrane fusion during nuclear pore complex biogenesis in *S. cerevisiae*" for consideration by *eLife*. Your article has been reviewed by 3 peer reviewers, one of whom is a member of our Board of Reviewing Editors, and the evaluation has been overseen by Vivek Malhotra as the Senior Editor. The following individual involved in the review of your submission has agreed to reveal their identity: Richard W Wozniak (Reviewer #3).

Essential revisions:

There is considerable enthusiasm for the quality of the data presented in the manuscript and the conclusions, which are well justified. There was debate, however, over the relative advance of the work in comparison to recent studies that have also clearly implicated Brl1 as an NPC assembly factor. Thus, at least two of the reviewers felt that the major conceptual and mechanistic advances of this paper were centered on the role of the Brl1 amphipathic helix in NPC assembly, which remains ill-defined. Thus, in addition to addressing all of the concerns of the reviewers with relevant changes to the text or figures, the request is to provide some additional experimental support for the role that the AH plays in either membrane curvature and lipid sensing or in altering Brl1's interactions with nups.

*Reviewer #1 (Recommendations for the authors):*

1) Beyond the expert implication of Brl1 as an NPC biogenesis factor, the novelty of this work lies in the identification of a critical role for the lumenal amphipathic helix in potentially contributing to the execution of the INM-ONM fusion step. Although direct reconstitution of membrane fusion by Brl1 is beyond the scope of this study, some additional information about the AH could nonetheless be informative to the mechanism. For example, a clear implication of the model is that the AH might have a positive-membrane curvature sensing ability that could contribute to its sorting to an NPC biogenesis site and/or a putative direct role in membrane remodeling. This could be tested by AH floatation on liposomes of different curvatures. Such experiments could also incorporate testing the binding of the AH to liposomes of different lipid compositions. As several studies have implicated both DAG and PA as being important in the NPC biogenesis pathway, such experiments could also inform how local lipid composition might be used to recruit factors like Brl1 to the NPC assembly site as well.

Alternatively, in vivo experiments might also be helpful to provide insight into the function of the AH. For example, it remains unclear what causes the remarkable onion-like herniations, but the implication is that it is a combination of the sequestration of Brl1 and the overexpression of the dominant Brl1 that lacks a functional AH. It remains possible, for example, that the AH is required for the formation of the herniations even in this compromised state. Thus, a suggestion is to perform the overexpression of the I395D mutant in the context of the Auxin-induced degradation of Brl1. As additional cryo-FIB/ET might be too much to ask, even examining how the I395D mutant localizes at the level of fluorescence microscopy may be informative. i.e. is the AH required to recruit the Brl1I395D mutant to an NPC assembly site where further multimerization contributes to the elaborate hernia? Another alternative for in vivo experiments would be to replace the AH with other established curvature sensitive (or not) AHs and test their functionality in complementing viability and nup localization in the Brl1 knockout or ostir backgrounds.

*Reviewer #2 (Recommendations for the authors):*

1. The authors have previously established the KARMA methodology to characterize the NPC assembly process (Onischenko et al., 2020). Here, by similarly performing affinity purifications (APs) using distinct baits (Brl1, Nup170), they report a significant degree of post-lysis intermixing, i.e. NPC subunits exchanging between the lysate and the immunopurified complexes during the experimental procedure. Although they suggest that intermixing likely leads to an under-estimation of the labeling differences they observe for different NPC sub-complexes, this observation raises additional questions. Could the authors comment on the fact that APs performed using the scaffold subunit Nup170 as a bait show intermixing in control conditions (ethanol treatment, Figure S3D), while Mlp1 APs showed very low levels of exchanges in their previous study? In the case of Brl1 APs (Figure 2), intermixing could arise from the dynamic nature of Brl1 association with NPC intermediates. Is it observed for Brl1 when identified as a prey in Nups APs? While this observation does not challenge the authors' conclusions, which are validated by orthogonal approaches, the text should clearly warn about the necessity to systematically control for intermixing in such experiments.

2. Although Brl1 preferentially associates with early NPC intermediates (Figures 1-2), it is remarkable that Brl1 APs do not identify the transmembrane nucleoporins (Pom34 and Ndc1). Since detergent is added after lysis and removal of insoluble material by centrifugation (Methods section), it is possible that only a fraction of these transmembrane proteins are actually accessible during the pull-down. Detergent has been shown to be critical to solubilize Brl1 from cell lysates (see Figure 5 in Saitoh et al., 2005). Could the authors comment on this issue, which could potentially introduce biases in Brl1 APs?

3. The authors' data convincingly demonstrate that overexpression of the AH-deficient Brl1 mutant halts NPC biogenesis at a stage preceding membrane fusion, but do not prove that Brl1 directly acts as a fusogen. The title should thereby be modified accordingly (for example, an amphipathic helix in Brl1 is required for nuclear pore complex biogenesis in *S. cerevisiae*). Additional characterization of AH features is expected to provide mechanistic clues to Brl1 function in the fusion process:

– The AH motif could interact with the luminal side of the INM, stabilizing membrane curvature at NPC assembly sites, and/or bind to the luminal side of the ONM, favoring the fusion. Addressing these questions is challenging, yet the authors should test whether Brl1 AH interacts preferentially with small diameter, highly-curved liposomes in floatation assays (Figure 5E).

– Does Brl1 AH contribute to the interaction with Brr6, which was previously scored in immunoprecipitation experiments (Lone et al., 2015)? It should also be stated whether Brl1 partners (Brr6, Apq12) are identified among the factors enriched with early NPC intermediates (Figure S1A) or in Brl1 APs.

*Reviewer #3 (Recommendations for the authors):*

There are a few instances where corrections or improved data would improve the manuscript. These include the analysis of the in vivo interactions of the AH with membranes and interpretation of the Nup82 localization data in Figure 6D. These are detailed in specific comments to the authors.

1) As pointed out in the public review, a significant addition to the manuscript would be to examine the interactions of an ahBrl1 point mutant (e.g. Brl1(I395D)) with Nups using the KARMA approach. This could provide important insights into the nature of the Brl1(I395D) interactions with the observed NPC intermediates. I don't see these data as absolutely necessary for the manuscript, but they would provide important insight into the functional consequences of AH mutations.

2) In the Brl1(I395D)-mCherry expressing cells (Figure 6D), Nup82 appears reduced but not absent or excluded from areas where Brl1(I395D)-mCherry accumulates as suggested by the authors. This can be seen in the upper half of the Nup82-yeGFP image panel of Brl1(I395D)-mCherry expressing cells. Moreover, the maximum intensity plot of Nup82 shows a small but visible intensity increase at the NE. I suggest the authors address this point in their discussion of these data.

3) The authors' assessment of the in vivo interactions of the Brl1 AH with membranes is not strong. Localization of the ahBrl1-yEGFP fusion protein to the NE in cells (Figure 5F) does not show that the construct binds membranes in vivo. Curiously, the ahBrl1-yEGFP seems to bind the NE but not other membranes (e.g. the vacuolar membranes). This is especially surprising since the ahBrl1-yEGFP fusion protein is presumably positioned on the cytoplasmic side of the NE membrane. The significance of the membrane association of the ahBrl1-yEGFP construct would be more convincing if the I395D or F391D mutant were to not bind the membranes when expressed in cells.

4) It appears the 3h and 6h time point images in Figure 6B are reversed.

5) In the Discussion (pg. 22), the authors reference several manuscripts for their proposed models of "inside out" NPC assembly. It should be noted that these citations are preceded by studies in yeast where a similar model was proposed.

---

## [Author Response]

Essential revisions:There is considerable enthusiasm for the quality of the data presented in the manuscript and the conclusions, which are well justified. There was debate, however, over the relative advance of the work in comparison to recent studies that have also clearly implicated Brl1 as an NPC assembly factor. Thus, at least two of the reviewers felt that the major conceptual and mechanistic advances of this paper were centered on the role of the Brl1 amphipathic helix in NPC assembly, which remains ill-defined. Thus, in addition to addressing all of the concerns of the reviewers with relevant changes to the text or figures, the request is to provide some additional experimental support for the role that the AH plays in either membrane curvature and lipid sensing or in altering Brl1's interactions with nups.Reviewer #1 (Recommendations for the authors):1) Beyond the expert implication of Brl1 as an NPC biogenesis factor, the novelty of this work lies in the identification of a critical role for the lumenal amphipathic helix in potentially contributing to the execution of the INM-ONM fusion step. Although direct reconstitution of membrane fusion by Brl1 is beyond the scope of this study, some additional information about the AH could nonetheless be informative to the mechanism. For example, a clear implication of the model is that the AH might have a positive-membrane curvature sensing ability that could contribute to its sorting to an NPC biogenesis site and/or a putative direct role in membrane remodeling. This could be tested by AH floatation on liposomes of different curvatures. Such experiments could also incorporate testing the binding of the AH to liposomes of different lipid compositions. As several studies have implicated both DAG and PA as being important in the NPC biogenesis pathway, such experiments could also inform how local lipid composition might be used to recruit factors like Brl1 to the NPC assembly site as well.

We thank this and the other reviewers for their valuable suggestion that we further characterize the lipid binding of ahBrl1. We performed the requested experiment by testing the binding propensity of ahBrl1 to liposomes of different diameters in an in vitro flotation assay (new Figure 5F). Interestingly, ahBrl1 preferentially binds to highly curved lipid membranes. As reviewer 1 points out, this preference could contribute to the proper localization of Brl1 to early NPC assembly sites. Further, binding to highly curved lipid membranes also indicates that ahBrl1 potentially induces and/or stabilizes the high membrane curvature that is required for INM-ONM fusion.

We agree that testing different lipid compositions could be interesting. However, such experiments might be difficult to interpret since, aside from evidence that PA might be enriched at NPC assembly sites [7,8], the lipid composition at the inner NE and at NPC assembly sites remains uncertain.

We have added the experiment to Figure 5 (Figure 5F) and changed the text in the Results and Discussion section accordingly:

“Fusion of lipid membranes is typically accompanied by the formation of highly curved fusion intermediates. We therefore tested the affinity of ahBrl1 to membranes with different curvature in a liposome flotation assay with liposomes of various sizes (Figure 5F). Indeed, we observed that MBP-ahBrl1-yEGFP preferentially binds to highly curved liposomes with a diameter of 30 nm. Enrichment with 100 nm liposomes was less pronounced and binding to 400 nm liposomes did not exceed the level of the non-lipid binding control construct (MBP-yEGFP). Together, these results demonstrate that ahBrl1 binds to highly curved lipid membranes in vitro and is essential for cell viability.”

“The mechanisms by which Brl1 is recruited and concentrated at assembly sites is not clear but the unstructured N-terminus of Brl1 might contribute. This is supported by the non-punctate localization of Brr6 which contains only a short N-terminus [4]. A localization preference of Brl1 to highly curved membranes of INM evaginations could be an alternative explanation, which is supported by the binding preference of ahBrl1 to highly curved lipid-membranes (Figure 5F).”

“Interestingly, the surface of the highly curved liposomes to which ahBrl1 preferentially binds, topologically resemble the luminal side of INM evaginations. Both structures have a similar high positive curvature (Figure 3—figure supplement 1D, 5F and model in figure 8A). This raises the possibility that Brl1 can bind, and potentially induce and/or stabilize, highly curved, energetically unfavorable pre-fusion intermediates.”

Alternatively, in vivo experiments might also be helpful to provide insight into the function of the AH. For example, it remains unclear what causes the remarkable onion-like herniations, but the implication is that it is a combination of the sequestration of Brl1 and the overexpression of the dominant Brl1 that lacks a functional AH. It remains possible, for example, that the AH is required for the formation of the herniations even in this compromised state. Thus, a suggestion is to perform the overexpression of the I395D mutant in the context of the Auxin-induced degradation of Brl1. As additional cryo-FIB/ET might be too much to ask, even examining how the I395D mutant localizes at the level of fluorescence microscopy may be informative. i.e. is the AH required to recruit the Brl1I395D mutant to an NPC assembly site where further multimerization contributes to the elaborate hernia? Another alternative for in vivo experiments would be to replace the AH with other established curvature sensitive (or not) AHs and test their functionality in complementing viability and nup localization in the Brl1 knockout or ostir backgrounds.

We thank the reviewer for raising an interesting point regarding the role of endogenous and functional Brl1 in the development of the NPC assembly abnormalities induced by Brl1(I395D) overexpression. We performed the suggested experiment and monitored the localization of overexpressed Brl1(I395D) in cells inducibly depleted of the endogenous Brl1 by fluorescence microscopy. The respective figure panel has been added in Figure 6—figure supplement 1C. We did not observe any significant difference in the number of nuclear envelope foci containing Brl1(I395D), irrespective of whether endogenous Brl1 was degraded or not. Furthermore, the mislocalization of Nup82 is the same in the two conditions (Figure 5C). These results suggest that the NPC assembly defects, including the recruitment of Brl1(I395D) to NPC assembly sites and the formation of the multi-layered herniations, are not dependent on the presence of functional Brl1.

Reviewer #2 (Recommendations for the authors):1. The authors have previously established the KARMA methodology to characterize the NPC assembly process (Onischenko et al., 2020). Here, by similarly performing affinity purifications (APs) using distinct baits (Brl1, Nup170), they report a significant degree of post-lysis intermixing, i.e. NPC subunits exchanging between the lysate and the immunopurified complexes during the experimental procedure. Although they suggest that intermixing likely leads to an under-estimation of the labeling differences they observe for different NPC sub-complexes, this observation raises additional questions. Could the authors comment on the fact that APs performed using the scaffold subunit Nup170 as a bait show intermixing in control conditions (ethanol treatment, Figure S3D), while Mlp1 APs showed very low levels of exchanges in their previous study? In the case of Brl1 APs (Figure 2), intermixing could arise from the dynamic nature of Brl1 association with NPC intermediates. Is it observed for Brl1 when identified as a prey in Nups APs? While this observation does not challenge the authors' conclusions, which are validated by orthogonal approaches, the text should clearly warn about the necessity to systematically control for intermixing in such experiments.

We are grateful to the reviewer for raising the topic of post-lysis intermixing and its influence on the interpretation of the KARMA readouts. First, we would like to note that when comparing the lysate intermixing results for the Mlp1 bait in [6] and the Nup170 bait in this study (Figure 4—figure supplement 1), one has to consider the different scaling used in the plots. While in Onischenko et al. [6] the labeling in the intermixing experiment is plotted directly (ranging from 0 – 0.5), the current study expresses the intermixing as the extent of labeling compared to the average labeling for non-specifically isolated proteins (0% – 100%). Nevertheless, we fully agree with the reviewer that the intermixing rates with the Nup170 bait are significantly higher than what was observed in the Mlp1 AP. We believe that these differences arise from the extremely late recruitment of Mlp1 to the NPC [6] effectively only probing the pool of mature and stable complexes. This is in stark contrast to the Nup170 APs that probe a significant pool of early assembly intermediates that contain Nup170 [6], which are likely more dynamic. This is further supported by our comparative analysis of intermixing rates in Nup170 APs in NPC assembly intermediates halted upon Brl1 depletion, where we observe a more dynamic behavior (Figure 4—figure supplement 1D). Of note, the prime reason why the NPC assembly model was tested for resilience to the Mlp1 AP post-lysis intermixing in [6] was the outstandingly large maturation pools for Mlp1, where the effect of post-lysis intermixing in absolute terms is much bigger compared to all other baits.

To acknowledge the differences in lysate intermixing rates, we added the following note to the *“Lysate intermixing assays*” appendix detailing the purpose and interpretation of the lysate intermixing assays:

“The high intermixing rates could also be contributed by a lower stability of early NPC assembly intermediates bound by Brl1 compared to the mature NPCs (Figure 4—figure supplement 1D). This view is supported (i) by considerably higher intermixing rates for NUPs co-isolated with the early tier Nup170 bait upon Brl1 depletion (Figure 4—figure supplement 1D) and (ii) the much higher NUP dynamic exchange rates observed in the early tier Nup170 APs (~60%) compared to previously reported for the outstandingly late recruited Mlp1 bait (~20%, [6]), which likely isolates a population of otherwise completely assembled NPCs.”

In KARMA assays proteins can dynamically exchange between bait-bound fractions of NPC assembly intermediates and fractions not normally bound by the bait, or just within bait-bound fractions (Appendix 1—Figure 1E). The labeling in the KARMA assays is only sensitive to intermixing happening between unbound and bait-bound fractions, where any dynamic exchange diminishes the observed labeling deviations from the source cell lysate. At the same time, our lysate intermixing assays test the total extent of intermixing, including that within the bait-bound fractions (to which KARMA readouts are immune). Therefore, there is no contradiction between the high in vitro intermixing rates that are observed for NUPs in the Brl1 APs (Appendix 1—Figure 1C) and the over-labelling of the first tier NUPs that was observed in the KARMA assays with Brl1 (Figure 2B and 2D). The only reasonable explanation for these observations is that Brl1 dynamically interacts with immature NUP complexes. To further support this conclusion, we followed the reviewer's suggestion and re-analysed the KARMA data [6] for Brl1 as a prey. We observe that Brl1 indeed shows the signature of a dynamic protein, with labeling patterns similar to dynamic NUPs and NTRs (Author response image 1).

We added following lines to the results chapter “*Brl1 binds to assembling nuclear pore complexes*” that addresses the likely origin of high lysate intermixing in Brl1 APs:

“Despite the different labeling rates in KARMA assays, the analysis of Brl1 APs from mixtures of labeled and unlabeled yeast lysates showed almost complete intermixing of Brl1-bound NUP complexes during the AP procedure, pointing to a highly dynamic binding of Brl1 to NPC assembly intermediates (for details see Appendix 1 section: “Lysate intermixing assays”).”

Lastly, we have commented on the impact that post-lysis intermixing has on the evaluation of KSM parameters in Appendix 1 – “Kinetic state modeling”, and added a figure panel (Appendix 1—figure 1E) to highlight the different forms of dynamic exchange and to illustrate the effect of post-lysis intermixing on the KARMA readouts:

“For the quantitative interpretation of KARMA readouts, it is important to consider the effects of dynamic exchange. In KARMA assays, the intermixing between assembly intermediates bound by the bait and those that are not (Appendix 1—figure 1E, red arrows) would diminish deviations of labeling rates from the source cell lysates. It is therefore conceivable that our labeling values (Figure 2D and 4C) are underestimated due to post-lysis intermixing. Although our lysate intermixing assays indeed show high exchange rates of Brl1 bound NUPs (Appendix 1—figure 1C), dynamic exchange within the bait-bound fraction (Appendix 1—figure 1E, green arrows), which does not have an influence on the labeling in KARMA assays, contributes to labeling in the intermixing assay. In the future, for more accurate evaluation of labeling kinetics it might be advantageous to minimize post-lysis intermixing by a mild fixation step similar to what has been done before [9,10].”

2. Although Brl1 preferentially associates with early NPC intermediates (Figures 1-2), it is remarkable that Brl1 APs do not identify the transmembrane nucleoporins (Pom34 and Ndc1). Since detergent is added after lysis and removal of insoluble material by centrifugation (Methods section), it is possible that only a fraction of these transmembrane proteins are actually accessible during the pull-down. Detergent has been shown to be critical to solubilize Brl1 from cell lysates (see Figure 5 in Saitoh et al., 2005). Could the authors comment on this issue, which could potentially introduce biases in Brl1 APs?

It is unlikely that membranes are removed in the centrifugation step that we perform after the mechanical bead-beater lysis, as this is a very mild centrifugation (30 s at 15’000 g). Common membrane pelletting protocols in yeast lysates employ an ultracentrifugation step for 30 min with higher (~32,000 g) forces than the one we use [11]. Furthermore, detergents are added during the affinity purification procedure , likely solubilizing Brl1-bound complexes.

In support of the above, the transmembrane NUP Pom152, which tightly interacts with Pom34 and Ndc1 [12], is among the proteins with the most detected precursors in all our MS assays, and Ndc1 and Pom34 precursors are also detected (Author response image 1). Thus, we conclude that both Ndc1 and Pom34 are present in our AP eluates, but, unfortunately, they do not produce high-quality precursors, precluding a reliable quantification of their labeling kinetics. These proteins are likely inaccessible to our MS analysis due to a combination of factors: first, both of them are transmembrane proteins with a considerable portion of their sequence made up from helical transmembrane segments. Peptides resulting from these hydrophobic stretches are notoriously difficult to assay by MS [13]. In addition, Pom34 is a very small protein (34 kDa), further reducing the number of potentially good quality precursors. Finally, Ndc1 dynamically associates with the NPC [6,10] and probably has a lower binding affinity than stably incorporated NUPs.

3. The authors' data convincingly demonstrate that overexpression of the AH-deficient Brl1 mutant halts NPC biogenesis at a stage preceding membrane fusion, but do not prove that Brl1 directly acts as a fusogen. The title should thereby be modified accordingly (for example, an amphipathic helix in Brl1 is required for nuclear pore complex biogenesis in *S. cerevisiae*). Additional characterization of AH features is expected to provide mechanistic clues to Brl1 function in the fusion process:

We agree and modified the title according to the reviewers suggestion to:

“An Amphipathic Helix in Brl1 is Required for Nuclear Pore Complex Biogenesis in *S. cerevisiae*”.

– The AH motif could interact with the luminal side of the INM, stabilizing membrane curvature at NPC assembly sites, and/or bind to the luminal side of the ONM, favoring the fusion. Addressing these questions is challenging, yet the authors should test whether Brl1 AH interacts preferentially with small diameter, highly-curved liposomes in floatation assays (Figure 5E).

Thank you for this suggestion. We have now performed these experiments. Please see our response to reviewer 1 point 1.

– Does Brl1 AH contribute to the interaction with Brr6, which was previously scored in immunoprecipitation experiments (Lone et al., 2015)? It should also be stated whether Brl1 partners (Brr6, Apq12) are identified among the factors enriched with early NPC intermediates (Figure S1A) or in Brl1 APs.

As the reviewer pointed out, an interaction between Brr6 and Brl1 was previously shown by immunoprecipitation [4,5]. It is plausible that Brl1 AH contributes to the interaction with Brr6, yet, we don’t have any data to either support or reject this hypothesis. Brr6 and Apq12 are not detected in any of our MS assays, neither with the Brl1 bait nor with any of the NUP baits that were used in our previous study. This might be because they are very small proteins (Brr6: 22 kDa, Apq12: 16 kDa) largely composed of transmembrane segments, which are -as discussed above- difficult to assay with MS (see Reviewer 2, point 2). Of note, most of the precursors that we identified for Brl1 stem from its long N-terminus that is entirely missing from Brr6 or Apq12 (Author response image 1).

**Author response image 1. sa2fig1:** 

Reviewer #3 (Recommendations for the authors):There are a few instances where corrections or improved data would improve the manuscript. These include the analysis of the in vivo interactions of the AH with membranes and interpretation of the Nup82 localization data in Figure 6D. These are detailed in specific comments to the authors.1) As pointed out in the public review, a significant addition to the manuscript would be to examine the interactions of an ahBrl1 point mutant (e.g. Brl1(I395D)) with Nups using the KARMA approach. This could provide important insights into the nature of the Brl1(I395D) interactions with the observed NPC intermediates. I don't see these data as absolutely necessary for the manuscript, but they would provide important insight into the functional consequences of AH mutations.

We agree that this would be a very interesting experiment. However, we have seen that overexpressed Brl1(I395D) strongly accumulates in large multi-layered herniations and is thus spatially separated from the NPC structure (Figure 6). Therefore, we expect only a minor enrichment of NPC components in APs with Brl1(I395D). Furthermore, performing KARMA assays with the Brl1(I395D) bait only provides information until the stage where NPC assembly is blocked. To get interpretable labeling kinetics for NUP incorporation rates we would need data for the whole population of NPCs. This would be obtainable by performing KARMA assays with a NUP bait, similar to what we did in the Brl1-depleted conditions, upon overexpression of Brl1(I395D). However, our fluorescence microscopy analysis suggests that the NPC assembly intermediates at the base of the NE herniations in Brl1-depleted conditions and upon overexpression of Brl1(I395D) are similar, with the central scaffold NUPs being present and the cytoplasmic face of the NPC being absent (Figure 4D and Figure 6D). Thus, it is likely that our KARMA assays in Brl1(I395D)-overexpressing cells would look very similar to the Brl1 depletion condition that we already investigated (Figure 4). Because of this and the time-consuming nature of this experiment, we decided to not perform the KARMA analysis in Brl1(I395D)-expressing cells.

2) In the Brl1(I395D)-mCherry expressing cells (Figure 6D), Nup82 appears reduced but not absent or excluded from areas where Brl1(I395D)-mCherry accumulates as suggested by the authors. This can be seen in the upper half of the Nup82-yeGFP image panel of Brl1(I395D)-mCherry expressing cells. Moreover, the maximum intensity plot of Nup82 shows a small but visible intensity increase at the NE. I suggest the authors address this point in their discussion of these data.

We agree that Nup82 is not *entirely* absent from NE areas with Brl1(I395D)-mCherry puncta, as originally stated. However, the overall intensity at the NE (including NE areas with Brl1(I395D)-mCherry puncta) is strongly reduced. The increase in intensity at the NE can be explained by the higher intensity in the nucleus (compared to the lineplots of the other Nups which also show an increased intensity level in the nucleus). We changed the text from:

“In contrast, Nup82 is entirely absent from NE-areas with Brl1(I395D)-mCherry puncta.”

to

“In contrast, Nup82 intensity at NE-areas with Brl1(I395D)-mCherry puncta was strongly reduced.”

3) The authors' assessment of the in vivo interactions of the Brl1 AH with membranes is not strong. Localization of the ahBrl1-yEGFP fusion protein to the NE in cells (Figure 5F) does not show that the construct binds membranes in vivo. Curiously, the ahBrl1-yEGFP seems to bind the NE but not other membranes (e.g. the vacuolar membranes). This is especially surprising since the ahBrl1-yEGFP fusion protein is presumably positioned on the cytoplasmic side of the NE membrane. The significance of the membrane association of the ahBrl1-yEGFP construct would be more convincing if the I395D or F391D mutant were to not bind the membranes when expressed in cells.

We agree with the reviewer that this experiment is not very conclusive, and we have therefore removed the panel entirely (see also response to reviewer 1 points 1 and 4). Instead, we now include additional in vitro data demonstrating that ahBrl1 preferentially binds to highly curved membranes.

4) It appears the 3h and 6h time point images in Figure 6B are reversed.

Thank you for pointing this out. We corrected the figure panels.

5) In the Discussion (pg. 22), the authors reference several manuscripts for their proposed models of "inside out" NPC assembly. It should be noted that these citations are preceded by studies in yeast where a similar model was proposed.

Originally, we only cited the review by Thaller and Lusk 2018 [14] to acknowledge previous studies proposing an inside-out assembly mode but now include the primary literature in which inside-out assembly was suggested:

“Thus, our data support an inside-out mode of interphase NPC assembly, similar to previous observations in yeast and mammalian cells [15-17] [6,18-20].”

References

1. Saitoh, Y.H.; Ogawa, K.; Nishimoto, T. Brl1p -- a novel nuclear envelope protein required for nuclear transport. *Traffic* 2005, *6*, 502-517, doi:10.1111/j.1600-0854.2005.00295.x.

2. de Bruyn Kops, A.; Guthrie, C. An essential nuclear envelope integral membrane protein, Brr6p, required for nuclear transport. *EMBO J* 2001, *20*, 4183-4193, doi:10.1093/emboj/20.15.4183.

3. Liu, G.; Yong, M.Y.; Yurieva, M.; Srinivasan, K.G.; Liu, J.; Lim, J.S.; Poidinger, M.; Wright, G.D.; Zolezzi, F.; Choi, H.; et al. Gene Essentiality Is a Quantitative Property Linked to Cellular Evolvability. *Cell* 2015, *163*, 1388-1399, doi:10.1016/j.cell.2015.10.069.

4. Lone, M.A.; Atkinson, A.E.; Hodge, C.A.; Cottier, S.; Martinez-Montanes, F.; Maithel, S.; Mene-Saffrane, L.; Cole, C.N.; Schneiter, R. Yeast Integral Membrane Proteins Apq12, Brl1, and Brr6 Form a Complex Important for Regulation of Membrane Homeostasis and Nuclear Pore Complex Biogenesis. *Eukaryot Cell* 2015, *14*, 1217-1227, doi:10.1128/EC.00101-15.

5. Zhang, W.; Neuner, A.; Ruthnick, D.; Sachsenheimer, T.; Luchtenborg, C.; Brugger, B.; Schiebel, E. Brr6 and Brl1 locate to nuclear pore complex assembly sites to promote their biogenesis. *J Cell Biol* 2018, *217*, 877-894, doi:10.1083/jcb.201706024.

6. Onischenko, E.; Noor, E.; Fischer, J.S.; Gillet, L.; Wojtynek, M.; Vallotton, P.; Weis, K. Maturation Kinetics of a Multiprotein Complex Revealed by Metabolic Labeling. *Cell* 2020, *183*, 1785-1800 e1726, doi:10.1016/j.cell.2020.11.001.

7. Zhang, W.; Khan, A.; Vitale, J.; Neuner, A.; Rink, K.; Luchtenborg, C.; Brugger, B.; Sollner, T.H.; Schiebel, E. A short perinuclear amphipathic α-helix in Apq12 promotes nuclear pore complex biogenesis. *Open Biol* 2021, *11*, 210250, doi:10.1098/rsob.210250.

8. Thaller, D.J.; Tong, D.; Marklew, C.J.; Ader, N.R.; Mannino, P.J.; Borah, S.; King, M.C.; Ciani, B.; Lusk, C.P. Direct binding of ESCRT protein Chm7 to phosphatidic acid-rich membranes at nuclear envelope herniations. *J Cell Biol* 2021, *220*, doi:10.1083/jcb.202004222.

9. Subbotin, R.I.; Chait, B.T. A pipeline for determining protein-protein interactions and proximities in the cellular milieu. *Mol Cell Proteomics* 2014, *13*, 2824-2835, doi:10.1074/mcp.M114.041095.

10. Hakhverdyan, Z.; Molloy, K.R.; Subbotin, R.I.; Fernandez-Martinez, J.; Chait, B.T.; Rout, M.P. Measuring in vivo protein turnover and exchange in yeast macromolecular assemblies. *STAR Protoc* 2021, *2*, 100800, doi:10.1016/j.xpro.2021.100800.

11. Panaretou, B.; Piper, P. Isolation of yeast plasma membranes. In *Yeast Protocol*; Springer: 2006; pp. 27-32.

12. Onischenko, E.; Stanton, L.H.; Madrid, A.S.; Kieselbach, T.; Weis, K. Role of the Ndc1 interaction network in yeast nuclear pore complex assembly and maintenance. *J Cell Biol* 2009, *185*, 475-491, doi:10.1083/jcb.200810030.

13. Eichacker, L.A.; Granvogl, B.; Mirus, O.; Muller, B.C.; Miess, C.; Schleiff, E. Hiding behind hydrophobicity. Transmembrane segments in mass spectrometry. *J Biol Chem* 2004, *279*, 50915-50922, doi:10.1074/jbc.M405875200.

14. Thaller, D.J.; Patrick Lusk, C. Fantastic nuclear envelope herniations and where to find them. *Biochem Soc Trans* 2018, *46*, 877-889, doi:10.1042/BST20170442.

15. Wente, S.R.; Blobel, G. A temperature-sensitive NUP116 null mutant forms a nuclear envelope seal over the yeast nuclear pore complex thereby blocking nucleocytoplasmic traffic. *J Cell Biol* 1993, *123*, 275-284, doi:10.1083/jcb.123.2.275.

16. Murphy, R.; Watkins, J.L.; Wente, S.R. GLE2, a *Saccharomyces cerevisiae* homologue of the Schizosaccharomyce *S. pombe* export factor RAE1, is required for nuclear pore complex structure and function. *Mol Biol Cell* 1996, *7*, 1921-1937, doi:10.1091/mbc.7.12.1921.

17. Zabel, U.; Doye, V.; Tekotte, H.; Wepf, R.; Grandi, P.; Hurt, E.C. Nic96p is required for nuclear pore formation and functionally interacts with a novel nucleoporin, Nup188p. *J Cell Biol* 1996, *133*, 1141-1152, doi:10.1083/jcb.133.6.1141.

18. Makio, T.; Stanton, L.H.; Lin, C.C.; Goldfarb, D.S.; Weis, K.; Wozniak, R.W. The nucleoporins Nup170p and Nup157p are essential for nuclear pore complex assembly. *J Cell Biol* 2009, *185*, 459-473, doi:10.1083/jcb.200810029.

19. Marelli, M.; Lusk, C.P.; Chan, H.; Aitchison, J.D.; Wozniak, R.W. A link between the synthesis of nucleoporins and the biogenesis of the nuclear envelope. *J Cell Biol* 2001, *153*, 709-724, doi:10.1083/jcb.153.4.709.

20. Otsuka, S.; Bui, K.H.; Schorb, M.; Hossain, M.J.; Politi, A.Z.; Koch, B.; Eltsov, M.; Beck, M.; Ellenberg, J. Nuclear pore assembly proceeds by an inside-out extrusion of the nuclear envelope. *ELife* 2016, *5*, doi:10.7554/*eLife*.19071.